

**Sensitivity of a Q-ACSM to chamber generated SOA with different**
**oxidation states**
Xiaoxiao Li[1], Yan Ma[2,3*], Hui Chen[2,3], Youling Jiang[2,3], Xin Ma[2,3], Rujin Yin[1], Dongsen Yang[2,3],
Xiaowen Shi[2,3], Jiming Hao[1], Jingkun Jiang[1], and Jun Zheng[2,3*]
[1]State Key Joint Laboratory of Environment Simulation and Pollution Control, School of
Environment, Tsinghua University, 100084 Beijing, China
[2]Collaborative Innovation Center of Atmospheric Environment and Equipment Technology,
Nanjing University of Information Science & Technology, Nanjing 210044, China
[3]Joint Laboratory for Air Quality and Climate, Nanjing University of Information Science &
Technology, Nanjing 210044, China
*Corresponding authors: Drs. Jun Zheng and Yan Ma*
*Email: zheng.jun@nuist.edu.cn and mayan@nuist.edu.cn*
*Address: School of Environmental Science and Engineering, Nanjing University of Information*
*Science & Technology, Nanjing 210044, China*
*Tel.: +86-18251919852*
*Fax: +86-25-58731090*
**Abstract**
The accuracy in quantification of secondary organic aerosols (SOA) using a Q-ACSM has
been comprehensively investigated in this work. SOA samples were generated under simulated
photochemical oxidation conditions in a 4.5 m$^3$ Teflon chamber from three different volatile
organic compounds (VOC) of atmospheric relevant concentrations (dozens of ppbv): α-pinene,





isoprene, and toluene, representing both biogenic and anthropogenic VOC. Different SOA
oxidation states were achieved by changing the relative ratio of the VOC precursor to the oxidants
($O_3$ or OH). A scanning mobility particle sizer (SMPS) and an aerosol particle mass analyzer
(APM) were used to determine the number-size distribution and the exact mass of the chamber-
generated SOA, which were then used to deduce the SOA effective density and mass concentration.
Results showed that aerosol mass concentration measured by the Q-ACSM based on SMPS
calibration alone may be associated with considerable errors due to the fact that the effective density
of SOA at different oxidation state can change substantially. More importantly, the sensitivity of
the Q-ACSM to a specific type of SOA was found to be anti-correlated with the aerosol oxidation
state regardless of the VOC precursors. This may be due to the decreasing of relative ionization
efficiency (RIE) or the collection efficiency (CE) of the Q-ACSM for more oxidized SOA. To
pinpoint the actual cause, ammonium sulfate (($NH_4$)$_2SO_4$) seed particles were injected into the
chamber before SOAs were produced and the CE for a specific SOA sample was hence determined
with reference to the changes in sulfate signals. Our experiment results along with previous
literature reports strongly implied that as the SOA oxidation state increases, SOA will transform
gradually from a liquid state (CE $\approx$ 1) into a solid (or glassy) state with a CE of 0.2~0.5. Meanwhile,
the RIE of OA decreased substantially when SOA transformed from hydrocarbon-like OA (HOA)
into more oxygenated OA (OOA) and may further decrease as O/C continued to increase. Our
results indicated that the current Q-ACSM calibration procedure using a constant RIE may lead to
somewhat underestimation of more oxidized OOA but overestimation of less oxidized HOA, i.e.,
a variable RIE shall be applied, most likely as a function of the SOA oxidation state.

**Key words:**  Organic Aerosol; ACSM Calibration; Relative Ionization Efficiency; Collection
Efficiency; Effective Density.



# 1. Introduction


Organic aerosol (OA) have been recognized as a major component in ambient particles,
contributing 20-90% to the total submicron particles around the world (Hallquist et al., 2009a;
Kanakidou et al., 2005; Salcedo et al., 2006; Zhang et al., 2007), which can substantially affect the
climate directly by interacting with solar radiation and indirectly by affecting cloud microphysics
(IPCC, 2014). Recent studies demonstrated that haze events in China were largely driven by
secondary organic aerosol (SOA) (Huang et al., 2014) and may exert significant adverse effects on
human health (Poschl, 2005; Poschl and Shiraiwa, 2015). The chemical composition and the
dramatic changes of OA in ambient remain less understood compared to the inorganic species.
Measurements of OA has been a challenging task not only for the fact that OA contains nearly
countless chemical species but also due to its relatively short lifetime and rapid transformation in
the atmosphere. Traditionally, the sampling and the ensuing chemical analyses of aerosol
composition are mainly utilizing filter based offline methods, followed by analysis with gas
chromatography/mass spectrometry (GC/MS), liquid chromatograph/mass spectrometry (LC/MS),
nuclear magnetic resonance(NMR) or Fourier transform infrared spectroscopy (FTIR). Evidently,
offline techniques cannot catch the dramatic change of ambient OA (Hallquist et al., 2009b).
Therefore, many in-situ instrumentations have been developed to conduct aerosol measurements in
real time. For example, the particles-into-liquid sampler system (PILS) (Sorooshian et al., 2006)
firstly strips out the gas-phase pollutants and dissolves remaining particulate samples into water
solutions, which are then send to ion chromatography for further analyses. Clearly, PILS can
prevent sample loss by eliminating pre-processing processes. However, only water-soluble
components can be analyzed by PILS and no size-resolved measurement can be achieved (Orsini
et al., 2003; Weber et al., 2001). Similarly, another widely used thermal-optical organic
carbon/elementary carbon (OC/EC) analyzers can only quantify total OC with very limited
information (Birch and Cary, 1996).



Recently, a new powerful method, known as aerosol mass spectrometry (AMS) has been
successfully developed and widely deployed to do size-resolved aerosol measurements in real time
(Jayne et al., 2000). A typical AMS is equipped with a set of aerodynamic lens (Liu et al., 1995a,
b) to effectively focus and transmit particles (~50 - 1000 nm) ($PM_1$) into the instrument, a time-of-
flight (ToF) chamber to determine the aerosol size, a thermal vaporizer (heated to ~600°C) to
evaporate non-refractory (NR) components into the gas-phase, and a 70 eV electron impact (EI)
ionization source to ionize the gaseous samples before they can be analyzed by a mass analyzer
(either a quadrupole or a time-of-flight mass spectrometer) (DeCarlo et al., 2004; Jayne et al.,
2000). Evidently, compared to previous techniques, AMS can provide elemental composition of
the organic species and can achieve much higher time and size resolution. More recently, a newer
version of AMS, i.e., the aerosol chemical speciation monitor (ACSM) was developed (Ng et al.,
2011). ACSM is basically a simplified AMS without the aerosol ToF chamber and thus is much
smaller and affordable. Although ACSM cannot obtain high resolution mass spectra, elementary
information can still be readily recovered from some symbolic fragments, such as $C_nH_{2n+1}$ (*m/z* 27,
29, 41, 43, 55, 57, 69…) and $CO_2^+$ (*m/z* 44), representing hydrocarbon-like organic aerosol (HOA)
and oxygenated organic aerosol (OOA) (Ng et al., 2011). Therefore, ACSM is especially suitable
for long term field operation. Nevertheless, Aerodyne AMS/ACSM have been widely used to
conduct researches on $NR-PM_1$ around the world in both field and chamber studies (Zhang et al.,
2007; Zhou et al., 2016).
Although the performance of AMS/ACSM has been demonstrated to be in accordance with
many other measuring techniques (Drewnick et al., 2003; Jimenez et al., 2016; Kondo et al., 2007;
Takegawa et al., 2005), the quantification of aerosol composition by AMS/ACSM still needs to be
further refined. In theory, the sensitivity of AMS is affected by the particle transmission efficiency
through the aerodynamic lens, the ion transmission efficiency inside the mass analyzer, the particle
collection efficiency (CE) by the vaporizer due to the bouncing effect, the ionization efficiency (IE)





and the possible fragmentation during the thermo-vaporization and EI ionization (70eV) (Allan et
al., 2003; Canagaratna et al., 2015; Jayne et al., 2000; Jimenez et al., 2003). In practice, AMS
calibrations in terms of CE and IE are mostly conducted with inorganic species only and the
calibration factors for organics are indirectly inferred and can be highly uncertain (Jimenez et al.,

2016).

It has been suggested that CE can be affected by particle chemical composition, particle phase,

particle size as well as RH. Based on inter-comparisons in sulfate measurements between AMS,
PILS-IC and other instruments, a CE of 0.5 for all compounds has been recommended with the
assumption that particles are internally mixed (Drewnick et al., 2003; Takegawa et al., 2005).
Although organics quantified using a CE of 0.5 has been found correlating well with independent
OC or VOCs measurement in most field works (Allan et al., 2004; de Gouw et al., 2005; Takegawa
et al., 2005; Venkatachari et al., 2006), studies have shown that CE can vary substantially for
various chamber generated organic particles (Bahreini et al., 2005; Docherty et al., 2013).
Similarly, IE of inorganic components is usually calibrated directly with ammonium sulfate
$((NH_4)_2SO_4)$ and ammonium nitrate $(NH_4NO_3)$ aerosols of known mass concentration, while the
organic components is assigned with a constant relative ionization efficiency (RIE, i.e., the ratio of
the electron impact ionization efficiency of a given species to the measured ionization efficiency
of nitrate on a per unit mass basis) of 1.4. However, RIE of organic aerosols ($RIE_{org}$) has been
suggested to be significantly different for different OA species (Murphy, 2016). For instance, it
appears that the RIE values of primary OA (POA) are significantly different from those of SOA
(Dzepina et al., 2007; Jimenez et al., 2016; Slowik et al., 2004). Any uncertainty associated with
the $RIE_{org}$ may lead to erroneous AMS/ACSM measurement results, especially in the case of
interpreting aerosol samples from various environment around the world. Therefore,
comprehensive researches on the CE and RIE of different SOA species are of practical importance
to constrain the AMS/ACSM measurements. Some methods have been introduced to quantify or




eliminate the side-effects of RIE and CE in AMS measurements. For example, using laser-based
vaporizer makes it possible to directly measure aerosol CE (Cross et al., 2007). However, the laser
can only be used for particles larger than 250 nm. Most recently, a new type of capture vaporizer
has been developed to achieve a unit CE (Hu et al., 2017). However, the capture vaporizer will
increase the residence time of aerosol inside the vaporizer and thus change the fragmentation
pattern to produce many smaller fragments, which will highly complicate the AMS quantification
process.

In this work, the performance of a quadrupole based ACSM (Q-ACSM) was comprehensively

investigated for chamber-generated SOA samples under simulated photochemical oxidation
conditions, including both hydroxyl radical (OH) oxidations and ozonolysis ($O_3$). Three different
volatile organic compounds (VOC) (i.e., α-pinene, isoprene, and toluene) at atmospheric relevant
concentrations (dozens of ppbv) were chosen to represent both biogenic and anthropogenic VOC.
The SOA mass concentrations were directly measured by an aerosol particle mass analyzer (APM)
to achieve higher accuracy. The sensitivity of Q-ACSM to SOA at different oxidation state or O/C
ratio was quantified.
**2. Experimental Methods**
*2.1 Chamber Setup*

A 4.5 m$^3$ collapsible atmospheric-pressure fluoropolymer (Teflon) smog chamber (L = 1.8 m;

W = 1.5 m; H = 1.7 m) (see Fig. 1) was used to generate organic aerosols under atmospheric relevant
conditions. The chamber was essentially the same as the one used in our earlier work (Yao et al.,
2014; Yuan et al., 2017) and has been described in details previously. Before each set of the
experiments, the chamber was thoroughly cleaned by irradiation with black light UV-lamps and
exposure to a high concentration (a few ppmv) of $O_3$ for more than 6 hours. Before each experiment,
the chamber was flushed by pure air generated by a zero-air generator (Acdco 737) until less than



10 particles cm$^{-3}$ was detected inside the chamber. Reactants and scavengers (if used) were carried
into the chamber through a T-shaped glass bulb by pure air. The glass ball was heated gently to
insure completely injection. At the bottom center of the smog chamber was installed a Teflon coated
fan for rapid mixing of all reactants. The chamber temperature and RH were maintained at 20±1°C
and 10%-15%, respectively.

### *2.2 SOA Generation*

During each experiment, known amount of α-pinene (Sigma-Aldrich, >98%), isoprene
(Sigma-Aldrich, >99%), or toluene (Sigma-Aldrich, >99.5%) was firstly dissolved into
cyclohexane (TEDIA Inc., HPLC grade >99.5%) and then was injected into the chamber through
a stream of pure air. The concentrations of these precursor VOCs were set to atmospheric relevant
levels to make the results of this work more applicable to ambient measurements. Ozone was
generated by exposing pure oxygen ($O_2$) to a low-pressure mercury (Hg) lamp (Jelight, Model 600).
When $O_3$ was used as the oxidant, OH radical scavenger, cyclohexane, was injected into the
chamber before the experiment. When OH was used as the oxidant, trace amount of self-
synthesized methyl nitrite was firstly injected into the chamber and then the black light bulbs
around the chamber were turned on for several minutes to start the OH-initiated oxidations (Yao et
al., 2014). The mixing ratios of $O_3$ inside the chamber was monitored continuously by a Thermo
Fisher Scientific ozone monitor (Model 49i) throughout the experiment. Different SOA oxidation
states were achieved by changing the ratio between the VOC precursor and the oxidant ($O_3$ or OH).

### *2.3 Q-ACSM Operation*

The sampling interval of Q-ACSM was set to be 8-15 minutes, depending on the particle mass
concentration generated in the chamber. The sampling tube was made of a 6.3 mm OD, 0.8 m long
stainless-steel tube. One liter per minute (lpm) air was sampled from the chamber instead of the
original 3 lpm to minimize the total gases pulled out from the chamber. The Ionization efficiency



for $NO_3^-$ of Q-ACSM was calibrated before and after each experiment using $NH_4NO_3$ following
the procedure described by Ng et al. (2011). The measured organic mass concentration was
calculated using a default RIE value of 1.4 and a CE value of 0.5, which have been widely used
during most of the field and laboratory measurements (Canagaratna et al., 2007). The nitrogen ($N_2$)
peak ($m/z$ = 28) and the internal naphthalene standard peak ($m/z$ = 128) were used before the
experiment to do mass calibration. The mass dependent transmission efficiency of the Q-mass
spectrometer was also calibrated before data analyses.
*2.4 Aerosol Mass Measurement*
A scanning mobility particle sizer (SMPS), consisting of a differential mobility analyzer
(DMA, TSI Model 3081) and a condensation particle counter (CPC, TSI Model 3776), was used
for real-time monitoring of the particle number size distribution between 15 nm-650 nm particles
and aerosol volume concentrations were deduced assuming spherical shape. The time resolution
was 5 mins. An aerosol particle mass analyzer (APM, Kanomax Model 3601) combined with a
DMA (TSI Model 3081) and a CPC (TSI Model 3776) were used to determine the particle mass
and thus the particle effective density (ρ) was calculated from the measured mobility diameter
(DeCarlo et al., 2004; McMurry et al., 2002). During each experiment, the diameter of the particle
to be analyzed by the APM was manually set to be the same as the peak value of the particle number
size distribution measured by the SMPS, as the particles were continuously growing in the chamber.
One DMA-APM scan took about 5 mins. Hence, the particle mass concentration was calculated
from the measured particle volume concentration and the corresponding effective density.
**3. Results and Discussion**
*3.1 Determine the Q-ACSM Response Factor for Chamber Generated SOA*
The evolution of SOA during a typical chamber experiment was shown in Fig. 2, where 80
ppbv $O_3$ and 20 ppbv α-pinene were injected into the chamber. Shortly after injection, strong





nucleation event occurred and was marked by a typical banana-shaped three-dimensional (3-D)
number size distribution plot (Fig. 2a). Figure 2b showed the time series of aerosol mass
concentrations measured by Q-ACSM, integrated from SMPS measurements, and the
corresponding f44 (the ratio of *m/z* 44 signal over total organic ion signal intensity) and effective
density of aerosol calculated from the Q-ACSM and APM measurements, respectively. Since *m/z*
44 signal basically reflected the oxygen content in OA, the O/C ratio can be deduced directly from
f44, both of which have been widely used to represent the oxidation state of OA (Canagaratna et
al., 2015). The number size-distributions at one hour interval during the chamber experiment were
also depicted in Fig. 2c. Evidently, as the ozonolysis reaction proceeded, the size and mass
concentration of the aerosol swiftly increased, especially during the initial two hours. The effective
density of the formed aerosol also increased as the aerosol became more compact and approached
a spherical shape. The initial high values of f44 may be due to the fact that initially formed SOA
particles were highly oxidized because of gas/particle partitioning (Shilling et al., 2009a) . The f44
factor in the following period fluctuated between 0.115 and 0.135, indicating that the oxidation
state of aerosols did not change significantly within the experimental period. After 2~3 hours, the
aerosol size reached 80~100 nm, mass concentration increased to 40~50 $\mu g\ m^{-3}$, and the effective
density and f44 were about 1.2 g $cm^{-3}$ and 0.13, respectively. By then a relative steady-state was
reached and the response factor (RF) of the Q-ACSM was thus determined through a linear
correlation analysis using the relative steady-state data, as shown in Fig. 3.

In a similarly way, fourteen independent chamber experiments were conducted in this work.

Each experiment typically lasted for 4-6 hours until f44, effective density, and mass concentration
all reached relatively steady states. The slight variation of f44 after reaching relatively steady state
in each experiment was mainly due to the measurement uncertainty associated with the Q-ACSM.
The VOC used in this work included α-pinene, isoprene, and toluene, representing both biogenic
and anthropogenic VOC emissions. Both ozonolysis and OH initiated oxidation processes were



studied except that in the case of toluene only OH reactions were investigated. To generate SOA
samples under atmospheric relevant conditions and thus avoid secondary VOC oxidation products,
the mixing ratios of these VOC precursors ranged from 10 to 60 ppbv for α-pinene, 60 to 200 ppbv
for isoprene, and 30 to 60 ppbv for toluene. The oxidant concentrations were also limited to 50-80
ppbv for $O_3$ and 50-300 μL for methyl nitrite. The detailed experimental conditions and results
were listed in Table 1.
*3.2 Effects of f44*

To investigate the effects of oxidation state of the SOA samples (i.e., the measured f44) on

the Q-ACSM detection sensitivity, for each pair of VOC/oxidant several experiments were repeated
with different relative concentration ratio between the VOC and the oxidant. Evidently, the
characteristics of the generated SOA from the oxidation of isoprene, α-pinene, and toluene
appeared to be significantly different among various oxidation conditions. Especially the RF of the
Q-ACSM changed substantially for different f44. Both isoprene and α-pinene are relatively reactive
toward $O_3$ and OH. However, the dominant degradation processes in the atmosphere for isoprene
and α-pinene are typically through oxidations by OH and $O_3$, respectively. Especially, laboratory
studies have shown that the ozonolysis of α-pinene can lead to considerably higher SOA yield than
that from OH initiated reactions (Yao et al., 2014). Therefore, in this work we were focusing on
the isoprene-OH and α-pinene-$O_3$ reactions only. In the case of toluene, the experiment was
relatively straightforward since toluene only reacted with OH radicals. The oxidation states of SOA
generated in each experiment was shown in different colors in triangle plot (see Fig. 4). Since some
$O_3$ will be produced during OH initiated chain reactions (Finlayson-Pitts and Pitts, 1999), the
oxidation state of isoprene-generated SOA varied more significantly as the experiment proceed.
For α-pinene, however, f44 did not vary considerably as isoprene during the one experiment period
and in between experiments. Even when $O_3$ to α-pinene ratio was increased substantially, only
slight increase in f44 was observed, which was most likely due to the fact that the first-generation



oxidation products of α-pinene were mainly partitioned into the aerosol phase and cannot be further
oxidized by $O_3$. Nevertheless, the Q-ACSM RF for all chamber-generated SOA decreased linearly
as the f44 increased (see Fig. 5a).
The effective density of all chamber-generated SOA ranged between 1.09-1.36 g cm$^{-3}$, which
covered a much larger range compared to the reported values of 1.22-1.28 by Zelenyuk et al. (2008)
and 1.3±0.1 by Kiendler-Scharr et al. (2009) for biogenic SOA. Also, clearly shown in Fig. 5b was
that the SOA effective density increased linearly with increasing f44, most likely due to the fact
that when more oxidants were present, more highly oxygenated products were produced and led to
the formation of more compact SOA. Our results indicated that the oxidation state and effective
density of atmospheric SOA may vary significantly from different ambient oxidation environment
and can change dynamically at different stage of the aging process. Therefore, it was reasonable to
assume that quantification of ambient OA by Q-ACSM using a constant conversion factor may
induce significant error in aerosol mass concentration. Accordingly, the Q-ACSM RF for OA
should be systematically calibrated with laboratory-generated aerosols produced not only from
various VOC precursors but also under different atmospheric-relevant reaction conditions.
Typically, Q-ACSM RF for OA is determined indirectly using nitrate salt standards as
reference and can be mathematically expressed as the product of three factors:
$$RF = IE_{NO3} \cdot RIE_{org} \cdot CE_{org},$$    (E1)
i.e., the ionization efficiency of nitrate salt ($IE_{NO3}$), the relative ionization efficiency of OA ($RIE_{org}$),
and the collection efficiency of OA ($CE_{org}$) of the Q-ACSM. $IE_{NO3}$ can be calibrated before and
after the experiments and is independent from the properties of OA. However, the other two terms
may vary with different OA samples, the observed anti-correlation of RF with f44 could be due to
either CE or RIE.
***3.3 Effects of CE***





To investigate and evaluate the possibility that CE may contribute to the observed anti-
correlations between RF and f44 in this work, a set of chamber experiments using $(NH_4)_2SO_4$ (AS)
seed particles were conducted. The difference between this set of experiments from the previous
ones was that here nebulized dry AS seed particles were injected into the chamber before the SOA
was produced, i.e., the gas phase products will condense onto the AS seed surface instead of
initiating new particle formation. After fully coated with SOA, the measured sulfate mass
concentration by Q-ACSM will change due to variation in $CE_{SO4}$ and the CE of SOA material
($CE_{org}$) can be deduced accordingly. In each experiment, similar amount and size of AS seeds were
used to avoid other possible affecting factors. Figure 6 displayed the time series of a typical AS
chamber experiment. Initially, AS seed aerosol was injected and ACSM measured sulfate mass
concentration was about 35 μg m$^{-3}$. After VOC precursor was injected, the ACSM measured sulfate
increased rapidly to about 50 μg m$^{-3}$. As shown in Fig. 6, the sulfate signal increased substantially
after coated with SOA but the measured f44 did not change significantly as SOA was continuously
produced. Thus, $CE_{org}$ at certain f44 can be evaluated. Evidently, the accuracy of the absolute value
of $CE_{org}$ will depend on $CE_{SO4}$ and $RIE_{SO4}$. It is worth noting that $CE_{SO4}$ may vary from 0.2 to 1 as
a function of RH (Matthew et al., 2008). $CE_{SO4}$ was measured to be 0.28 in this work with RH<15%.
In the case of $RIE_{SO4}$, a value of 1.15 was used here as suggested by other studies (Canagaratna et
al., 2007; Ng et al., 2011; Petit et al., 2015). Although recent researches have shown that $RIE_{SO4}$
may vary from instrument to instrument (Budisulistiorini et al., 2014; Crenn et al., 2015), the exact
$RIE_{SO4}$ value would not affect the conclusion of this research. For the scope of this work, only the
relative changes in sulfate concentration were noted.
Total nine AS-chamber experiments were conducted in this work and the detailed experiment
conditions and results were listed in Table 2. However, $CE_{org}$ appeared to be affected by the SOA
coating thickness (see Fig. 7), which would decide the mixing state of the SOA coated AS particles.
When the coating material was not enough to fully cover the seed particles (dark blue points), there





was a possibility that the seed core would hit the collection surface directly and thus behaved as an
AS particle. To demonstrate this possibility, an experiment under extreme condition was conducted,
i.e., substantially excess VOC precursor was added. The result was indicated by the red point in
Fig. 7. This near unit CE (~0.97) suggested that the AS aerosols were fully covered with SOA and
behaved as a pure SOA, which may assume a "sticky" liquid state.  However, no higher f44 values
could be achieved with such large amount of organics generated, which was possibly related to the
loading-dependent gas-particle partitioning (Shilling et al., 2009b) and made it difficult to address
f44 influence on $CE_{org}$ via experiment. The coating layer of the light blue points should be thick
enough to cover the AS core, which was indicated by the decreasing $CE_{org}$ from 1 to ~0.5 with
increasing f44. However, there was still a possibility that the CE values of these light blue points
were the results of a combination of AS core and organic shell.

CE for biogenic SOA has been reported to be close to one based on both chamber experiments

(Kiendler-Scharr et al., 2009) and field measurements conducted in amazon, where aerosols were
dominated by liquid SOA (Allan et al., 2014; Chen et al., 2009). However recently, it has been
proposed in theoretical, chamber, and field studies that organic particles can exist in semi-solid or
solid state under ambient temperature, rather than been in liquid state (Shiraiwa et al., 2011; Vaden
et al., 2010; Virtanen et al., 2010).

The oxidation products of VOCs under ambient conditions are mainly consisted of carbonyl

compounds and carboxylic acids (Finlayson-Pitts and Pitts, 1999), the saturation vapor pressures
of which normally decrease with increasing oxidation level. As they are more oxidized, SOA may
transit from liquid phase gradually into solid phase under ambient temperature, which are consistent
with our observations of higher effective density at higher oxidation state. Accordingly, the surface
property of SOA can change dramatically and may induce considerable change in CE as SOA
transforms from a "sticky" liquid-drop into a "bouncing" solid-ball. Particle morphology research
based on glass transition temperature ($T_g$) indicated that during oxidation, the SOA particles will





change from liquid state to semi-solid state and finally to solid or glassy state (Koop et al., 2011),
resulting from a combined effect of increasing molecular weight and O/C. In previous studies, $CE_{org}$
has also been demonstrated to be variable for particles of different chemical composition, phase
and under different RH, with a value ranging from 0.2 to ~1 (Alfarra, 2004; Docherty et al., 2013).
*3.4 Effects of RIE*
RIE for a specific molecule can be evaluated as following (Canagaratna et al., 2007; Jimenez
et al., 2003):
$$RIEs = \frac{MW_{NO3}}{IE_{NO3}} \cdot \frac{IEs}{MWs} , \qquad (E2)$$
where RIEs is the relative ionization efficiency for a specific organic molecule S. IEs and MWs are
respectively the ionization efficiency and the molecular weight of S. Theoretically, IEs is directly
proportion to σ, i.e., the electron impact ionization cross section of the molecule, which is linearly
related to the number of electrons in the molecule. Since the number of electrons is roughly
proportional to the molecular weight, RIEs of molecules with similar structure and function groups
are suggested to be similar to each other. RIEs values for hydrocarbons and oxygenated species,
however, are believed to be different since their oxygen contents can vary substantially
(Canagaratna et al., 2007), ranging from less than 1 to more than 3 (Dzepina et al., 2007; Jimenez
et al., 2016; Slowik et al., 2004). Consequently, the anti-correlation between RF and f44 may also
due to different RIEs related to the oxygen contents.
**4. Conclusion**
The sensitivity of Q-ACSM to chamber-generated SOA in various oxidation states was
comprehensively investigated and an anti-correlation between the instrument sensitivity, RF, and
SOA oxidation state, represented by f44, was obtained regardless of the type of VOC precursors.
Therefore, our results strongly indicated that ambient OA measurements by Q-ACSM using a





constant conversion factor may induce significant error in aerosol mass concentration. Accordingly,
the Q-ACSM RF for OA should be systematically calibrated with laboratory-generated aerosols
produced not only from various VOC precursors but also under different atmospheric-relevant
reaction conditions. Based on our chamber experiment results and previously reported observation
in chamber and ambient studies, a comprehensive view of $RF_{org}$, $RIE_{org}$, $CE_{org}$, and $\rho_{effective}$ of OA
at different oxidation state (indicated by f44) was proposed (see Fig. 8). It was reasonably to assume
that as a SOA particle was in low oxidation state, it was basically in liquid state with a CE of close
to one. With the increase of O/C, the liquid state slowly changed into semi-solid and finally the
solid/glassy state with a CE ranging from 0.2~0.5. The RIE of organics would decrease
substantially from hydrocarbon-like compounds to oxygen containing compounds but would only
continue to decrease at a slower rate with further increasing O/C. The observed anti-correlation
between RF and O/C in this work can be explained by the combined effects of CE and RIE. Our
results suggested that under certain circumstance a Q-ACSM calibrated using the traditional
method may underestimate OOA content but overestimate HOA in previous studies. Accordingly,
different $RIE_{org}$ values should be used for HOA and OOA. In addition, early AMS calibrations
based on SMPS measured mass concentration may be associated with considerable errors due to
the fact that the effective density of SOA at different oxidation state can change substantially.
***Acknowledgements***

This work is supported by the National Key Research and Development Project

(2016YFC0202401), National Natural Science Foundation of China (41575122, 41675126, and
41730106), and the Priority Academic Program Development of Jiangsu Higher Education
Institutions. The data used are listed in the tables and references. The data used in this work are
available from the authors upon request (zheng.jun@nuist.edu.cn).





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





**Table 1. Summary of Chamber Experiment Conditions.**

| Precursor | # | VOC (ppbv) | O₃/OH [1] (ppbv/μL) | PM (μg m⁻³) [2] | ρ (g cm⁻³) [3] | f44 (%) [4] | ACSM/SMPS [5] | R² |
|---|---|---|---|---|---|---|---|---|
| | 1 | 200 | 300 μL | 140 | 1.09 | 0.057 | 2.14 | 0.99 |
| | 2 | 100 | 150 μL | 15 | 1.13 | 0.081 | 1.81 | 0.91 |
| isoprene | 3 | 60 | 90 μL | 8 | 1.26 | 0.138 | 1.40 | 0.85 |
| | 4 | 80 | 120 μL | 10 | 1.22 | 0.129 | 1.45 | 0.81 |
| | 5 | 160 | 60 μL | 22 | 1.24 | 0.160 | 1.24 | 0.96 |
| | 6 | 200 | 60 μL | 39 | 1.28 | 0.159 | 1.42 | 0.96 |
| | 1 | 20 | 80 ppbv | 40 | 1.20 | 0.125 | 1.22 | 0.97 |
| α-pinene | 2 | 60 | 75 μL | 110 | 1.26 | 0.135 | 1.44 | 0.99 |
| | 3 | 10 | 80 ppbv | 10 | 1.29 | 0.146 | 1.31 | 0.95 |
| | 4 | 60 | 60 ppbv | 11 | 1.25 | 0.152 | 1.17 | 0.88 |
| | 1 | 30 | 75 μL | 18 | 1.21 | 0.146 | 0.92 | 0.95 |
| toluene | 2 | 60 | 150 μL | 40 | 1.35 | 0.190 | 0.79 | 0.94 |
| | 3 | 30 | 50 μL | 16 | 1.10 | 0.079 | 1.52 | 0.81 |
| | 4 | 60 | 75 μL | 11 | 1.26 | 0.129 | 1.42 | 0.99 |

[1] The amount of OH is expressed as the amount of methyl nitrite (μL)
[2] The concentration of PM (particulate matter) is the average mass concentration of SMPS in relative stable state which have already been amended by the measured
density.
[3] ρ is the average value of density after density reaches a relative steady state.
[4] f44 is the average fraction of signal of m/z 44 among the total organic signal after f44 reaches a relatively stable value.
[5] ACSM/SMPS is the slope of the fitting curve.





**Table 2. Summary of Smog Chamber Experiments for CE calibration of (NH₄)₂SO₄ when coating with SOA.**

| Precursor | # | VOC (ppbv) | O$_3$/OH [1] (ppbv/µL) | (NH$_4$)$_2$SO$_4$ [2] (µg m$^{-3}$) | (NH$_4$)$_2$SO$_4$ [3] (µg m$^{-3}$) | Org (µg m$^{-3}$) [4] | f44 | CE$_{SOA}$ [4] |
|---|---|---|---|---|---|---|---|---|
| α-pinene | 1 | 20 | 50 µL | 57.5 | 66.0 | 33.5 | 0.202 | 0.333 |
| | 2 | 20 | 80 ppbv | 50.3 | 56.19 | 20.1 | 0.193 | 0.324 |
| | 3 | 60 | 80 ppbv | 43.9 | 48.37 | 55.8 | 0.147 | 0.327 |
| | 4 | 60 | 50 µL | 40.3 | 134.24 | 303 | 0.127 | 0.966 |
| toluene | 1 | 80 | 150 µL | 51.2 | 81.35 | 79.7 | 0.209 | 0.461 |
| | 2 | 80 | 100 µL | 55.6 | 71.0 | 25.6 | 0.237 | 0.370 |
| | 3 | 100 | 50 µL | 59.0 | 70.1 | 15.2 | 0.262 | 0.345 |
| | 4 | 60 | 200 µL | 48.3 | 57.5 | 30.7 | 0.210 | 0.345 |
| | 5 | 80 | 200 µL | 53.2 | 84.8 | 106.4 | 0.190 | 0.462 |

[1] The amount of OH is expressed in terms of the amount of methyl nitrite injected (µL).
[2] The average mass concentration of (NH₄)₂SO₄ seed aerosol measured by ACSM before SOA was generated.
[3] The average mass concentration of SOA coated (NH₄)₂SO₄ seed aerosol measured by ACSM after coated.
[4] The generated SOA mass concentration after it reaches relatively constant value.
[5] CE of SOA coated (NH₄)₂SO₄. CE for dry, pure (NH₄)₂SO₄ was 0.28 in every experiment.





**Figure Captions:**

Figure 1. Schematic of the collapsible atmospheric-pressure fluoropolymer (Teflon) smog chamber and the instrument setup.

Figure 2. (a) A typical banana-shaped plot of aerosols generated from ozonolysis of α-pinene; (b) Time series of aerosol mass concentration measured by Q-ACSM and integrated from SMPS measurements, and the corresponding f44 and effective density of SOA calculated from the Q-ACSM and APM measurements, respectively; (c) The number size-distributions of aerosols at one hour interval during the experiment.

Figure 3. The response factor determined from the chamber experiment by fitting the Q-ACSM measured mass concentration to that deduced from SMPS-APM measurements. Only data points in relative steady-state was used.

Figure 4. Triangle plots of (left) α-pinene, (middle) toluene, (right) isoprene experiments. The color codes represent data points in a certain experiment. f44 and f43 represent more oxidized and more reduced form of organic components, respectively.

Figure 5. The linear correlations between Q-ACSM RFs (a) and effective density (b) with respective to f44 obtained from all experiments. The error bars were evaluated from different data points in the stable state within one experiments.

Figure 6. Organic, sulfate mass concentrations and f44 measured by Q-ACSM. Arrow 1 indicates when $(NH_4)_2SO_4$ seed particles were introduced; Arrow 2 indicates when VOC precursor were injected.

Figure 7. CE of SOA coated $(NH_4)_2SO_4$ particles. The color scale represents the Org/$SO_4$ mass ratio.

Figure 8. Overall view of SOA particle phase, effective density (ρ), response factor(RF), relative ionization efficiency(RIE) and collection efficiency(CE) variation with increasing O/C ratio in the process of SOA oxidation.





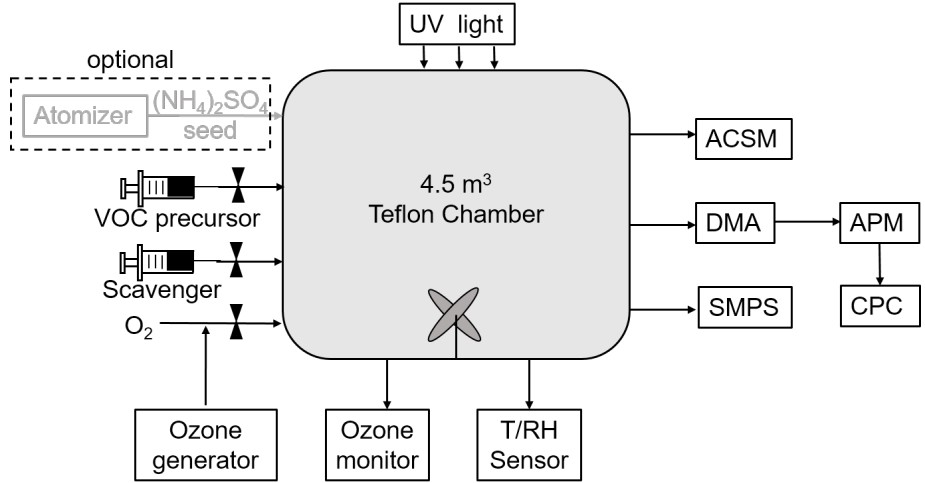


Figure 1. Schematic of the collapsible atmospheric-pressure fluoropolymer (Teflon) smog chamber
and the instrument setup.



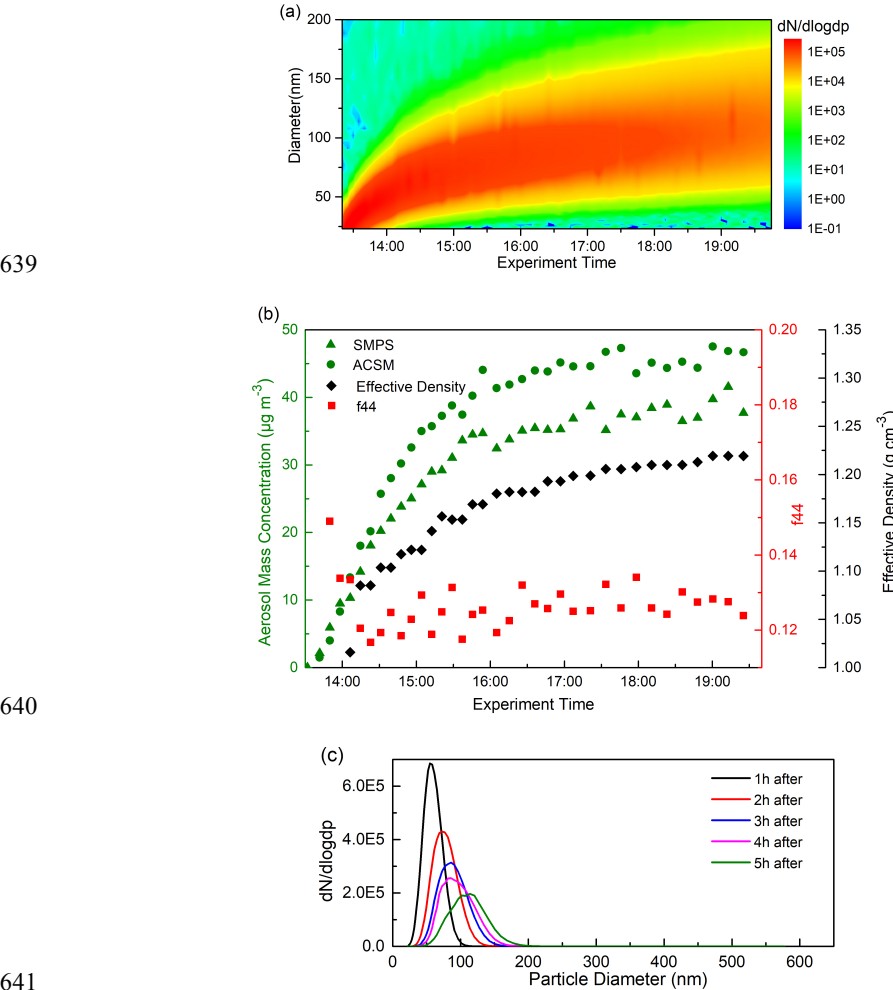




Figure 2. (a) A typical banana-shaped plot of aerosols generated from ozonolysis of α-pinene; (b)
Time series of aerosol mass concentration measured by Q-ACSM and integrated from SMPS
measurements, and the corresponding f44 and effective density of SOA calculated from the Q-
ACSM and APM measurements, respectively; (c) The number size-distributions of aerosols at one
hour interval during the experiment.





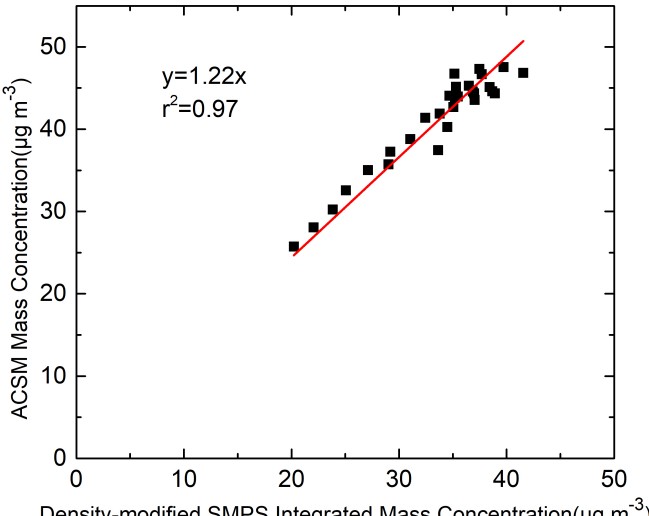


Figure 3. The response factor determined from the chamber experiment by fitting the Q-ACSM
measured mass concentration to that deduced from SMPS-APM measurements. Only data points
in relative steady-state was used.





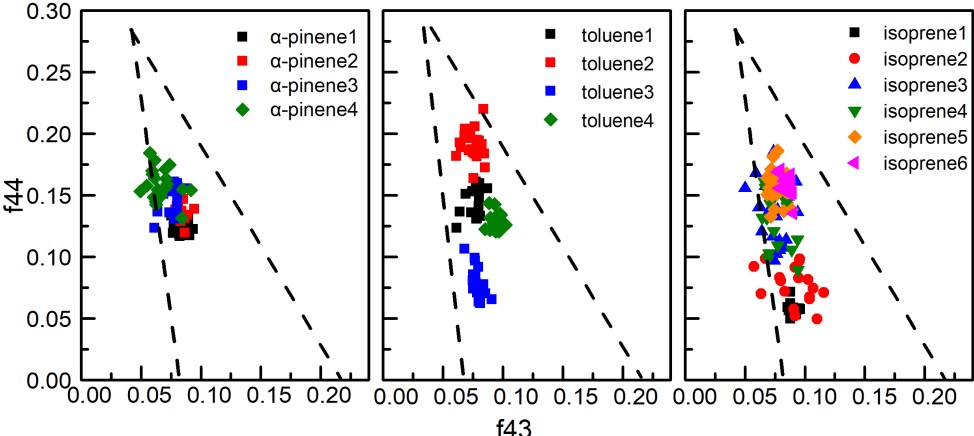


Figure 4. Triangle plots of (left) α-pinene, (middle) toluene, (right) isoprene experiments. The color

codes represent data points in a certain experiment. f44 and f43 represent more oxidized and more

reduced form of organic components, respectively.





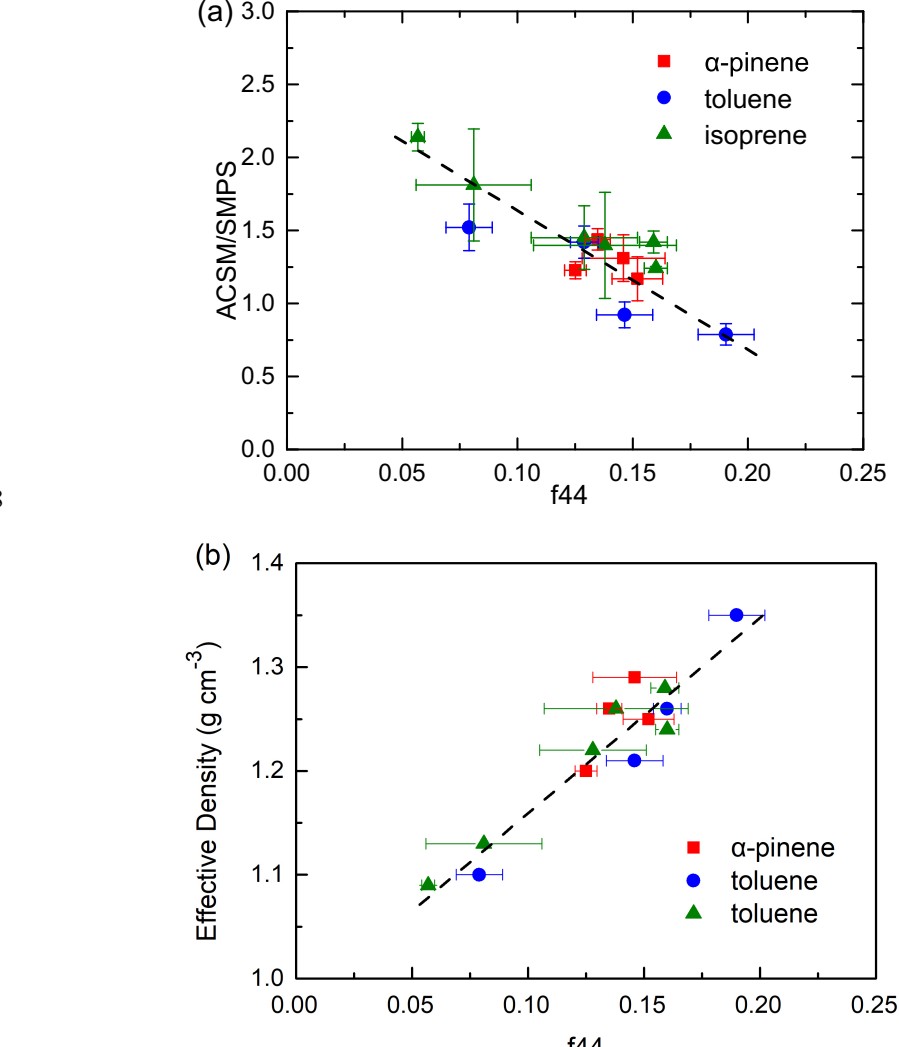


Figure 5. The linear correlations between Q-ACSM RFs (a) and effective density (b) with

respective to f44 obtained from all experiments. The error bars were evaluated from different data

points in the stable state within one experiments.



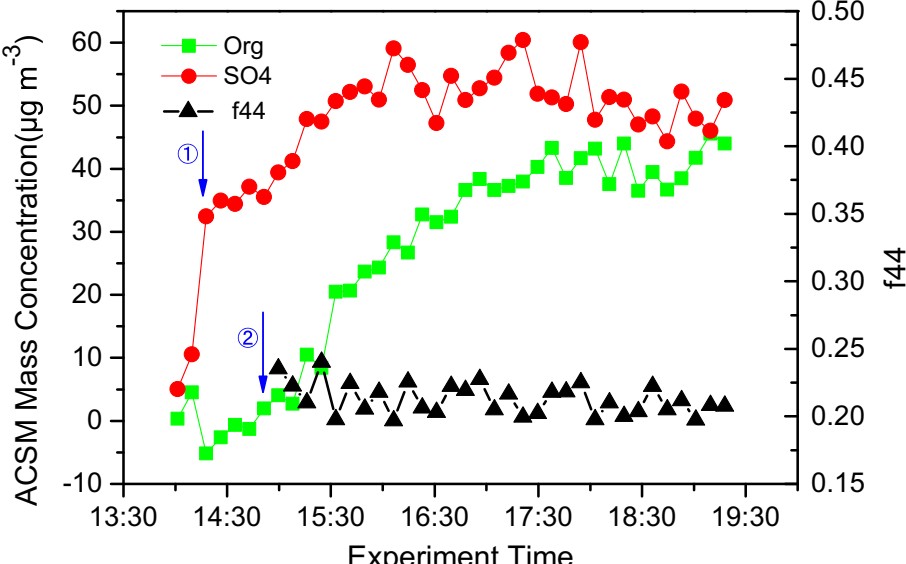


Figure 6. Organic, sulfate mass concentrations and f44 measured by Q-ACSM. Arrow 1 indicates

when $(NH_4)_2SO_4$ seed particles were introduced; Arrow 2 indicates when VOC precursor were

injected.






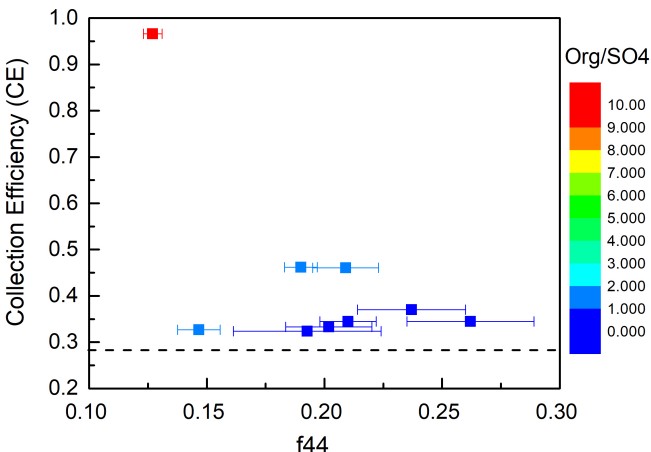


Figure 7. CE of SOA coated $(NH_4)_2SO_4$ particles. The color scale represents the Org/SO$_4$ mass
ratio.





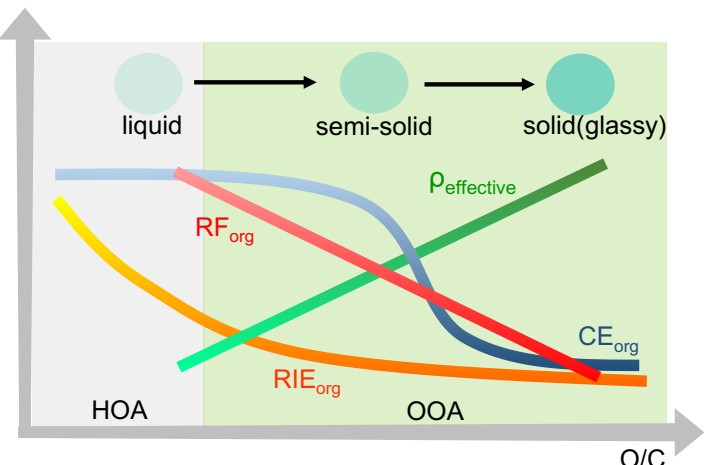


Figure 8. Overall view of SOA particle phase, effective density (ρ), response factor(RF), relative

ionization efficiency(RIE) and collection efficiency(CE) variation with increasing O/C ratio in the

process of SOA oxidation.