# Peer review of "Sensitivity of a Q-ACSM to chamber generated SOA with different oxidation states"

_Atmospheric Measurement Techniques, 2018_

## Referee Comment (RC1) · Anonymous Referee #1 · 23 Apr 2018

The manuscript by Li et al. investigated the sensitivity of a Q-ACSM for the measurements of chamber secondary organic aerosol (SOA) from three volatile organic compounds. The authors found that the sensitivity of the Q-ACSM to SOA was found to be anti-correlated with the aerosol oxidation state regardless of the VOC precursors. This study also shows the applicability of Q-ACSM in chamber studies although rare studies use Q-ACSM for chamber studies because of low sensitivity and time resolution. This manuscript is generally well written and fits within the scope of AMT. However, some conclusions are overstated and need more evidence.

Major comments:

1. The authors concluded CE as a function of particle phase states, which needs to be reconsidered. First, the RH was maintained at 10 – 15% during the experiments, we could not expect the changes in phase states of SOA particles. Second, Docherty et al. (2013) also showed the variations of CE as a function of f44/f57, which is primarily caused by particle bounce at the vaporizer.

2. It is difficult for this manuscript to conclude that RIE changes as SOA evolves from HOA to more oxygenated OOA. This is an SOA experiment, which cannot give you any information on HOA. Many factors can affect the anti-correlation between RF and f44, but RIE would not be the important one. As shown in the latest research by Xu et al. (2018), organic aerosol with OS below -1.5 can have much high RIE, but in this study, the f44 does not support this.

3. I would also suggest the authors comparing the result in this study with those previously reported by HR-ToF-AMS. For example, f44 vs. f43 could be different. Q-ACSM might reports much higher f44 than that of HR-ToF-AMS (Fröhlich et al., 2015).

References:

Docherty, K. S., Jaoui, M., Corse, E., Jimenez, J. L., Offenberg, J. H., Lewandowski, M., and Kleindienst, T. E.: Collection Efficiency of the Aerosol Mass Spectrometer for Chamber-Generated Secondary Organic Aerosols, Aerosol Sci. Tech., 47, 294-309, 10.1080/02786826.2012.752572, 2013.

Fröhlich, R., Crenn, V., Setyan, A., Belis, C. A., Canonaco, F., Favez, O., Riffault, V., Slowik, J. G., Aas, W., Aijälä, M., Alastuey, A., Artiñano, B., Bonnaire, N., Bozzetti, C., Bressi, M., Carbone, C., Coz, E., Croteau, P. L., Cubison, M. J., Esser-Gietl, J. K., Green, D. C., Gros, V., Heikkinen, L., Herrmann, H., Jayne, J. T., Lunder, C. R., Minguillón, M. C., Močnik, G., O'Dowd, C. D., Ovadnevaite, J., Petralia, E., Poulain, L., Priestman, M., Ripoll, A., Sarda-Estève, R., Wiedensohler, A., Baltensperger, U., Sciare, J., and Prévôt, A. S. H.: ACTRIS ACSM intercomparison – Part 2: Intercomparison of ME-2 organic source apportionment results from 15 individual, co-located aerosol mass spectrometers, Atmos. Meas. Tech., 8, 2555-2576, 10.5194/amt-8-2555-2015, 2015.

Xu, W., Lambe, A., Silva, P., Hu, W., Onasch, T., Williams, L., Croteau, P., Zhang, X., Renbaum-Wolff, L., Fortner, E., Jimenez, J. L., Jayne, J., Worsnop, D., and Canagaratna, M.: Laboratory evaluation of species-dependent relative ionization efficiencies in the Aerodyne Aerosol Mass Spectrometer, Aerosol Sci. Tech., 0-0, 10.1080/02786826.2018.1439570, 2018.

---

## Referee Comment (RC2) · Anonymous Referee #2 · 23 Apr 2018

General Comments:

This paper reports results from very difficult experiments aimed at understanding the sensitivity of the ACSM (Aerosol Chemical Speciation Monitor) instruments to organic material from aerosol particles. The topic of varying relative ionization efficiency for organic aerosol mass with the Aerosol Mass Spectrometer (AMS) has been debated in recent literature (Murphy, 2016a;Jimenez et al., 2016;Murphy, 2016b). Because the AMS has similar characteristics to the ACSM (aerodynamic aerosol focusing lens followed by thermal vaporization, electron impact ionization, and MS detection), most of the results described in this paper are potentially applicable to the AMS.

The findings of this paper are fascinating and the reported changes in SOA "sensitivity" are qualitatively consistent with many previous observations. A major strength of this

work is that surrogates for ambient secondary organic aerosols (SOA) were produced and important details on the chamber experiments are not needed to demonstrate the overall method for accurately measuring the SOA response factors for the ACSM. Unfortunately, this work does not provide any new, useful and quantitative information for the ACSM/AMS community and it is possible that the data from this work are unable to constrain the largest uncertainties for a broader application of these results. To make a significant contribution in the field, the work presented here should show all of the relevant calibration details and significantly reduce the largest uncertainties in the reported measurements. At a minimum, the manuscript must show that the methods work for aerosols with known chemical composition (ammonium nitrate and ammonium sulfate) and the uncertainties must be fully propagated. The paper has several major flaws that need to be addressed in revising this manuscript to make it acceptable for publication in AMT.

Specific Comments:

Writing and Presentation of Key Concepts

This paper was very difficult to read due to numerous grammatical errors and informal usage of important terminology (e.g. response factor or RF, relative ionization efficiency or RIE, collection efficiency or CE, and oxidation state). Broad claims of "accuracy," interpretation of the results, and the relevance of this work are not supported by the actual evidence presented. The manuscript would significantly benefit from more careful technical writing and editing.

The primary subject of this study, the response factor (RF) for the ASCM, should be clearly defined at the beginning of the manuscript with Equation 1 moved up to the end of the paragraph on the AMS/ACSM quantification (page 5 line 103). This equation shows that both RIE and CE are important factors in determining the RF, therefore any variation in RF could be due to either one or both of these factors changing. While this was described later in the manuscript (Sections 3.3 Effects of CE and 3.4 Effects

of RIE), these concepts are key to understanding and interpreting the reported RFs from these studies. The abstract should state that the RF values reported here were obtained using the default RIE for organic species (1.4) along with a CE of 0.5.

Accurately quantifying RIE and CE factors is crucial in determining the AMS/ACSM sensitivity to aerosol components and they need to be studied independently in order to generally apply the findings of this work. The work made an attempt at separating these factors, but did not adequately examine them for the analysis. The current debate in the literature is about RIE for organic aerosol (Murphy, 2016a, b;Jimenez et al., 2016) and the present work appears to contradict itself in the findings on RIE. This is probably due to careless usage of the terms "RIE" and "RF," as well as not incorporating the CE findings into calculations for the RIEs from the measured RFs. A thorough examination of the writing of the paper would presumably clarify the inconsistencies and demonstrate if the RIE varied significantly or not for these measurements.

The last sentence in the abstract states "Our results indicated that the current Q-ACSM calibration procedure using a constant RIE may lead to somewhat underestimation of more oxidized OOA but overestimation of less oxidized HOA, i.e., a variable RIE shall be applied, most likely as a function of the SOA oxidation state." In addition to the obvious mistakes in the grammar and use of terminology, this statement implies that previous measurements from the ACSM/AMS are incorrectly reporting OA mass concentrations. This statement needs to be supported with clear evidence of such problems. The results here, while not directly showing it, imply that the variations in the response factor can be largely explained by variations in CE rather than variations in RIE. Many studies have been using a higher CE for HOA that accounts for discrepancies between independent measurements and the AMS, for example the 2002 Pittsburgh study (Zhang et al., 2005) and subsequent work. How do the authors reconcile their results with those showing numerous, good correlations of ambient ACSM/AMS data with independent organic carbon aerosol or total mass measurements?

Accuracy and Uncertainties for the Number Distributions and Effective Density Measurements

In order to calculate the SOA mass concentrations being generated in the chamber, the measured number distributions as a function of particle size need to be converted into mass distributions using the particle density. Determining the overall uncertainty in the mass concentrations from the number distribution and the particle density is important. Here, the number distributions were measured with a scanning mobility particle analyzer (SMPS). Information about calibrating this instrument was not given in the paper, so it is unclear what the accuracy is for these measurements. Also, how well did the two number distributions match between the SMPS and the differential mobility analyzer (DMA) plus condensation particle counter (CPC) systems? A simple statement about using polystyrene latex spheres to calibrate both systems would provide additional confidence in these measurements.

Knowing the accuracy and uncertainty in the particle effective density measurements is also necessary. The particle effective density was determined by selecting the mobility diameter at the peak in the SMPS number distribution with the DMA and the peak in the mass per particle was scanned with an aerosol particle mass (APM) analyzer with a CPC. The literature reports using a similar DMA/APM/CPC system can provide accurate effective density measurements with 95% confidence intervals of 10-30%, depending on the configuration of the system (Johnson et al., 2013). The accuracy of the system used for this study was not presented as additional calibration information. For example, how accurate and precise are the effective densities for ammonium nitrate and ammonium sulfate (or other known particles)? The demonstrated uncertainties in the effective density measurements need to be propagated to the rest of the results.

The peak in the number distribution is not the same as the peak in the mass distribution. Were additional sizes of mobility diameter scanned (not just the peak in the number distribution) to ensure that the effective density was constant across all sizes in the distribution? If not, it should be clearly stated that it was assumed that the measured effective density was the same for the entire distribution.

Uncertainties in Collection Efficiency

One of the largest issues (uncertainties) in ACSM/AMS measurements is the collection efficiency (CE). In particular, the CE factor includes lens transmission and focusing in addition to particle bounce (Canagaratna et al., 2007;Huffman et al., 2005). Lens transmission (Liu et al., 1995a, 1995b) can often significantly affect the overall (i.e., observed) CE and it was neglected in this work. Furthermore, lens transmission can vary for nominally identical lenses (Bahreini et al., 2008) and the actual lens transmission for the instrument used in this study needs to be quantified. To fairly compare an external measurement of mass with the signals from an ACSM (or AMS), a correction to the volume distribution needs to be applied that accounts for particle losses in the lens. If the actual lens transmission cannot be determined, the theoretical lens transmission must be used to determine if the ACSM lens is transmitting the majority of the mass measured by the size distribution instrumentation.

For the data presented in Figure 2, the overlap between the measured number distribution when the system was stable after 5 hours and the lens transmission might possibly be close to 100%. Thus, the effect of lens transmission losses could be minimal for that particular experiment. Figure 3 shows that the measured signals from the ACSM using the default RIE of 1.4 and CE=0.5 agree very well with the SMPS mass measurements using an effective density of 1.2; the bias is +22% which is probably within combined experimental uncertainties. For the coatings experiments, it is possible that particles grew too large to be efficiently transmitted through the lens or that the increase in sulfate mass concentration was due to smaller particles becoming more efficiently transmitted. A reader cannot estimate lens transmission losses for these and the other experiments because the mass distributions are not shown in the manuscript (or supplemental information). Therefore, the lens transmission needs to be examined for all of the experiments to ensure that the mass sampled by the AMS is the same as the mass measured with the SPMS system.

The CE for dry, ammonium sulfate particles was used as a basis for the CE results

shown in Table 2 and Figures 6 and 7. Therefore, uncertainties in the CE for dry ammonium sulfate would affect the derived CE for the organic coatings. How was CE = 0.28 determined for dry ammonium sulfate? Prior work indicates that the CE for this species is 0.24+/-3% (Matthew et al., 2008). How stable is the ammonium sulfate seed source? The manuscript states that a "similar amount and size" of seed particles were used in the experiments, and the variability is a little more than 10% by averaging the mass of uncoated seeds in Table 2. Figure 6 shows that the sulfate mass concentrations are at least this variable (and possibly increasing) prior to (and shortly after) injection of the VOC precursors. Toward the end of the experiment, the sulfate mass concentrations appear to be decreasing while the organic mass loading increased more slowly than the sulfate at the middle of the experiment and seemed more stable than the sulfate at the end. (The data in Figure 6 do not appear to correspond with any of the experiments shown in Table 2.) Loss rates in the chamber would affect the interpretation of the changes in mass concentrations over the time of the coatings experiments. All of these factors contribute to uncertainties in the CE derived from these experiments, which should be added to the manuscript.

Volatility Effects

One potential issue that should be addressed is the possibility of the SOA being partially volatile. Particles with low f44 (such as hydrocarbon-like organic aerosol or HOA) are known to be the most volatile SOA species (Paciga et al., 2016;Huffman et al., 2009). Depending on how they are operated, the SMPS and DMA/APM/CPC systems could cause evaporative losses. There should be a brief discussion on how this might affect some of the results. How might variations in volatility affect the background (filter) subtraction in these ACSM measurements? Did the background levels change as a function of f44?

Technical Corrections

Page 1 Lines 1-2 (Title): It is important to distinguish the term "sensitivity" from the "response factor," here and throughout the manuscript, where the "response factor" was measured and the "sensitivity" was inferred. In addition, "different oxidation states" were not measured and should be replaced in the title with "varying average oxygen content" or similar phrasing. "Oxidation state" should be replaced throughout the manuscript with more appropriate phases, such as "f44" or "average oxygen content" depending on the context.

Page 1 Lines 20-21 (Abstract): This statement is not true as written. Should delete or replace the word "comprehensively."

Page 1 Line 21 (Abstract): revise "SOA samples were generated . . ." to "SOA was generated . . ."

Page 1 Line 23 (Abstract): The "dozens of ppbv" concentrations of the precursor VOCs used in this study are not typical. Suggest deleting/revising the phrase "atmospheric relevant concentrations" at this location as well as other places in the manuscript.

Page 2 Lines 24- 25 (Abstract): The sentence starting with "Different SOA oxidation states were achieved by . . ." should be re-written to something along the lines of "For some experiments, varying degrees of average oxygen content in the SOA were obtained by ... "

Page 2 Lines 26-28 (Abstract): Needs to be revised – circular argument about the "exact mass" "used to deduce" the "effective density" and "mass concentration."

Page 2 Line 29 (Abstract): The sentence starting here needs to be revised. Several parts of it are not clear and not valid. It indicates that "considerable errors" in the SMPS calibration occurred from variations in the effective density, but the 23% difference between the reported effective densities do not correspond to the factor of 3 difference in the measured response factors (from about 0.75 to about 2.15), which appeared to be corrected for varying effective density.

Page 2 Line 31 (Abstract): The term "can change substantially" should be replaced with

the actual values with uncertainties. The measurements indicate that effective density changes from 1.1 to 1.35, but the accuracy of the method was not demonstrated to interpret the significance of the change.

Page 2 Lines 31-33 (Abstract): Should replace "sensitivity" here and elsewhere with "response factor." This statement and the title of the manuscript are similarly imprecise, because the paper reports changes in the RF as a function of f44, which is a measure of the average oxygen content in the total organic aerosol mass and not the "oxidation state." A problem with this sentence and following statements is that the values for the response factors and the RIE/CE combination used to obtain them were not stated specifically in the abstract.

Page 2 Line 33-34 (Abstract): This statement should be revised because it implies that RIE and/or CE should be decreased relative to the values used (1.4 and 0.5, respectively) as the oxygen content increases.

Page 2 Lines 34-37 (Abstract): Sentence has grammatical and technical errors and needs to be re-written. There is no context for changes in sulfate signals.

Page 2 Lines 37-39 (Abstract): More grammatical and technical errors in this sentence. There is no evidence in this paper of transforming SOA "gradually" from a liquid state to a solid (or glassy) state. This work does show that the CE values vary depending on the average oxygen content as indicated by f44, and the statement should be revised accordingly. However, the CE values for the solid (or glassy) state are reported here as "0.2∼0.5," whereas the measured values in the paper show that CE is above 0.3. Knowing the CE values precisely is an important part of reducing the uncertainties in the response factors, since CEs of 0.2, 0.3, and 0.5 represent changes in RF by factors of 5, 3, and 2, respectively.

Page 2 Lines 39-41 (Abstract): Again, there is no evidence in these observations of SOA transformations occurring as stated (here from hydrocarbon-like OA to more oxygenated OA). Furthermore, the changes in CE as a function of f44 are consistent with

the changes in the response factor without a varying RIE. Specifically, if CE=1 instead of 0.5 for the lowest f44 values, the response factor would decrease from 2.15 to 1.1 and if CE=0.3 instead of 0.5 for the highest f44, the response factor would increase from 0.75 to 1.25. Both revised values for the response factors (1.1 and 1.25) are well within the uncertainties of ACSM measurements without changing the RIE significantly from 1.4.

Page 2 Lines 41-44 (Abstract): More grammatical and technical errors in this sentence. It appears that the term "RIE" was used here where the term "response factor" would be more accurate. The claims of "underestimation" and "overestimation" need to be quantified to be meaningful. The variations in the response factors appear to be consistent with the previously reported changes in CE as a function of average oxygen content for chamber-generated SOA (Docherty et al., 2013) and the 38% uncertainty in AMS organic mass concentration (Bahreini et al., 2009).

Other Corrections

There are other numerous grammatical errors in the rest of the manuscript, not specified here. Instead, noticeable technical errors are highlighted below:

Page 3 Line 57: add the word "air" or "aerosols"

Page 4 Line 92: The citation for Zhou et al., 2016 does not seem appropriate for the "widely used . . . around the world" context.

Page 5 Lines 119-121: There are many examples of ambient datasets showing the AMS mass concentrations agree well with other observations (Jimenez et al., 2016). This should be mentioned here.

Page 6 Line 130: This is a good place to cite the recent ACSM characterization paper (Xu et al., 2017).

Page 7 Line 169: mention the inner diameter instead of the "OD"

[Figure]

Page 9 Lines 207-208: f44 could also be artificially low when particles initially form if the ion signals from large m/z are not properly included in the total organic mass (and dividing the high S/N peak of m/z 44 with an artificially lower total organic signal). This issue could be checked for the first three data points in Figure 2 by comparing relatively high m/z ion peak intensities in their spectra with spectra from higher aerosol mass concentrations.

Page 12 Lines 284-285: Does "0.28 in this work" refer to the Matthew et al. study (Matthew et al., 2008) or the work for the manuscript? The Matthew et al. paper states a lower CE with uncertainties assigned to it (24+/-3%), making the 0.28 value outside the uncertainty band.

Page 12 Lines 286-290: The Matthew et al. work showed that the number of sulfate ions per particle did not change with additional water or organic coating, so it should be cited here that the RIE for sulfate does not change during the coating experiment. This point is important because any changes in the sulfate mass loadings during the course of the experiment are not because the RIE for sulfate changed.

Page 12 Line 292 and following: the "coating thickness" should be calculated using the size distribution data and included in Table 2. The size distribution data would also indicate other changes, like newly formed SOA particles for the "extreme condition" experiment.

Page 13 Lines 302-305: Both sentences appear to say that the light blue points are AS cores with an organic shell, but the connecting word is "however". Need to revise to clarify.

Page 13 Lines 314-315: "SOA may transit from liquid phase ..." Need to revise/check grammar.

Page 13 Lines 315-316: Higher oxidation state and higher effective density are not necessarily linked to changing phase states if the composition is also changing. Need

to revise.

Page 14 Line 323: To complete the section on CE, studies on CE of HOA and aged ambient OA could be added/mentioned, such as (Slowik et al., 2004;Zhang et al., 2005;Middlebrook et al., 2012).

Page 14 Lines 336-337: While this statement may be true, the section on CE as a function of f44 seems to indicate that the large changes in RF are mostly due to changes in CE. Once the CE is properly taken into account for the RF, RIE for the organics studied here does not seem to be varying by as much as the studies cited in the previous sentence. Consider adding more statements at the end of this section.

Table 1: Add errors, mainly to PM, density, f44, and ACSM/SMPS columns.

Table 2: Digits listed for ammonium sulfate, Org, f44, and CE are not significant. Revise and add errors to these columns. Add one more column (with errors) for the calculated thickness of organic coating, based on size distribution data and derived effective density.

Figure 1: The air in the chamber (how generated and introduced) is not shown. Is the chamber filled like a balloon or is it a flowing reactor? How is air pumped out of the chamber? Is air added to the chamber while the instruments are sampling? It is unclear how the ammonium sulfate seeds are dried and added to the chamber. The figure caption needs a few more details explaining a typical experiment.

Figure 2: This is the only figure with number distributions. They should be converted into mass distributions, the effect of the ACSM lens transmission should be shown (calculated transmission into the ACSM), and additional mass distributions should be added to supplementary information. Should add the "start time" for the experiment, along with an arrow indicating when the system was considered to be in "relative steady-state." Add that the AMS mass concentrations were calculated using the default RIE=1.4 and CE=0.5. Also include in supplementary information how the system

was calibrated for the ionization efficiency of ammonium nitrate and how accurately the ammonium nitrate effective density was measured using this method.

Figure 3: Should state in the caption what value of effective density was applied to the SMPS mass concentrations. Also add the default RIE=1.4 and CE=0.5 was used for the ACSM mass concentrations. Note that a bias of 22% (slope here) is within the +/-38% uncertainties of the AMS measurements (Bahreini et al., 2009).

Figure 4: This could be moved to the supplementary information, since it is not directly relevant to the discussion.

Figure 5: The conclusions of this paper about the response factor (and effective density) varying with f44 rely mainly on a few points affecting the trends (linear relationships). Errors need to be propagated to provide strength to this observation. There is a cluster of data points near f44=0.14, that have average effective densities of 1.25+/-0.05 and response factors of 1.25+/-0.3. These values are within the combined uncertainties. Toluene is listed twice in the effective density (bottom plot) legend – green triangles should be isoprene. Add to the caption the default RIE=1.4 and CE=0.5 were used for the calculations of the ACSM mass concentrations.

Figure 6: The experiment shown here is described in the text but does not appear to be listed in Table 2. As a second panel, the mass distributions should be shown. Why are the sulfate concentrations not "zero" before the seeds were added? What VOC precursor was added? When were the lights turned on? What are the various organic coating thicknesses estimated as the organic mass increases for a relatively constant sulfate mass? This experiment possibly indicates that the CE changed slightly when the SOA began to be detected, then did not change (within some uncertainty band) as the sulfate particles become coated with more organic material. Unfortunately, the sulfate mass concentrations do not appear to be stable enough to state this conclusively. Were there other experiments where the sulfate concentrations were more stable before adding the VOC and turning on the lights? See other comments on the discussion

of this figure on the lack of stability in the sulfate concentration during the course of the experiment and changes to the mass distributions potentially affecting lens transmission.

Figure 7: This plot is potentially the key to understanding the reason for changes in the response factors as a function of varying f44. However, it does not contain sufficient conclusive evidence on how f44 and coating thickness independently relate to CE. Are the changes in CE at f44 of about 0.2 as a function of increasing Org/SO4 ratios significant? Why are the range of values for f44 in these experiments forming SOA from alpha-pinene or toluene inconsistent with those shown in Figures 4 and 5? Which points are from alpha-pinene and which are from toluene? All of the mass distributions for these experiments (and the effect of ACSM lens transmission) should be shown in the supplementary information. Do the mass distributions indicate that the ammonium sulfate particles were "uniformly" coated, rather than new organic particles being formed? If so, the color scale should indicate coating fraction. Need error bars applied to CE.

Figure 8: This figure could be deleted because it does not show quantitatively the responses to organic aerosols for the ACSM measurements shown in this paper.

References

Bahreini, R., Dunlea, E. J., Matthew, B. M., Simons, C., Docherty, K. S., DeCarlo, P. F., Jimenez, J. L., Brock, C. A., and Middlebrook, A. M.: Design and operation of a pressure-controlled inlet for airborne sampling with an aerodynamic aerosol lens, Aerosol Sci. Technol., 42, 465-471, 10.1080/02786820802178514, 2008.

Bahreini, R., Ervens, B., Middlebrook, A. M., Warneke, C., de Gouw, J. A., DeCarlo, P. F., Jimenez, J. L., Brock, C. A., Neuman, J. A., Ryerson, T. B., Stark, H., Atlas, E., Brioude, J., Fried, A., Holloway, J. S., Peischl, J., Richter, D., Walega, J., Weibring, P., Wollny, A. G., and Fehsenfeld, F. C.: Organic aerosol formation in urban and industrial plumes near Houston and Dallas, Texas, Journal of Geophysical Research-

Atmospheres, 114, D00F16, 10.1029/2008JD011493, 2009.

Canagaratna, M. R., Jayne, J. T., Jimenez, J. L., Allan, J. D., Alfarra, M. R., Zhang, Q., Onasch, T. B., Drewnick, F., Coe, H., Middlebrook, A., Delia, A., Williams, L. R., Trimborn, A. M., Northway, M. J., DeCarlo, P. F., Kolb, C. E., Davidovits, P., and Worsnop, D. R.: Chemical and microphysical characterization of ambient aerosols with the Aerodyne aerosol mass spectrometer, Mass Spectrometry Reviews, 26, 185-222, 10.1002/mas.20115, 2007.

Docherty, K. S., Jaoui, M., Corse, E., Jimenez, J. L., Offenberg, J. H., Lewandowski, M., and Kleindienst, T. E.: Collection Efficiency of the Aerosol Mass Spectrometer for Chamber-Generated Secondary Organic Aerosols, Aerosol Science and Technology, 47, 294-309, 10.1080/02786826.2012.752572, 2013.

Huffman, J. A., Jayne, J. T., Drewnick, F., Aiken, A. C., Onasch, T., Worsnop, D. R., and Jimenez, J. L.: Design, modeling, optimization, and experimental tests of a particle beam width probe for the Aerodyne aerosol mass spectrometer, Aerosol Sci. Technol., 39, 1143-1163, 10.1080/02786820500423782, 2005.

Huffman, J. A., Docherty, K. S., Aiken, A. C., Cubison, M. J., Ulbrich, I. M., DeCarlo, P. F., Sueper, D., Jayne, J. T., Worsnop, D. R., Ziemann, P. J., and Jimenez, J. L.: Chemically-resolved aerosol volatility measurements from two megacity field studies, Atmos. Chem. Phys., 9, 7161-7182, 10.5194/acp-9-7161-2009, 2009.

Jimenez, J. L., Canagaratna, M. R., Drewnick, F., Allan, J. D., Alfarra, M. R., Middle-brook, A. M., Slowik, J. G., Zhang, Q., Coe, H., Jayne, J. T., and Worsnop, D. R.: Comment on "The effects of molecular weight and thermal decomposition on the sensitivity of a thermal desorption aerosol mass spectrometer", Aerosol Science and Technology, 50, i-xv, 10.1080/02786826.2016.1205728, 2016.

Johnson, T. J., Symonds, J. P. R., and Olfert, J. S.: Mass–Mobility Measurements Using a Centrifugal Particle Mass Analyzer and Differential Mobility Spectrometer, Aerosol

[Figure]

Science and Technology, 47, 1215-1225, 10.1080/02786826.2013.830692, 2013.

Liu, P., Ziemann, P. L., Kittelson, D. B., and McMurry, P. H.: Generating Particle Beams of Controlled Dimensions and Divergence: I. Theory of Particle Motion in Aerodynamic Lenses and Nozzle Expansion, Aerosol Science and Technology, 22, 293-313, 1995a.

Liu, P., Ziemann, P. L., Kittelson, D. B., and McMurry, P. H.: Generating Particle Beams of Controlled Dimensions and Divergence: II. Experimental Evaluation of Particle Motion in Aerodynamic Lenses and Nozzle Expansions, Aerosol Science and Technology, 22, 314-324, 1995b.

Matthew, B. M., Middlebrook, A. M., and Onasch, T. B.: Collection efficiencies in an Aerodyne aerosol mass spectrometer as a function of particle phase for laboratory generated aerosols, Aerosol Science and Technology, 42, 884-898, 10.1080/02786820802356797, 2008.

Middlebrook, A. M., Bahreini, R., Jimenez, J. L., and Canagaratna, M. R.: Evaluation of composition-dependent collection efficiencies for the Aerodyne aerosol mass spectrometer using field data, Aerosol Science and Technology, 46, 258-271, 10.1080/02786826.2011.620041, 2012.

Murphy, D. M.: The effects of molecular weight and thermal decomposition on the sensitivity of a thermal desorption aerosol mass spectrometer, Aerosol Science and Technology, 50, 118-125, 10.1080/02786826.2015.1136403, 2016a.

Murphy, D. M.: Reply to "Comment on the effects of molecular weight and thermal decomposition on the sensitivity of a thermal desorption aerosol mass spectrometer" by Jimenez et al, Aerosol Science and Technology, 50, 1277-1283, 10.1080/02786826.2016.1254347, 2016b.

Paciga, A., Karnezi, E., Kostenidou, E., Hildebrandt, L., Psichoudaki, M., Engelhart, G. J., Lee, B. H., Crippa, M., Prévôt, A. S. H., Baltensperger, U., and Pandis, S. N.: Volatility of organic aerosol and its components in the megacity of Paris, Atmos. Chem.

Phys., 16, 2013-2023, 10.5194/acp-16-2013-2016, 2016.

Slowik, J. G., Stainken, K., Davidovits, P., Williams, L. R., Jayne, J. T., Kolb, C. E., Worsnop, D. R., Rudich, Y., DeCarlo, P. F., and Jimenez, J. L.: Particle morphology and density characterization by combined mobility and aerodynamic diameter measurements. Part 2: Application to combustion-generated soot aerosols as a function of fuel equivalence ratio, Aerosol Science and Technology, 38, 1206-1222, 10.1080/027868290903916, 2004.

Xu, W., Croteau, P., Williams, L., Canagaratna, M., Onasch, T., Cross, E., Zhang, X., Robinson, W., Worsnop, D., and Jayne, J.: Laboratory characterization of an aerosol chemical speciation monitor with PM2.5 measurement capability, Aerosol Science and Technology, 51, 69-83, 10.1080/02786826.2016.1241859, 2017.

Zhang, Q., Canagaratna, M. R., Jayne, J. T., Worsnop, D. R., and Jimenez, J. L.: Time‐ and size‐resolved chemical composition of submicron particles in Pittsburgh: Implications for aerosol sources and processes, Journal of Geophysical Research: Atmospheres, 110, doi:10.1029/2004JD004649, 2005.

---

## Referee Comment (RC3) · Anonymous Referee #3 · 16 May 2018

In this work, Li et al. have investigated the response factor of a Q-ACSM to secondary organic aerosols generated under different oxidation states and from various VOC precursors. The subject of this work is certainly within the scope of AMT and the results of this work are of practical interests to the ACSM/AMS community. The advantage of this work is that the aerosol standards were generated in situ and their oxidation states can be pre-defined. The experiments were designed with some novelty and were carefully executed. The manuscript was clear written. However, the manuscript still can be improved. A native English speaker is highly recommended to proofread the manuscript before it can be published. Overall, I think this work can be a valuable contribution to the AMS/ACSM community. Therefore, I would recommend this manuscript for publication after the authors address the following comments:

[Figure]

Specific comments:

1) The authors shall pay more attention to the special terms and jargons used by the ACSM/AMS community. For example, the RF is not clearly defined here. The sensitivity of the ACSM/AMS has been investigated extensively in previous works. The results of this work seem to be in line with previous results. But the authors need to reorganize the manuscript to make the results of this work inter-comparable with other similar works, i.e., the results of this work can be evaluated under the same established framework.

2) For Fig. 8, how is this figure generated and how valid is this result? The author may want to provide more discussion and support information for validation.

―――――――――――――――――

---

## Referee Comment (RC4) · Anonymous Referee #4 · 29 Jun 2018

**Review of Li et al. "Sensitivity of a Q-ACSM to chamber generated SOA with different oxidation states"**

https://doi.org/10.5194/amt-2018-45

**Description:**

This manuscript describes measurements of the sensitivity of an ACSM to OA for three oxidant/VOC SOA systems produced in a mid-sized Teflon chamber using an ACSM, SMPS, DMA-APMA. Relationships of sensitivity and effective density to f44 are explored. Experiments were conducted in attempt to separate effects on sensitivity due to collection efficiency (CE) and relative ionization efficiency (RIE) of OA. Conclusions are made on trends of CE, RIE, phase state, HOA vs OOA, effective density, and oxidation state.

The manuscript is generally poorly written and makes a range of conclusions using flawed logic and speculation based on incomplete experimental evidence and lack of context from the existing literature. Due to some critical flaws (discussed below), I do not believe the manuscript is publishable in its present form. If it was published, it would "pollute" the literature with trends that are not well supported and may be due to experimental factors completely unrelated to those that are discussed.

With substantial additional work (including additional experimental evidence, analysis, and calibrations) and documentation, greatly scaled back conclusions, more thorough context, and, the dataset may eventually be publishable. Even in that case, the authors would need to present a convincing case of what parameters they are truly quantifying, including quantifying their uncertainties, and why the results are useful and better inform the analytical study of OA in the laboratory. In its current form, the manuscript is misleading and does not advance findings that are adequately constrained, new, or useful to the literature. Therefore, I strongly recommend that the paper is rejected, and the authors are encouraged to do the additional work and resubmit a much improved paper at a later time. See details below.

**Critical Flaws**

The paper really only quantifies an empirically-defined response factor, which is the ratio of the mass concentrations reported by the ACSM and calculated from the SMPS volume and the measured particle density. As thoroughly documented in the AMS literature, that response factor can have trends due to several effects:

$$RF = (RIE_{real} / RIE_{assumed}) * (E_{b,real} / E_{b,assumed}) * (E_{L,real} / E_{L,assumed}) * (E_{S,real} / E_{S,assumed}) * (SMPS_{calc} * SMPSvol_{real}) * (\rho_{real} / \rho_{assumed})$$

where CE = $E_b$ * $E_L$ * $E_S$ shows the 3 components of CE, due to particle bounce in the vaporizer, incomplete transmission by the aerodynamic lens, and non-spherical particle effects, respectively as defined by Huffman et al. (2005). It is also clear that errors in the estimation of particle volume from the SMPS number distribution and/or the density will also appear in RF.

When one is trying to characterize a widely used instrument and publishing the observed results as if they are representative of all instruments of the same type, great care needs to be taken to examine each of these terms and their uncertainties. The present paper has some speculative interpretations of RF based on RIE and $E_b$, even though it completely ignores all the other effects! My best guess is that $E_L$ explains a substantial fraction, and potentially most of the trend in RF, as when the particles grow in the SOA experiments, the upper end of the distribution may fall outside of the ACSM lens transmission, thus creating an experiment and time-dependent trend on RF. It is possible that some of the other factors that are ignored in this paper may also contribute to the observed trend. For this reason the paper is not publishable.

The authors need to go back to the lab and present multiple additional pieces of information, including:

1) A detailed characterization of $E_L$ vs. particle size for their own instrument. It is well-know that $E_L$ varies for different instruments (and sometimes in time for a given instrument), especially at the upper end of the particle size. However, Figure 2 suggests that losses at the small end may also be a problem. These are challenging experiments, but they are doable. If the authors cannot accomplish this, they should not resubmit this paper elsewhere, as the results would be ambiguous, and it would be very misleading to report them as trends in $E_b$ and $RIE_{org}$ while ignoring $E_L$. Using the transmission curves from the literature is not acceptable, given substantial variability in $E_L$ across instruments. See below for literature references.

2) Using the measured SMPS volume distributions, the measured density, and the measured ACSM transmission vs particle size (in $d_{va}$ space, see DeCarlo et al., 2004), the authors can "trim" the SMPS volume to correspond to the volume actually sampled by the ACSM.

3) It does not seem acceptable to present density measurements at only one size, especially if it is the peak of the number distribution. A few points across the distribution, and importantly the peak of the volume distribution, should be included for each system.

4) Evidence of good alignment of the aerodynamic lens in the ACSM should be presented. The particle beam width depends on particle size and shape, and if the lens is not well-aligned, then there can be particle losses that are strongly size-dependent, giving rise to another source of variation of $E_L$ that would show up in RF.

5) Evidence of calibration of the SMPS size (using PSLs) and especially, volume concentration, needs to be presented. It is typical in field studies when multiple SMPSs are present (all operated by reputable groups) that the reported volume concentrations range a factor of 4-5. That's the typical accuracy that can be expected for an SMPS that is running on a lab and that has not been thoroughly calibrated and quality-checked. Even after careful work, discrepancies often remain, see for example Figure 8 in Wiedensohler et al. (2012). There are a number of papers that specify the checks that one has to do in order to ensure quantitative SMPS measurements, and it was especially glaring that this topic was not even mentioned in the present paper. See for example Wiedensohler et al. (2018) and Stolzenburg and McMurry (2018). An intercomparison to other instruments such as additional SMPSs and OPCs (for example UHSAS) would be useful to gain confidence in the measurements. Evidence of quantitative response of the CPC, ideally by comparison to an electrometer, but otherwise by comparison to several other CPCs (across the size range) is also needed.

6) Evidence of accurate density measurements using known materials, and across the particle size range contributing to the measurements shown, needs to be presented.

7) The uncertainty in each term of the equation for RF, as well as in 1-6 above, needs to be quantified, and the uncertainty needs to be propagated. Only then we can know if any of the reported trends for (for example) $E_b$ have any meaning.

Also it should be noted that for internally mixed particles such as those generated here $E_b$ will be a property of the particle that is influenced by its sulfate and OA content. So it is non-sensical to write $E_{b,SO4}$ and $E_{b,OA}$ (as the authors do in their notation) as if those were different quantities in the seeded experiments, or as if they were a property of the species and not the mixture.

Moreover, it is not standard practice to use ACSM or AMS measurements for quantitative yield experiments for chamber experiments in the first place, due to the complexities outlined above. Rather, researchers typically have used SMPS volume together with density estimates (or comparisons with filters as well as other methods), and OA/SO4 ratios, to quantify yields. Jimenez et al. 2016 (which is referenced in this manuscript) states "Finally, we reiterate the need for direct calibration in laboratory experiments utilizing specific organic compounds or mixtures." So, this issue is not new and nor do the authors offer an advance in addressing the issue.

As discussed on that reference, the trends of AMS response observed for field and laboratory experiments are often different, due to the wider range of particle materials that can be made in the lab. Lab experiments typically have trouble replicating the OA composition observed in the atmosphere. Therefore, no implication should be made that the results from a lab study apply to field measurements. Any statements about the performance of the AMS / ACSM in field

measurements should be derived from the analysis of field measurements, of which there is an extensive record in the published literature. Comparisons of AMS with SMPS in field studies are not consistent with the trends shown here, see for example Jimenez et al. (2016) and references therein.

**Other Important Comments:**

As was extensively detailed by Reviewer 2, the manuscript is full of grammatical errors, confusing text, logical flaws and overstatements of conclusions. I agree with the comments of Rev. 2, and will thus focus on the most important additional points, and a few similar points with additional context or emphasis. However, the list is not comprehensive as it would be too time consuming to point out all the issues and any effort to revise and resubmit the paper should go beyond just the issues detailed by the reviewers. A detailed review of the Abstract is below, followed by other issues organized by topic.

Abstract:
L21: "comprehensively" is an extreme overstatement.

L25/L31/L32: "oxidation states" were not measured in this study.

L27: "exact mass" doesn't make sense.

L35: The cause of decreasing RF was not "pinpointed" or even significantly constrained.

L36: The CE was not determined quantitatively.

L37-39: "Our experiment results along with previous literature reports strongly implied that as the SOA oxidation state increases, SOA will transform gradually from a liquid state (CE » 1) into a solid (or glassy) state with a CE of 0.2~0.5." This is a gross overstatement of what the evidence presented in this manuscript supports. The trends observed may have been any combination of lens transmission effects, $E_b$ effects of mixed dry ammonium sulfate / OA particles, OA RIE effects, OA CE effects, or variable errors on the SMPS or CPC used for comparison. Neither CE effects nor phase state effects were isolated in this study.

L39-41: "Meanwhile, the RIE of OA decreased substantially when SOA transformed from hydrocarbon-like OA (HOA) into more oxygenated OA (OOA) and may further decrease as O/C continued to increase." Like with $E_b$, the effects of RIE of OA were NOT isolated in this study. Moreover, the results presented here have no bearing on HOA since only chamber SOA was studied. Also, no data on O/C was presented here (inferring O/C from f44 for chamber SOA is not justified – see details below)

L41-44: "Our results indicated that the current Q-ACSM calibration procedure using a constant RIE may lead to somewhat underestimation of more oxidized OOA but overestimation of less oxidized HOA, i.e., a variable RIE shall be applied, most likely as a function of the SOA oxidation state." As stated above (and further explained below), RIE effects were not isolated.

More Detail on Lens Transmission Effects:
The ACSM lens transmission declines to zero for both small and large particles. In particular the upper size cut depends on the specific lens being used (Hu et al., 2017), and can even change in time after an instrument is moved to a different location. The transmission also depends strongly on the pressure inside the AMS aerodynamic lens, which can decline dramatically if the pressure in the lens is reduced (e.g. Bahreini et al., 2003) , for example due to a partially clogged critical orifice, or to just being set at the wrong pressure by using an improperly sized critical orifice. This value should be documented here for all experiments, and care should be taken to ensure that all SOA experiments are done with the same lens pressure used in characterizing $E_l$ vs. size. The trends of the response factor shown here may be significantly influenced, or may even be dominated, by differences in the particle size ranges analyzed by the ACSM and SMPS. It is not acceptable to publish a paper that attempts to quantify CE and RIE and the ACSM without a detailed and careful quantitative calibration of the transmission of the ACSM vs size, and applying that transmission curve to all SMPS measurements. Using transmission curves from the literature is not acceptable, as there is considerable variation in those, and it is not known which literature curve (if any) may apply to the ACSM used here. I would expect to see several figures and pages of text devoted to calibrating the AMS size transmission before any conclusions about CE or RIE could be credible.

No volume size distributions are shown here for the different experiments, so it is impossible for the reader to even estimate the extent to which differences in particle size transmission could be a problem here.

As Reviewer 2 points out, the decreasing trend in RF vs f44 (Fig. 5a) could be simply due to an artifact of increasing particle size extending beyond the upper size limit of 100% aerodynamic lens transmission efficiency. While a well-performing lens can transmit particles with 100% efficiency up to ~550 nm vacuum aerodynamic diameter (~400-450 nm mobility diameter; Knote et al., 2011; Hu et al., 2017), lenses with reduced transmission at larger sizes is not uncommon (Liu et al., 2007; Takegawa et al., 2009; Hu et al., 2017; Campuzano-Jost et al. 2017), resulting in large losses for sizes as small as 450 nm vacuum aerodynamic diameter (~300-350 nm mobility diameter). Such effect may also play a role in the results presented in Fig. 6 and Fig. 7, as suggested by Rev. 2. Only then should the trends in RF can be considered reliable.

Collection Efficiency and Relative Ionization Efficiency Sections:
The sections on CE (sect. 3.3) and RIE (sect. 3.4) are extremely under-supported, misleading, and highly speculative. Given the experiments conducted and methods used, separation of the effects of CE and RIE does not seem possible. This applies to both relative trends and absolute values. Those sections should be completely eliminated unless unambiguous new evidence can be provided. The RIE section provides no experimental evidence and only an incomplete discussion of the published literature on this topic. The CE section discusses experiments where SOA was coated on top of dry ammonium sulfate seed and the "Response Factor (RF)" was calculated. However interpretation of the results are flawed. The RIE of sulfate was not calibrated, even though it may vary substantially (e.g., Zhang et al., 2017). Despite the lack of sulfate calibration, the authors assume an RIE of sulfate and calculate a CE of ammonium sulfate that depends proportionally on that value. While it is stated that the absolute value is not important but rather just trends matter, in the same paragraph the argument that when a lot of SOA is added to the ammonium sulfate seed, a CE of unity is observed which supports that the aerosol is now in a liquid state. Not only does that line of logic require that ammonium sulfate is calibrated, it also requires that the RIE of the OA is known. So, generally the main figure of that section (Fig. 7) is very problematic since the RIE of SO4 and OA are not known. Also, note that calculation of a CE for dry ammonium sulfate of 0.28 in this work does not narrow the range of uncertainty from lack of calibration for ammonium sulfate RIE since a range of a factor of 2 (0.2-0.4) has been observed for dry ammonium sulfate (Hu et al., 2017 and references therein). Moreover, equating the trends in the RF to changes in the RIE of the SOA is not necessarily justified since the effects may be the "results of a combination of a AS core and organic shell" as the authors state in Line 305. Thus, any trends in Fig. 7 may be dominated by the SOA coating thicknesses. However, changes in CE and RIE of the SOA may also be factors. Consequently, any quantitative conclusions that can be drawn about CE of OA is extremely under-constrained here. In order to separate RIE and CE effects, a direct, unambiguous measurement of CE needs to be made (e.g., see Xu et al., 2018).

Incomplete Literature Context / Major Omissions:
Several instances stand out as having a major lack of literature review/support or simply omitting fundamental references on topics discussed. For example the Kuwata et al. (AS&T 2012) paper which formulates the relationship between SOA density and O/C and H/C is never referenced or discussed (or any other paper on such relationships). A reader might get the impression that this manuscript was the first to show a correlation between SOA density and oxidation. Also, the paper Xu et al. (AS&T 2018), which extensively investigates RIE of OA for a large range of compounds and SOA (unambiguously separating CE and RIE), is completely missing. Another example is the section "Effects of f44" (3.2), where only a few peripheral references are provided in discussion of these results; whereas, there is a large body of work discussing trends in f's and elemental ratios for SOA chamber studies. Even the papers introducing the f44 vs f43

diagram (Ng et al., 2010; 2011) are not referenced! Without such context, it is impossible to determine if anything new was learned from the present studies.

f44, O/C, Oxidation State:
The use of f44 is used throughout the manuscript to be equivalent to O/C and "oxidation state," which is an unjustified step for chamber data. The authors state in Lines 200-203: "Since $m/z$ 44 signal basically reflected the oxygen content in OA, the O/C ratio can be deduced directly from f44, both of which have been widely used to represent the oxidation state of OA (Canagaratna et al., 2015)." While this may be true for ambient OA, this does not necessarily apply to chamber data. Canagaratna et al. (2015) (Fig. 8a) shows that relationship for ambient, standards, and laboratory produced SOA (a-pinene, toluene, isoprene, sesquiterpenes). Ambient OA and some standards follow a well-established relationship. However, taken together the chamber SOA shows no correlation, which is not surprising since the presence of acids (which produce CO2+ in the AMS/ACSM) and other oxidized functional groups would not necessarily be expected to track for specific oxidant/VOC systems. Only sesquiterpenes and a-pinene fall on the ambient parameterization line and are clustered within a narrow region.

Moreover, if the RIEs of OA indeed vary a lot for the systems studied then it would seem very possible that changes in f44 could be controlled as much by changes in the overall OA sensitivity (the denominator of f44 = $m/z$ 44 / OA) as by the actual relative abundance of $m/z$ 44 in the SOA. $CO_2^+$ in the AMS is expected to be largely formed from the decarboxylation of organic acid groups on the vaporizer, forming $CO_2$ gas, which would have a constant RIE, and thus would not track changes in overall OA RIE.

Therefore, the use of the terms O/C and oxidation state should be avoided beyond an explanation that it might be roughly indicative of oxygen content and possibly of acid content.

Clarity in Scope:
All conclusions and interpretation should be scaled back to clearly state to apply to only RF (CE x RIE), effective density, f44 (not O/C or oxidation state), three oxidant/VOC SOA systems, dry conditions, and laboratory SOA. No extrapolation to ambient OA should be given or implied, since no evidence is presented and current evidence in literature suggests that such an extrapolation is not warranted – see e.g., Xu et al., AS&T 2018, Jimenez et al., AS&T 2016.

Figure 8:
This figure should be removed: no axis are given, and the data for several of the trends have not been derived in this paper, so this is really a cartoon from the author's imagination, which has no place in a paper. If the authors do all the extra work to make this a serious study, then the figure should be included with quantitative axes and with error bars for both X and Y.

In addition, the trend in density is not new and was published at least half a decade ago (Kuwata et al., 2012). The trends in CE and RIE of OA are not demonstrated in this paper and are especially misleading. The trends in RF may be dominated by lens transmission effects, and may have a contribution from the instrument that is being compared with, and can only be supported with further evidence. HOA is not studied here at all.

Also note that a published study (Pajunoja et al., 2016) reported the opposite trend for bounce vs. O/C, in that study bounce *decreased* as O/C increased.

Other Miscellaneous Comments (substantial and minor):
- L75: Real time composition measurements is the key development that AMS made possible/routine (not sized-resolved).
- L93-103: In this paragraph where CE and other effects on AMS sensitivity are introduced, it would be useful to instead use the more precise terms of each of the components of CE: lens, shape-related, and bounce effects (CE = $E_L * E_s * E_b$; Huffman et al., AS&T 2005).
- L101-103: This statement about the high uncertainty in CE and IE is unclear whether it applies to ambient or laboratory data and should be clarified (since it is not true for ambient data, per the Jimenez et al. 2016 reference cited).
- L124-125. AMS light scattering does not employ laser-based vaporization.
- L126-130. This statement claiming that the increased fragmentation in the capture vaporizer will "highly complicate the AMS quantification process" does not reflect conclusions of the Hu et al., 2017 paper cited here, nor any of the several other papers investigating the capture vaporizer. For example Hu et al. (2018a) state that "tracer ion marker fractions, which are used to characterize the impact of different sources are still present and usable in the CV." and Hu et al. (2018b) state "Consistent time series of positive matrix factorization (PMF) factors and their fractions of total OA were found across the CV and SV in the three very different ambient data sets ranging from biogenic- to anthropogenic-dominated, indicating limited loss of source determination information despite the increased fragmentation." It appears that this statement was added to dismiss this method as a practical way to reduce CE uncertainties. In fact, in the recent paper by Xu et al., AS&T, 2018, it was shown that the capture vaporizer can be used as a method to isolate changes in OA RIE. Given the other limitations of the ACSM, the authors may want to consider performing a similar study using a CV, which would make $E_b$~1 and thus substantially simplify the interpretation of the trends on RF.
- L162: "self-synthesized" is confusing. Consider alternative phrasing such as "synthesized in house".
- L177: "*m/z* calibration", not "mass calibration"
- L195: add "a" before "strong"
- L235-238: "Especially, laboratory studies have shown that the ozonolysis of α-pinene can lead to considerably higher SOA yield than that from OH initiated reactions (Yao et al., 2014). Therefore, in this work we were focusing on the isoprene-OH and α-pinene-O3 reactions only".

This does not seem like a good reason to omit these data from the analysis. OH oxidation is an important loss for a-pinene in the atmosphere and makes substantial SOA. This data should be shown and discussed. Also, Yao et al. should not be the primary reference here.
- L243: grammar.
- L291: grammar
- L293: grammar

**References:**

Bahreini, R., J.L. Jimenez, J. Wang, R.C. Flagan, J.H. Seinfeld, J.T. Jayne, and D.R. Worsnop. Aircraft-based Aerosol Size and Composition Measurements during ACE-Asia using an Aerodyne Aerosol Mass Spectrometer, Journal of Geophysical Research – Atmospheres, Vol. 108, No. D23, 8645, doi:10.1029/2002JD003226, 2003.

DeCarlo, P., Slowik, J.G., Worsnop, D.R., Davidovits, P., and Jimenez, J.L. Particle Morphology and Density Characterization by Combined Mobility and Aerodynamic Diameter Measurements. Part 1: Theory. Aerosol Science and Technology, 38: 1185–1205, 2004. DOI: 10.1080/027868290903907.

Hu, W., Campuzano-Jost, P., Day, D. A., Croteau, P., Canagaratna, M. R., Jayne, J. T., Worsnop, D. R. and Jimenez, J. L.: Evaluation of the new capture vapourizer for aerosol mass spectrometers (AMS) through laboratory studies of inorganic species, Atmos. Meas. Tech., 10(6), 2897–2921, doi:10.5194/amt-10-2897-2017, 2017.

Hu, W., D.A. Day, P. Campuzano-Jost, B.A. Nault, T. Park, T. Lee, P. Croteau, M.R. Canagaratna, J.T. Jayne, D.R. Worsnop, J.L. Jimenez. Evaluation of the new capture vaporizer for Aerosol Mass Spectrometers: characterization of organic aerosol mass spectra. Aerosol Sci. Technol., 52,725-739. doi: 10.1080/02786826.2018.1454584, 2018a.

Hu, W., D.A. Day, P. Campuzano-Jost, B.A. Nault, T. Park, T. Lee, P. Croteau, M.R. Canagaratna, J.T. Jayne, D.R. Worsnop, J.L. Jimenez. Evaluation of the new capture vaporizer for Aerosol Mass Spectrometers (AMS): Elemental composition and source apportionment of organic aerosols (OA). ACS Earth Space Chem., 2, 410-421, doi:10.1021/acsearthspacechem.8b00002, 2018b.

Huffman, J. A, Jayne, J. T., Drewnick, F., Aiken, a C., Onasch, T., Worsnop, D. R. and Jimenez, J. L.: Design, modeling, optimization, and experimental tests of a particle beam width probe for the aerodyne aerosol mass spectrometer, Aerosol Sci. Technol., 39(12), 1143–1163, doi:Doi 10.1080/02786820500423782, 2005.

Jimenez, J. L., Canagaratna, M. R., Drewnick, F., Allan, J. D., Alfarra, M. R., Middlebrook, A. M., Slowik, J. G., Zhang, Q., Coe, H., Jayne, J. T., Worsnop, D. R., Rami Alfarra, M., Middlebrook, A. M., Slowik, J. G., Zhang, Q., Coe, H., Jayne, J. T. and Worsnop, D. R.: Comment on "The effects of molecular weight and thermal decomposition on the sensitivity

of a thermal desorption aerosol mass spectrometer," Aerosol Sci. Technol., 50(9), i–xv, doi:10.1080/02786826.2016.1205728, 2016.

Knote, C., Brunner, D., Vogel, H., Allan, J., Asmi, A., Äijälä, M., Carbone, S., van der Gon, H. D., Jimenez, J. L., Kiendler-Scharr, A., Mohr, C., Poulain, L., Prévôt, A. S. H., Swietlicki, E. and Vogel, B.: Towards an online-coupled chemistry-climate model: evaluation of trace gases and aerosols in COSMO-ART, Geosci. Model Dev., 4(4), 1077–1102, doi:10.5194/gmd-4-1077-2011, 2011.

Kuwata, M., Zorn, S. R. and Martin, S. T.: Using Elemental Ratios to Predict the Density of Organic Material Composed of Carbon, Hydrogen, and Oxygen, Environ. Sci. Technol., 46(2), 787–794, doi:10.1021/es202525q, 2012.

Liu, Peter S. K., Rensheng Deng , Kenneth A. Smith , Leah R. Williams , John T. Jayne , Manjula R. Canagaratna , Kori Moore , Timothy B. Onasch , Douglas R. Worsnop & Terry Deshler (2007) Transmission Efficiency of an Aerodynamic Focusing Lens System: Comparison of Model Calculations and Laboratory Measurements for the Aerodyne Aerosol Mass Spectrometer, Aerosol Science and Technology, 41:8, 721-733, DOI: 10.1080/02786820701422278

Ng, N.L., M.R. Canagaratna, Q. Zhang, J.L. Jimenez, J. Tian, I.M. Ulbrich, J.H. Kroll, K.S. Docherty, P.S. Chhabra, R. Bahreini, S.M. Murphy, J.H. Seinfeld, L. Hildebrandt, N.M. Donahue, P.F. DeCarlo, V.A. Lanz, A.S.H. Prevot, E. Dinar, Y. Rudich, and D.R. Worsnop. Organic Aerosol Components observed in Northern Hemispheric Datasets measured with Aerosol Mass Spectrometry.Atmospheric Chemistry and Physics, 10, 4625-4641, doi:10.5194/acp-10-4625-2010, 2010

Ng, N.L., M.R. Canagaratna, J.L. Jimenez, P.S. Chhabra, J.H. Seinfeld, and D.R. Worsnop. Changes in Organic Aerosol Composition with Aging Inferred from Aerosol Mass Spectra. Atmospheric Chemistry and Physics, 11, 6465-6474, doi:10.5194/acp-11-6465-2011, 2011

Pajunoja, A., Hu, W., Leong, Y. J., Taylor, N. F., Miettinen, P., Palm, B. B., Mikkonen, S., Collins, D. R., Jimenez, J. L., and Virtanen, A.: Phase state of ambient aerosol linked with water uptake and chemical aging in the southeastern US, Atmos. Chem. Phys., 16, 11163-11176, https://doi.org/10.5194/acp-16-11163-2016, 2016.

Stolzenburg, Mark R. & Peter H. McMurry. Method to assess performance of scanning mobility particle sizer (SMPS) instruments and software. Aerosol Science and Technology, Volume 52, 2018 - Issue 6.

Takegawa, N., Miyakawa, T., Watanabe, M., Kondo, Y., Miyazaki, Y., Han, S., Zhao, Y., van Pinxteren, D., Bruggemann, E., Gnauk, T., Herrmann, H., Xiao, R., Deng, Z., Hu, M., Zhu, T. and Zhang, Y.: Performance of an Aerodyne Aerosol Mass Spectrometer (AMS) during Intensive Campaigns in China in the Summer of 2006, Aerosol Sci. Technol., 43(3), 189–204, doi:10.1080/02786820802582251, 2009.

Wiedensohler, A., Birmili, W., Nowak, A., Sonntag, A., Weinhold, K., Merkel, M., Wehner, B., Tuch, T., Pfeifer, S., Fiebig, M., Fjäraa, A. M., Asmi, E., Sellegri, K., Depuy, R., Venzac, H., Villani, P., Laj, P., Aalto, P., Ogren, J. A., Swietlicki, E., Williams, P., Roldin, P., Quincey, P., Hüglin, C., Fierz-Schmidhauser, R., Gysel, M., Weingartner, E., Riccobono, F., Santos, S., Grüning, C., Faloon, K., Beddows, D., Harrison, R., Monahan, C., Jennings, S. G., O'Dowd, C. D., Marinoni, A., Horn, H.-G., Keck, L., Jiang, J., Scheckman, J., McMurry, P. H., Deng, Z., Zhao, C. S., Moerman, M., Henzing, B., de Leeuw, G., Löschau, G., and Bastian, S.: Mobility particle size spectrometers: harmonization of technical standards and data structure to facilitate high quality long-term observations of atmospheric particle number size distributions, Atmos. Meas. Tech., 5, 657-685, https://doi.org/10.5194/amt-5-657-2012, 2012.

Wiedensohler, A., A. Wiesner, K. Weinhold, W. Birmili, M. Hermann, M. Merkel, T. Müller, S. Pfeifer, A. Schmidt, T. Tuch, F. Velarde, P. Quincey, S. Seeger & A. Nowak (2018) Mobility particle size spectrometers: Calibration procedures and measurement uncertainties, Aerosol Science and Technology, 52:2, 146-164, DOI: 10.1080/02786826.2017.1387229.

Xu, W., Lambe, A., Silva, P., Hu, W., Onasch, T., Williams, L., Croteau, P., Zhang, X., Renbaum-Wolff, L., Fortner, E., Jimenez, J. L., Jayne, J., Worsnop, D. and Canagaratna, M.: Laboratory evaluation of species-dependent relative ionization efficiencies in the Aerodyne Aerosol Mass Spectrometer, Aerosol Sci. Technol., doi:10.1080/02786826.2018.1439570, 2018.

Zhang, Y., Tang, L., Croteau, P. L., Favez, O., Sun, Y., Canagaratna, M. R., Wang, Z., Couvidat, F., Albinet, A., Zhang, H., Sciare, J., Prévôt, A. S. H., Jayne, J. T. and Worsnop, D. R.: Field characterization of the PM2.5 Aerosol Chemical Speciation Monitor: insights into the composition, sources, and processes of fine particles in eastern China, Atmos. Chem. Phys., 17(23), 14501–14517, doi:10.5194/acp-17-14501-2017, 2017.

Campuzano-Jost, P., Hu, W, Jimenez, J.L., et al., The importance of lens transmission calibration, 18th AMS (&CIMS) Users Meeting, Beijing, China 8-11 May 2017. http://cires1.colorado.edu/jimenez-group/UsrMtgs/UsersMtg18/2016_5_08_Weiwei_LensTransimission_V2.pdf

---

## Author Comment (AC1) · 1 Oct 2018

**We are very grateful for the reviewer's comments. A point-to-point response to reviewers' comments is provided below, and the manuscript has been revised accordingly. The line numbers in the response refer to those in the revised version of the manuscript.**

**Reviewer #1:** *"The manuscript by Li et al. investigated the sensitivity of a Q-ACSM for the measurements of chamber secondary organic aerosol (SOA) from three volatile organic compounds. The authors found that the sensitivity of the Q-ACSM to SOA was found to be anti-correlated with the aerosol oxidation state regardless of the VOC precursors. This study also shows the applicability of Q-ACSM in chamber studies although rare studies use Q-ACSM for chamber studies because of low sensitivity and time resolution. This manuscript is generally well written and fits within the scope of AMT. However, some conclusions are overstated and need more evidence."*

**Major comments:**

1. *"The authors concluded CE as a function of particle phase states, which needs to be reconsidered. First, the RH was maintained at 10 – 15% during the experiments, we could not expect the changes in phase states of SOA particles. Second, Docherty et al. (2013) also showed the variations of CE as a function of f44/f57, which is primarily caused by particle bounce at the vaporizer."*

Response: We agree with the reviewer that the particle bounce at the vaporizer can explain most of the observed anti-correlation between $CE_{org}$ and $f_{44}$. The "phase states" here were referring to the phase state of dry SOA particles without the uptake of water. For example, some organics (e.g., oleic acid and nonylaldehyde) with a CE of near unit are intrinsically liquid (Docherty et al., 2013), while smog chamber-generated SOA particles formed via the oxidation of biogenic emission from scots pine appear to be in amorphous semisolid or amorphous solid (glassy) states, rather than being liquid (Virtanen et al., 2010). Particle morphology research based on glass transition temperature ($T_g$) indicated that during oxidation of liquid pinene, $T_g$ increased from much lower than room temperature (175K) to near room temperature (290K), indicating that the SOA products might vary between liquid state, semi-solid state and solid (glassy) state resulting from a combined effect of increasing molecular weight and O/C (Koop et al., 2011). Accordingly, we believed that the anti-correlation between $CE_{org}$ and $f_{44}$ was actually caused by the phase change with $f_{44}$. This was supported by the measured effective density that increased from 1.07 to 1.34 with increasing $f_{44}$. The explanation could be that SOA particles with higher $f_{44}$ would have higher density and would act more like a "bouncing" solid ball instead of a "sticky" droplet. This has been discussed in lines 318-322.

2. *"It is difficult for this manuscript to conclude that RIE changes as SOA evolves from HOA to more oxygenated OOA. This is an SOA experiment, which cannot give you any information on HOA. Many factors can affect the anti-correlation between RF and f44, but RIE would not be the important one. As shown in the latest research by Xu et al. (2018), organic aerosol with OS below -1.5 can have much high RIE, but in this study, the f44 does not support this."*

Response: This work did not cover HOA particles and thus we removed all statements related to HOA. In the case of SOA, we have reevaluated the role of CE and RIE on the observed anti-correlation between RRF (relative response factor) and $f_{44}$. Within the studied SOA range, we found that the variation of $CE_{org}$ can largely explain the observed RFF changes with $f_{44}$, indicating that variations of $RIE_{org}$ with $f_{44}$ may be limited in our experiments. Recently, Xu et al. (2018) showed that $RIE_{org}$ for organic with -1.0 < $\overline{OSc}$ (average carbon oxidation states) < 0.5 varied between 1.6 ± 0.5 (2σ), which is relatively higher than

the widely used default value ($RIE_{org} = 1.4$). Clearly, a higher than default $RIE_{org}$ value is consistent with the overall positive deviations of RRF observed in our experiments (Lines 327-332).

3. *"I would also suggest the authors comparing the result in this study with those previously reported by HR-ToF-AMS. For example, f44 vs. f43 could be different. Q-ACSM might reports much higher f44 than that of HR-ToF-AMS (Fröhlich et al., 2015)."*

Response: As suggested by the reviewer, we have inserted *Section 3.4 Comparison with AMS Calibrations* (Line 299) into the revised manuscript. As indicated by the reviewer, these is intrinsic difference between a Q-ACSM and an AMS. For example, ammonium nitrate in the ambient aerosol sample may affect the m/z 44 signal in Q-ACSM due to the low mass resolution of Q-ACSM (Fröhlich et al., 2015). Nevertheless, the calibration results obtained from this work are expected to be comparable with those done with the AMS. The following statement has been inserted into section 3.4 (Lines 303-317):

"In previous studies, CE was demonstrated to be variable for particles with different chemical composition, phase state and under different RH, with $CE_{org}$ ranging from less than 0.15 to ~ 1 (Docherty et al., 2013;Alfarra, 2004;Robinson et al., 2015;Matthew et al., 2008). CE for biogenic SOA has been reported to be close to 1 based on both chamber experiments (Kiendler-Scharr et al., 2009) and field measurements conducted in Amazon, where aerosols were dominated by liquid SOA (Allan et al., 2014;Chen et al., 2009). However, it has been proposed in theoretical, chamber, and field studies that besides liquid state, organic particles can exist in semi-solid or solid state (Virtanen et al., 2010;Vaden et al., 2010;Shiraiwa et al., 2011), which might lead to lower $CE_{org}$. For example, $CE_{org}$ of liquid squalane (CE = 1) decreased with SOA condensation (Robinson et al., 2015). Most recent $CE_{org}$ calibration with light-scattering single-particle (LSSP) module also suggested that $CE_{org}$ (0.25 ~ 0.4) could be much less than 1 (Robinson et al., 2017). An inverse relationship between $CE_{org}$ and $f_{44}/f_{57}$ ratio in chamber SOA generation experiments (Docherty et al., 2013) was very similar to our results. However, our results could not be quantitatively compared with the previous ones because the Q-ACSM might report comparably higher $f_{44}$ than the AMS (Fröhlich et al., 2015)."

References:

Alfarra, M.: Insights into atmospheric organic aerosols using an aerosol mass spectrometer, University of Manchester, 2004.

Allan, J. D., Morgan, W. T., Darbyshire, E., Flynn, M. J., Williams, P. I., Oram, D. E., Artaxo, P., Brito, J., Lee, J. D., and Coe, H.: Airborne observations of IEPOX-derived isoprene SOA in the Amazon during SAMBBA, Atmos Chem Phys, 14, 11393-11407, 2014.

Chen, Q., Farmer, D. K., Schneider, J., Zorn, S. R., Heald, C. L., Karl, T. G., Guenther, A., Allan, J. D., Robinson, N., Coe, H., Kimmel, J. R., Pauliquevis, T., Borrmann, S., Poeschl, U., Andreae, M. O., Artaxo, P., Jimenez, J. L., and Martin, S. T.: Mass spectral characterization of submicron biogenic organic particles in the Amazon Basin, Geophysical Research Letters, 36, 10.1029/2009gl039880, 2009.

Docherty, K. S., Jaoui, M., Corse, E., Jimenez, J. L., Offenberg, J. H., Lewandowski, M., and Kleindienst, T. E.: Collection Efficiency of the Aerosol Mass Spectrometer for Chamber-Generated Secondary Organic Aerosols, Aerosol Sci Tech, 47, 294-309, 10.1080/02786826.2012.752572, 2013.

Fröhlich, R., Crenn, V., Setyan, A., Belis, C. A., Canonaco, F., Favez, O., Riffault, V., Slowik, J. G., Aas, W., and Aijälä, M.: ACTRIS ACSM intercomparison-Part 2: Intercomparison of ME-2 organic source apportionment results from 15 individual, co-located aerosol mass spectrometers, 2015.

Kiendler-Scharr, A., Zhang, Q., Hohaus, T., Kleist, E., Mensah, A., Mentel, T. F., Spindler, C., Uerlings, R., Tillmann, R., and Wildt, J.: Aerosol Mass Spectrometric Features of Biogenic SOA: Observations from a Plant Chamber and in Rural Atmospheric Environments, Environ Sci Technol, 43, 8166-8172, 10.1021/es901420b, 2009.

Koop, T., Bookhold, J., Shiraiwa, M., and Pöschl, U.: Glass transition and phase state of organic compounds: dependency on molecular properties and implications for secondary organic aerosols in the atmosphere, Physical Chemistry Chemical Physics, 13, 19238-19255, 2011.

Matthew, B. M., Middlebrook, A. M., and Onasch, T. B.: Collection efficiencies in an Aerodyne Aerosol Mass Spectrometer as a function of particle phase for laboratory generated aerosols, Aerosol Sci Tech, 42, 884-898, 10.1080/02786820802356797, 2008.

Mikhailov, E., Vlasenko, S., Martin, S., Koop, T., and Pöschl, U.: Amorphous and crystalline aerosol particles interacting with water vapor: conceptual framework and experimental evidence for restructuring, phase transitions and kinetic limitations, Atmos Chem Phys, 9, 9491-9522, 2009.

Robinson, E. S., Saleh, R., and Donahue, N. M.: Probing the evaporation dynamics of mixed SOA/squalane particles using size-resolved composition and single-particle measurements, Environ Sci Technol, 49, 9724-9732, 2015.

Robinson, E. S., Onasch, T. B., Worsnop, D., and Donahue, N. M.: Collection efficiency of [alpha]-pinene secondary organic aerosol particles explored via light-scattering single-particle aerosol mass spectrometry, Atmos Meas Tech, 10, 1139, 2017.

Shiraiwa, M., Ammann, M., Koop, T., and Pöschl, U.: Gas uptake and chemical aging of semisolid organic aerosol particles, Proceedings of the National Academy of Sciences, 108, 11003-11008, 2011.

Vaden, T. D., Song, C., Zaveri, R. A., Imre, D., and Zelenyuk, A.: Morphology of mixed primary and secondary organic particles and the adsorption of spectator organic gases during aerosol formation, Proceedings of the National Academy of Sciences, 107, 6658-6663, 2010.

Virtanen, A., Joutsensaari, J., Koop, T., Kannosto, J., Yli-Pirilä, P., Leskinen, J., Mäkelä, J. M., Holopainen, J. K., Pöschl, U., and Kulmala, M.: An amorphous solid state of biogenic secondary organic aerosol particles, Nature, 467, 824, 2010.

Xu, W., Lambe, A., Silva, P., Hu, W., Onasch, T., Williams, L., Croteau, P., Zhang, X., Renbaum-Wolff, L., Fortner, E., Jimenez, J. L., Jayne, J., Worsnop, D., and Canagaratna, M.: Laboratory evaluation of species-dependent relative ionization efficiencies in the Aerodyne Aerosol Mass Spectrometer, Aerosol Science and Technology, 52, 626-641, 10.1080/02786826.2018.1439570, 2018.

Young, K. D., and LeBoeuf, E. J.: Glass transition behavior in a peat humic acid and an aquatic fulvic acid, Environ Sci Technol, 34, 4549-4553, 2000.

---

## Author Comment (AC2) · 1 Oct 2018

We are very grateful for the reviewer's comments. A point-to-point response to reviewers' comments is provided below, and the manuscript has been revised accordingly. The line numbers in the response refer to those in the revised version of the manuscript.

**Reviewer #2:** "This paper reports results from very difficult experiments aimed at understanding the sensitivity of the ACSM (Aerosol Chemical Speciation Monitor) instruments to organic material from aerosol particles. The topic of varying relative ionization efficiency for organic aerosol mass with the Aerosol Mass Spectrometer (AMS) has been debated in recent literature (Murphy, 2016a; Jimenez et al., 2016; Murphy, 2016b). Because the AMS has similar characteristics to the ACSM (aerodynamic aerosol focusing lens followed by thermal vaporization, electron impact ionization, and MS detection), most of the results described in this paper are potentially applicable to the AMS.

The findings of this paper are fascinating and the reported changes in SOA "sensitivity" are qualitatively consistent with many previous observations. A major strength of this work is that surrogates for ambient secondary organic aerosols (SOA) were produced and important details on the chamber experiments are not needed to demonstrate the overall method for accurately measuring the SOA response factors for the ACSM. Unfortunately, this work does not provide any new, useful and quantitative information for the ACSM/AMS community and it is possible that the data from this work are unable to constrain the largest uncertainties for a broader application of these results. To make a significant contribution in the field, the work presented here should show all of the relevant calibration details and significantly reduce the largest uncertainties in the reported measurements. At a minimum, the manuscript must show that the methods work for aerosols with known chemical composition (ammonium nitrate and ammonium sulfate) and the uncertainties must be fully propagated. The paper has several major flaws that need to be addressed in revising this manuscript to make it acceptable for publication in AMT."

Response: We agree with the reviewer that calibration of a Q-ACSM or an AMS is a very difficult job and we also believe AMS/ACSM is a very powerful instrument that has made great contribution to the aerosol researches. Therefore, the accuracy of AMS/ACSM measurements is a critical factor that deserves extensive studies using various methodologies. In this work, SOA were generated inside a Teflon chamber from different VOC precursors under atmospheric relevant conditions. The volume concentration and effective density of chamber-generated SOAs were quantified by SMPS and APM, respectively. The Q-ACSM, SMPS, and APM were well calibrated before or during our experiments with ammonium nitrate, ammonium sulfate, or PSL spheres. The calibration procedures and results were provided in more details in the revised manuscript and the supplementary information. In addition, we conducted some additional calibration work on Q-ACSM (the transmission efficiency of particles in the aerodynamic lens) as suggested by the reviewer. With all these efforts, we were able to present a comprehensive evaluation of the measurement uncertainties. Our results strongly indicate that measurements of OA with various  $f_{44}$  by a Q-ACSM using constant conversion factors may induce significant errors in mass concentration measurements for laboratory-generated SOA. The results of this work appeared to be consistent with the results of previous work using different experimental settings and reconfirmed the reliability of the AMS/ACSM techniques. Currently, the uncertainties of OA measurement ( $2\sigma = \pm 38\%$ ) is still quite high. Although the results of this work may not be directly applicable to ambient Q-ACSM measurements, it is reasonable to suggest that similar comprehensive calibrations of Q-ACSM for complicated ambient conditions be done in the future to further constrain the uncertainties of ambient OA measurement.

**Specific comments:**

**Writing and Presentation of Key Concepts**

1. "This paper was very difficult to read due to numerous grammatical errors and informal usage of important terminology (e.g. response factor or RF, relative ionization efficiency or RIE, collection efficiency or CE, and oxidation state). Broad claims of "accuracy," interpretation of the results, and the relevance of this work are not supported by the actual evidence presented. The manuscript would significantly benefit from more careful technical writing and editing."

Response: As suggested by the reviewer, we have invited experienced native English speaker to fully revise the manuscript for grammatical errors. All the AMS related terminologies have been clearly defined in the revised manuscript. The definitions of "Collection Efficiency (CE)", "Ionization Efficiency (IE)", "Relative Ionization Efficiency (RIE)" and "Response Factor (RF)" were all described in the introduction section. All the obscure expressions, like "accuracy" and "sensitivity", were replaced with "RF" or "Relative Response Factor (RRF)", the latter was a new term introduced by the authors to quantitatively express the difference between Q-ACSM and SMPS-APM measurement. It has been discussed in lines 177-182: "To elucidate how Q-ACSM measured mass concentration ( $PM_{ACSM}$ , using default RIEorg = 1.4 and CEorg = 0.5) may deviate from the true mass concentration for particles entered Q-ACSM ( $PM_{SMPS}$ , the product of EL and ESamended SMPS-measured volume concentration and APM-measured  $\rho_{eff}$ , relative response factor (RRF) is defined as the ratio of  $PM_{ACSM}$  to  $PM_{SMPS}$ .

$$RRF = \frac{PM_{ACSM}(assuming \ default \ RIE_{org}=1.4, CE_{org}=0.5)}{PM_{SMPS}(amended \ by \ \rho_{eff}, Es, E_L)} = \frac{RIE_{org} \cdot CE_{org}}{1.4 \cdot 0.5}$$
(2)"

In addition, the conclusion of this work was reevaluated and has been constrained to the laboratorygenerated SOAs only. The original Figure 8 has been removed to avoid confusion.

2. "The primary subject of this study, the response factor (RF) for the ASCM, should be clearly defined at the beginning of the manuscript with Equation 1 moved up to the end of the paragraph on the AMS/ACSM quantification (page 5 line 103). This equation shows that both RIE and CE are important factors in determining the RF, therefore any variation in RF could be due to either one or both of these factors changing. While this was described later in the manuscript (Sections 3.3 Effects of CE and 3.4 Effects of RIE), these concepts are key to understanding and interpreting the reported RFs from these studies. The abstract should state that the RF values reported here were obtained using the default RIE for organic species (1.4) along with a CE of 0.5."

Response: As suggested by the reviewer, the original equation together with the definition of RF and its relationship with CE and RIE have been moved to the introduction (Line 84). When introducing the calculating method of RRF (section 2.4), it was stated clearly that "any non-unit value of RRF might be caused by the deviation of  $RIE_{org}$  or  $CE_{org}$  from the default values" (Lines 180-181). It has been clearly stated in the abstract and throughout the manuscript that the default RIE (1.4) and CE (0.5) were used to calculate the RRF in this work.

3. "Accurately quantifying RIE and CE factors is crucial in determining the AMS/ACSM sensitivity to aerosol components and they need to be studied independently in order to generally apply the findings of this work. The work made an attempt at separating these factors, but did not adequately examine them for the analysis. The current debate in the literature is about RIE for organic aerosol (Murphy, 2016a, b; Jimenez et al., 2016) and the present work appears to contradict itself in the findings on RIE. This is probably due to careless usage of the terms "RIE" and "RF," as well as not incorporating the CE findings into calculations for the RIEs from the measured RFs. A thorough examination of the writing of the paper would presumably clarify the inconsistencies and demonstrate if the RIE varied significantly or not for these measurements.

Response: As suggested by the reviewer, we have clarified the definitions of RRF and RIEorg (Lines 177-182). Although we have attempted to separate the effects of  $CE_{org}$  and  $RIE_{org}$ ,  $RIE_{org}$  cannot be quantitatively determined because of the relatively large uncertainties and different  $f_{44}$  ranges generated between experiments with and without AS seeds. Based on additional calibrations, uncertainties associated with each instrument has been well quantified. For example, we have performed extra experiments to calibrate

the ACSM lens transmission efficiency ( $E_L$ ) and the calibrated  $E_L$  was used to amend the PMSMPS. We have re-evaluated the uncertainties in SMPS and APM measurements. The SOA coating thickness for the AS seed particles has also been precisely determined. We clearly redefined "relative stable states" in each experiment. Based on these data analysis, we found that the variation of RRF as a function of  $f_{44}$  can be largely explained by the CEorg variation, which was possibly due to particle phase changes as indicated by the effective density variation. Nevertheless, we cannot exclude the possible influence of RIEorg. The manuscript has also been thoroughly revised.

4. "The last sentence in the abstract states "Our results indicated that the current Q-ACSM calibration procedure using a constant RIE may lead to somewhat underestimation of more oxidized OOA but overestimation of less oxidized HOA, i.e., a variable RIE shall be applied, most likely as a function of the SOA oxidation state." In addition to the obvious mistakes in the grammar and use of terminology, this statement implies that previous measurements from the ACSM/AMS are incorrectly reporting OA mass concentrations. This statement needs to be supported with clear evidence of such problems. The results here, while not directly showing it, imply that the variations in the response factor can be largely explained by variations in CE rather than variations in RIE. Many studies have been using a higher CE for HOA that accounts for discrepancies between independent measurements and the AMS, for example the 2002 Pittsburgh study (Zhang et al., 2005) and subsequent work. How do the authors reconcile their results with those showing numerous, good correlations of ambient ACSM/AMS data with independent organic carbon aerosol or total mass measurements?"

Response: Since HOA was not studied in this work, the last sentence in the original abstract has been removed to avoid overstating the conclusion of this work. After conducting more instrument calibrations, uncertainty evaluation, and data analysis work, we concluded that the variation of  $CE_{org}$  with  $f_{44}$  could explain a large fraction of the observed decrease in RRF, while the influence of  $RIE_{org}$  cannot be excluded. Our results of RRF and  $CE_{org}$  calibration were generally in line with previous studies conducted with the AMS in laboratory experiments. This was not in conflict with the various ambient measurements because organics are usually internally mixed with inorganics in ambient conditions and CE of the bulk aerosol can still keep around 0.5. However, the uncertainties for ambient OA measurements is still quite high (±38%), which have the potential to be constrained by applying more accurate RIEorg values according to  $f_{44}$ . It was a pity that we cannot accurately quantify RIEorg with our experiment, which was largely attribute to the low resolution of Q-ACSM and the large uncertainties in SMPS volume measurements. However, it is highly suggested that more comprehensive calibrations of RIEorg on AMS/ACSM should be done in both laboratory and ambient conditions to further lower the uncertainties of OA measurement.

**5. "Accuracy and Uncertainties for the Number Distributions and Effective Density Measurements:**

In order to calculate the SOA mass concentrations being generated in the chamber, the measured number distributions as a function of particle size need to be converted into mass distributions using the particle density. Determining the overall uncertainty in the mass concentrations from the number distribution and the particle density is important. Here, the number distributions were measured with a scanning mobility particle analyzer (SMPS). Information about calibrating this instrument was not given in the paper, so it is unclear what the accuracy is for these measurements. Also, how well did the two number distributions match between the SMPS and the differential mobility analyzer (DMA) plus condensation particle counter (CPC) systems? A simple statement about using polystyrene latex spheres to calibrate both systems would provide additional confidence in these measurements.

Knowing the accuracy and uncertainty in the particle effective density measurements is also necessary. The

particle effective density was determined by selecting the mobility diameter at the peak in the SMPS number distribution with the DMA and the peak in the mass per particle was scanned with an aerosol particle mass (APM) analyzer with a CPC. The literature reports using a similar DMA/APM/CPC system can provide accurate effective density measurements with 95% confidence intervals of 10-30%, depending on the configuration of the system (Johnson et al., 2013). The accuracy of the system used for this study was not presented as additional calibration information. For example, how accurate and precise are the effective densities for ammonium nitrate and ammonium sulfate (or other known particles)? The demonstrated uncertainties in the effective density measurements need to be propagated to the rest of the results.

The peak in the number distribution is not the same as the peak in the mass distribution. Were additional sizes of mobility diameter scanned (not just the peak in the number distribution) to ensure that the effective density was constant across all sizes in the distribution? If not, it should be clearly stated that it was assumed that the measured effective density was the same for the entire distribution. "

Response: As suggested by this reviewer and reviewer #4, we have provided more details of calibration methods and results of the SMPS and APM system in the manuscript (Lines 170-175): "The flow rates and the voltages applied to the DMAs were well calibrated before the experiments. Three sizes of polystyrene latex spheres (PSL,  $81\pm3nm$ ,  $147\pm3nm$ ,  $269nm\pm5nm$ ,  $1.05 \text{ g cm}^{-3}$ ) were aerosolized and dried with a collision atomizer (TSI, Model 3076) followed by a diffusion dryer filled with silica gel for size calibration of the SMPS system. The same PSL spheres were also used for the calibration of APM (see Fig. S3). The bias of SMPS size and APM effective density measurements were within  $\pm4\%$  and  $\pm3\%$ , respectively." Details of the calibrations can be found in the SI section 1 and 2."

When using APM to measure the effective density of AS seeds, " $\rho_{eff}$  of the dry 80 nm AS particles was measured to be  $1.64 \pm 0.08$  g cm-3, which was consistent with 1.65 g cm-3 (Zelenyuk et al., 2006) or slightly lower than 1.66-1.70 g cm-3 measured for 200 nm ( $d_m$ ) dry AS particles in a previous study (Matthew et al., 2008)" (Lines 246-249). However, as the bias were within  $\pm 3\%$ , it was within the evaluated uncertainties of our results.

In this work, we consider the chamber-generated SOA were mostly unimodal distributed (as shown in Fig. S4). The number-size distribution measured by the SMPS was used to deduce the aerosol total volume. The peak size of the distribution was considered the most representative size of the SOA population. The APM was used to measure the true mass of a certain size (mobility diameter) of particle and thus APM is usually used with a DMA for size selection first (McMurry et al., 2002). The mass distribution obtained by the APM was determined by the resolution of the APM, i.e., its rod rotational speed and its physical radius. The magnitude of the APM peak was only used to determine the exact mass of a certain size particle not the actual number concentration of that size of particle in the chamber. Accordingly, the effective density of the SOA population was calculated, which was then used to calculate the total mass of the SOA. As SOA continued to grow as the experiment proceeded, it was important to track the SOA changes with high timeresolution. Typically, one SMPS-APM scan can be completed within 10 min, which cannot be achieved by the traditional filter-based aerosol measurement technique. We have tested the APM with a certain size of PSL, slight change in peak selection of the DMA setting by a few nm, the mass measured by the APM was not significantly affected but the magnitude of the peak was substantially reduced. Therefore, we believe the measured effective density was the same for the entire distribution, especially for the AS seeded experiments, when only a single mode of AS particles was present in the chamber. The following discussion has been given in the manuscript (Line 184) and the supplementary information (Line 71):

$$PM_{SMPS} = V_{SMPS} \cdot \rho_{eff} \cdot E_L \cdot E_s = \sum_{i=1}^n \{\frac{\pi}{6} d_{m,i}^3 \cdot dN_i \cdot \rho_{eff,i} \cdot E_{L,i} \cdot E_{s,i}\}$$
(3)

Eq. (3) is used to calculate PMSMPS, where VSMPS is the volume concentration calculated with SMPS measured size distribution; n is the total number of bins for SMPS measurements;  $d_{m,i}$  and  $dN_i$  are respectively the geometric mean diameter and the number of counts in size bin i;  $E_{L,i}$  and  $E_{S,i}$  are the aerodynamic lens transmission efficiency and shape factor for particles of  $d_{m,i}$ . After applying  $E_{L,i}$  and  $E_{S,i}$ , any deviation in CEorg should be caused only by particle bounce on the vaporizer surface (Eb). The measured EL (Fig. S2) is in reasonable accordance with previous studies (Jayne et al., 2000; Knote et al., 2011; Hu et al., 2017), while larger than other reports (Liu et al., 2007) as for 300 nm~1000 nm ( $d_{va}$ ) particles. ES was assumed to be 1±5%.  $d_m$  and  $d_{va}$  were convertible via  $d_{va}=d_m \cdot \rho_{eff}/\rho_0$  for a reference density  $\rho_0$  of 1000 kg m-3 (Jayne et al., 2000; DeCarlo et al., 2004). The uncertainties of  $d_{m,i}$  (± 4%),  $dN_i$  (± 10% for 20~200 nm and ±20% for 200~800 nm particles) (Wiedensohler et al., 2012; Wiedensohler et al., 2018),  $\rho_{eff}$  (±3%), EL (±10%) and Es (± 5%) were well propagated to the calculation. As a result, the relative uncertainty of PMSMPS was between 19.4 ~ 26.0%, depending on the actual size distribution. Details of uncertainty evaluation can be found in supplementary information."

"Theoretically, the uncertainty of PMSMPS ( $\sigma_{PM-SMPS}$ ) was associated with dm,i, dNi,  $\rho_{eff,i}$ , EL(dm,i), and Es,i. As discussed in the instrument calibration in Section 1, uncertainties of dm,i ( $\sigma_{dm}$ ) and  $\rho_{eff}$  ( $\sigma_{\rho}$ ) were within  $\pm 4\%$  and  $\pm 3\%$ , respectively. The uncertainty of EL ( $\sigma_{EL}$ ) in the range of 40-600 nm (dva) was estimated to be  $\pm 10\%$  according to Figure S2. The uncertainty of dNi ( $\sigma_N$ ), which was the combination of the uncertainties of particle charging efficiency ( $\sigma_{char}$ ), diffusion loss ( $\sigma_{diff}$ ), and CPC counting efficiency ( $\sigma_{diff}$ ) (Buonanno et al., 2009), was assigned to be  $\pm 10\%$  for particles with a diameter of 20-200 nm and  $\pm 20\%$  for 200-800 nm (Wiedensohler et al., 2012; Wiedensohler et al., 2018). The uncertainty of Es ( $\sigma_{ES}$ ) was estimated to be  $\pm 5\%$ . The uncertainty of PMSMPS was hence estimated by Eq. (S3) using the averaged sized distribution (as shown in Figures S4 and S6) and  $\rho_{eff}$  in each experiment:

$$= \sqrt{\sum_{i=1}^{n} PM_{SMPS}^{2}(d_{m,i}) + \sum_{i=1}^{n} PM_{SMPS}^{2}(dN_{i}) + \sum_{i=1}^{n} PM_{SMPS}^{2}(\rho_{eff,i}) + \sum_{i=1}^{n} PM_{SMPS}^{2}(E_{L}(d_{m,i})) + \sum_{i=1}^{n} PM_{SMPS}^{2}(E_{s,i})}$$
(S3)

Where,  $PM_{SMPS}(x_i)$  is the product of the uncertainties of each bin  $x_i$  ( $\sigma_{x,i}$ ), and the corresponding sensitivity coefficient  $\frac{\partial PM_{SMPS,i}}{\partial x_i}$ , as shown in Eq. (S4):

$$PM_{SMPS}(x_i) = \frac{\partial PM_{SMPS,i}}{\partial x_i} \cdot \sigma_{x,i}$$
(S4)

Then, Eq (S3) can be simplified as Eq. (S5):

$$\sigma_{PM-SMPS} = \sqrt{\sum_{i=1}^{n} (9\sigma_{dm,i}^2 + \sigma_{dN,i}^2 + \sigma_{\rho_{eff,i}}^2 + \sigma_{EL,i}^2 + \sigma_{ES,i}^2) P M_{SMPS,i}^2} = \sum_{i=1}^{n} (9\sigma_{dm,i}^2 + \sigma_{dN,i}^2 + \sigma_{\rho_{eff,i}}^2 + \sigma_{EL,i}^2 + \sigma_{ES,i}^2) P M_{SMPS,i}^2 = \sum_{i=1}^{n} (9\sigma_{dm,i}^2 + \sigma_{dN,i}^2 + \sigma_{\rho_{eff,i}}^2 + \sigma_{EL,i}^2 + \sigma_{ES,i}^2) P M_{SMPS,i}^2 = \sum_{i=1}^{n} (9\sigma_{dm,i}^2 + \sigma_{dN,i}^2 + \sigma_{\rho_{eff,i}}^2 + \sigma_{EL,i}^2 + \sigma_{ES,i}^2) P M_{SMPS,i}^2 = \sum_{i=1}^{n} (9\sigma_{dm,i}^2 + \sigma_{dN,i}^2 + \sigma_{Peff,i}^2 + \sigma_{EL,i}^2 + \sigma_{ES,i}^2) P M_{SMPS,i}^2 = \sum_{i=1}^{n} (9\sigma_{dm,i}^2 + \sigma_{dN,i}^2 + \sigma_{Peff,i}^2 + \sigma_{EL,i}^2 + \sigma_{ES,i}^2) P M_{SMPS,i}^2 = \sum_{i=1}^{n} (9\sigma_{dm,i}^2 + \sigma_{dN,i}^2 + \sigma_{Peff,i}^2 + \sigma_{EL,i}^2 + \sigma_{ES,i}^2) P M_{SMPS,i}^2 = \sum_{i=1}^{n} (9\sigma_{dm,i}^2 + \sigma_{dN,i}^2 + \sigma_{Peff,i}^2 + \sigma_{EL,i}^2 + \sigma_{ES,i}^2) P M_{SMPS,i}^2 = \sum_{i=1}^{n} (9\sigma_{dm,i}^2 + \sigma_{EL,i}^2 + \sigma_{ES,i}^2) P M_{SMPS,i}^2 = \sum_{i=1}^{n} (9\sigma_{dm,i}^2 + \sigma_{EL,i}^2 + \sigma_{EL,i}^2 + \sigma_{EL,i}^2 + \sigma_{EL,i}^2) P M_{SMPS,i}^2 = \sum_{i=1}^{n} (9\sigma_{dm,i}^2 + \sigma_{EL,i}^2 + \sigma_{EL,i}^2 + \sigma_{EL,i}^2 + \sigma_{EL,i}^2 + \sigma_{EL,i}^2 + \sigma_{EL,i}^2) P M_{SMPS,i}^2 = \sum_{i=1}^{n} (9\sigma_{dm,i}^2 + \sigma_{EL,i}^2 + \sigma_{E$$

 $\sqrt{\sum_{i=1}^{n^*} 0.0378PM_{SMPS,i}^2 + \sum_{i=n^*}^{n} 0.0678PM_{SMPS,i}^2}$

Where, n\* corresponds to the bins with  $d_m$  larger than 200 nm. Overall, the uncertainty of PMSMPS was estimated to be between  $\pm$  19.4% and  $\pm$  26.0% with the exact value depending on the size distribution (Figures S4 and S6)."

(S5)

~

**6. "Uncertainties in Collection Efficiency:**

One of the largest issues (uncertainties) in ACSM/AMS measurements is the collection efficiency (CE). In particular, the CE factor includes lens transmission and focusing in addition to particle bounce (Canagaratna et al., 2007;Huffman et al., 2005). Lens transmission (Liu et al., 1995a, 1995b) can often significantly affect the overall (i.e., observed) CE and it was neglected in this work. Furthermore, lens transmission can vary for nominally identical lenses (Bahreini et al., 2008) and the actual lens transmission for the instrument used in this study needs to be quantified. To fairly compare an external measurement of mass with the signals from an ACSM (or AMS), a correction to the volume distribution needs to be applied that accounts for particle losses in the lens. If the actual lens transmission cannot be determined, the theoretical lens transmission must be used to determine if the ACSM lens is transmitting the majority of the mass measured by the size distribution instrumentation.

For the data presented in Figure 2, the overlap between the measured number distribution when the system was stable after 5 hours and the lens transmission might possibly be close to 100%. Thus, the effect of lens transmission losses could be minimal for that particular experiment. Figure 3 shows that the measured signals from the ACSM using the default RIE of 1.4 and CE=0.5 agree very well with the SMPS mass measurements using an effective density of 1.2; the bias is +22% which is probably within combined experimental uncertainties. For the coatings experiments, it is possible that particles grew too large to be efficiently transmitted through the lens or that the increase in sulfate mass concentration was due to smaller particles becoming more efficiently transmitted. A reader cannot estimate lens transmission losses for these and the other experiments because the mass distributions are not shown in the manuscript (or supplemental information). Therefore, the lens transmission needs to be examined for all of the experiments to ensure that the mass sampled by the AMS is the same as the mass measured with the SPMS system.

The CE for dry, ammonium sulfate particles was used as a basis for the CE results shown in Table 2 and Figures 6 and 7. Therefore, uncertainties in the CE for dry ammonium sulfate would affect the derived CE for the organic coatings. How was CE = 0.28 determined for dry ammonium sulfate? Prior work indicates that the CE for this species is 0.24+/-3% (Matthew et al., 2008). How stable is the ammonium sulfate seed source? The manuscript states that a "similar amount and size" of seed particles were used in the experiments, and the variability is a little more than 10% by averaging the mass of uncoated seeds in Table 2. Figure 6 shows that the sulfate mass concentrations are at least this variable (and possibly increasing) prior to (and shortly after) injection of the VOC precursors. Toward the end of the experiment, the sulfate mass concentrations appear to be decreasing while the organic mass loading increased more slowly than the sulfate at the middle of the experiment and seemed more stable than the sulfate at the end. (The data in Figure 6 do not appear to correspond with any of the experiments shown in Table 2.) Loss rates in the chamber would affect the interpretation of the changes in mass concentrations over the time of the coatings experiments. All of these factors contribute to uncertainties in the CE derived from these experiments, which should be added to the manuscript."

Response: As suggested by the reviewer, the lens transmission efficient ( $E_L$ ) of the Q-ACSM was calibrated using the same setup as in Figure S1 and the details were given in the supplementary information (SI Lines 32-51):

"The lens transmission efficient ( $E_L$ ) of the Q-ACSM was calibrated using the same setup as in Figure S1. The only difference was that the DMA was set to select a variety of particles with diameters ranging from 200 nm to 600 nm instead of a fixed single diameter of 300 nm. To minimize multiple charge problem for larger particles, we conducted  $E_L$  calibration under two sets of comparably low NH4NO3 solution concentrations (~2 mM and ~0.5 mM). However, for the set of experiment with extremely low NH4NO3 concentration (~0.5 mM), the Q-ACSM sensitivity was not high enough to detect NH4NO3 particles, which therefore was not shown here. Consequently, the calibration result of the lens transmission efficiency was shown in Figure S2.

Figure S2. Lens transmission efficiency as a function of particle vacuum aerodynamic diameter (dva).

Lens transmission efficiencies for 300~600nm particle were nearly unity, which was in reasonable accordance with some previous studies (Jayne et al., 2000; Knote et al., 2011; Hu et al., 2018), but the diameter range with a unity lens transmission efficiency was much broader than that reported by Liu et al. (2007). The blue line in Figure S2 was used to modify size distribution in our experiments (i.e., 0% transmission for  $d_{va}$  below 40 nm; linear increase in transmission vs  $log(d_{va})$ , from 0% to 100% at  $d_{va}=100$  nm; 100% transmission between 100 nm and 580 nm; linear decrease in transmission vs  $log(d_{va})$  from 100% at 580 nm to 0% at 1200 nm). For  $E_L$  below 300 nm, we used the results reported by Knote et al. (2011) indicated by the red dashed line in Figure S2. About 10% of uncertainty (shaded area) was applied to the  $E_L$  used in this study as indicated by the blue curve in Figure S2."

For the experiments without SA seed particles, a small portion of smaller particles (<100 nm) were lost in the lens. This has been discussed in SI: "For most of the experiments, more than 94% of the total particle mass was transported into ACSM through aerodynamic lens. While for three of them (Exp. 4, 5, 7), a portion of particles were too small to pass the lens, only 87-89% of the total mass was transported" (Fig. S4 caption). While For the experiments with SA seed particles, more than 95% particles transported through the lens. The statement is "For all the experiments, more than 95% particles were transported through the lens" (Fig. S6 caption)

AS seed particles were generated by an atomizer (Model 3076, TSI) with AS solution. The generated AS seed particle size distributions were very stable as the geometric mean diameter for AS in all the experiments were  $\sim$ 80 nm. The mass concentration variation is due to the variation in total number concentration, which is somehow difficult to control by injection time. However, this number concentration was not observed to affect our results.

**7. "Volatility Effects:**

One potential issue that should be addressed is the possibility of the SOA being partially volatile. Particles with low f44 (such as hydrocarbon-like organic aerosol or HOA) are known to be the most volatile SOA species (Paciga et al., 2016;Huffman et al., 2009). Depending on how they are operated, the SMPS and DMA/APM/CPC systems could cause evaporative losses. There should be a brief discussion on how this might affect some of the results. How might variations in volatility affect the background (filter) subtraction in these ACSM measurements? Did the background levels change as a function of f44?"

Response: Volatile HOA particles with low  $f_{44}$  values may cause higher RRF. However, during our experiments of low  $f_{44}$ , most oxidation products may remain in the gas phase and we did not observe the background signal of the Q-ACSM varying regularly with  $f_{44}$ , as the background signals were more related with the particle concentration. Hence, we did not expect volatility would affect our experiments.

**8. "Technical Corrections:**

Page 1 Lines 1-2 (Title): It is important to distinguish the term "sensitivity" from the "response factor," here and throughout the manuscript, where the "response factor" was measured and the "sensitivity" was inferred. In addition, "different oxidation states" were not measured and should be replaced in the title with "varying average oxygen content" or similar phrasing. "Oxidation state" should be replaced throughout the manuscript with more appropriate phases, such as "f44" or "average oxygen content" depending on the context.

Response: Throughout the manuscript the term "sensitivity" has been replaced by "RF" or "RRF". As suggested by the reviewer, the term " $f_{44}$ " or "average oxygen content" were used throughout the revised manuscript to replace the "oxidation state".

9. "Page 1 Lines 20-21 (Abstract): This statement is not true as written. Should delete or replace the word "comprehensively."

Response: The sentence has been revised as: "The response factor (RF) of a quadrupole based aerosol chemical speciation monitor (Q-ACSM) for secondary organic aerosols (SOA) has been investigated in this work."

10. "Page 1 Line 21 (Abstract): revise "SOA samples were generated ..." to "SOA was generated ..."

Response: It has been revised as: "SOA were generated under simulated photochemical oxidation conditions in a 4.5 m3 Teflon chamber from three different volatile organic compounds (VOC)".

11. "Page 1 Line 23 (Abstract): The "dozens of ppbv" concentrations of the precursor VOCs used in this study are not typical. Suggest deleting/revising the phrase "atmospheric relevant concentrations" at this location as well as other places in the manuscript."

Response: To avoid confusion, the phrase has been deleted in the manuscript.

12. "Page 2 Lines 24- 25 (Abstract): The sentence starting with "Different SOA oxidation states were achieved by . . ." should be re-written to something along the lines of "For some experiments, varying degrees of average oxygen content in the SOA were obtained by ..."

Response: The sentence has been revised as: "For some experiments, different degrees of average oxygen content of SOA, indicated by the ratio of m/z 44 signal over total organic ion signal intensity ( $f_{44}$ ), were obtained by changing the ratio of the VOC precursors to the oxidants (O3 or OH)."

13. "Page 2 Lines 26-28 (Abstract): Needs to be revised – circular argument about the "exact mass" "used to deduce" the "effective density" and "mass concentration."

Response: The sentence has been revised as: "An aerosol particle mass analyzer (APM) and a scanning mobility particle sizer (SMPS) were used to determine SOA effective density ( $\rho_{eff}$ ) and volume concentrations."

14. "Page 2 Line 29 (Abstract): The sentence starting here needs to be revised. Several parts of it are not clear and not valid. It indicates that "considerable errors" in the SMPS calibration occurred from variations in the effective density, but the 23% difference between the reported effective densities do not correspond to the factor of 3 difference in the measured response factors (from about 0.75 to about 2.15), which appeared to be corrected for varying effective density."

Response: The sentence has been revised as: "Relative response factor (RRF) of the Q-ACSM to organics was determined by dividing Q-ACSM measured mass (the relative ionization efficiency ( $RIE_{org}$ ) and organic collection efficiency ( $CE_{org}$ ) were set to default values of 1.4 and 0.5, respectively) by SMPS-APM deduced mass (amended by aerodynamic lens transmission efficiency ( $E_L$ ) and particle loss due to non-spherical shape ( $E_s$ ))."

15. "Page 2 Line 31 (Abstract): The term "can change substantially" should be replaced with the actual values with uncertainties. The measurements indicate that effective density changes from 1.1 to 1.35, but the accuracy of the method was not demonstrated to interpret the significance of the change."

Response: The sentence has been revised as: "Our results showed that RRF for a specific type of SOA anticorrelated with  $f_{44}$  regardless of the VOC precursors. RRF decreased from 2.16 to 0.81 when  $f_{44}$  increased from 0.057 to 0.191. Meanwhile,  $\rho_{eff}$  increased from 1.09 to 1.34 g cm-3, implying more compact structure for more oxidized SOA."

16. "Page 2 Lines 31-33 (Abstract): Should replace "sensitivity" here and elsewhere with "response factor." This statement and the title of the manuscript are similarly imprecise, because the paper reports changes in the RF as a function of f44, which is a measure of the average oxygen content in the total organic aerosol mass and not the "oxidation state." A problem with this sentence and following statements is that the values for the response factors and the RIE/CE combination used to obtain them were not stated specifically in the abstract."

Response: The terms "sensitivity" and "oxidation state" have been replaced with "response factor" and " $f_{44}$  or oxygen content" throughout the manuscript.

17. "Page 2 Line 33-34 (Abstract): This statement should be revised because it implies that RIE and/or CE should be decreased relative to the values used (1.4 and 0.5, respectively) as the oxygen content increases."

Response: It has been revised as: "The anti-correlation between RRF and  $f_{44}$  might be due to the decreased RIEorg or CEorg for more oxidized SOA."

18. "Page 2 Lines 34-37 (Abstract): Sentence has grammatical and technical errors and needs to be rewritten. There is no context for changes in sulfate signals."

Response: The sentence has been revised as: "To further explore the actual cause, ammonium sulfate (AS) seed particles were injected into the chamber before SOA were produced. After the 80 nm AS particles were fully coated with SOA (coating thickness > 18-25 nm), CE of the aerosol changed from  $CE_{AS}$  (0.285  $\pm$  0.067) to  $CE_{org}$  of the SOA shell."

19. "Page 2 Lines 37-39 (Abstract): More grammatical and technical errors in this sentence. There is no evidence in this paper of transforming SOA "gradually" from a liquid state to a solid (or glassy) state. This work does show that the CE values vary depending on the average oxygen content as indicated by f44, and the statement should be revised accordingly. However, the CE values for the solid (or glassy) state are reported here as " $0.2 \leftarrow 0.5$ ," whereas the measured values in the paper show that CE is above 0.3. Knowing the CE values precisely is an important part of reducing the uncertainties in the response factors, since CEs of 0.2, 0.3, and 0.5 represent changes in RF by factors of 5, 3, and 2, respectively."

Response: The sentence has been revised as: "The experiment showed that  $CE_{org}$  decreased from 0.882 to 0.313-0.475 when  $f_{44}$  increased from 0.127 to 0.209, most likely caused by a phase change of SOA from liquid to solid/glassy."

20. "Page 2 Lines 39-41 (Abstract): Again, there is no evidence in these observations of SOA transformations occurring as stated (here from hydrocarbon-like OA to more oxygenated OA). Furthermore, the changes in CE as a function of f44 are consistent with the changes in the response factor without a varying RIE. Specifically, if CE=1 instead of 0.5 for the lowest f44 values, the response factor would decrease from 2.15 to 1.1 and if CE=0.3 instead of 0.5 for the highest f44, the response factor would increase from 0.75 to 1.25. Both revised values for the response factors (1.1 and 1.25) are well within the uncertainties of ACSM measurements without changing the RIE significantly from 1.4."

Response: The sentence has been revised as: "The large uncertainties of  $CE_{org}$  was principally caused by the uncertainty in SMPS measurement. The variation of  $CE_{org}$  with  $f_{44}$  could explain a large fraction of the observed decrease in RRF, while the influence of  $RIE_{org}$  cannot be excluded."

21. "Page 2 Lines 41-44 (Abstract): More grammatical and technical errors in this sentence. It appears that the term "RIE" was used here where the term "response factor" would be more accurate. The claims of "underestimation" and "overestimation" need to be quantified to be meaningful. The variations in the response factors appear to be consistent with the previously reported changes in CE as a function of average oxygen content for chamber-generated SOA (Docherty et al., 2013) and the 38% uncertainty in AMS organic mass concentration (Bahreini et al., 2009)."

Response: The statements containing OOA or HOA have been removed. The sentence has been revised as: "The trends in RRF and  $CE_{org}$  for Q-ACSM were in accordance with those done with the high-resolution AMS, demonstrating the capability of the low-resolution Q-ACSM in doing SOA chamber studies and the necessity to calibrate RF in laboratory experiments."

**22. "Other Corrections:**

There are other numerous grammatical errors in the rest of the manuscript, not specified here. Instead,

noticeable technical errors are highlighted below:

Page 3 Line 57: add the word "air" or "aerosols"

Page 4 Line 92: The citation for Zhou et al., 2016 does not seem appropriate for the "widely used . . . around the world" context.

Page 5 Lines 119-121: There are many examples of ambient datasets showing the AMS mass concentrations agree well with other observations (Jimenez et al., 2016). This should be mentioned here.

Page 6 Line 130: This is a good place to cite the recent ACSM characterization paper (Xu et al., 2017).

Page 7 Line 169: mention the inner diameter instead of the "OD"

Response: The manuscript has been revised by a native English speaker for grammatical errors. All errors above have been revised accordingly.

23. "Page 9 Lines 207-208: f44 could also be artificially low when particles initially form if the ion signals from large m/z are not properly included in the total organic mass (and dividing the high S/N peak of m/z 44 with an artificially lower total organic signal). This issue could be checked for the first three data points in Figure 2 by comparing relatively high m/z ion peak intensities in their spectra with spectra from higher aerosol mass concentrations."

Response: The sentence has been revised as: "The initial high values of  $f_{44}$  could partially due to  $f_{44}$  calculation bias in low concentration (0-5 µg m-3) or the fact that initially formed SOA particles were highly oxidized because of gas to particle partitioning (Shilling et al., 2009)." (Lines 201-203)

24. "Page 12 Lines 284-285: Does "0.28 in this work" refer to the Matthew et al. study (Matthew et al., 2008) or the work for the manuscript? The Matthew et al. paper states a lower CE with uncertainties assigned to it (24+/-3%), making the 0.28 value outside the uncertainty band."

Response: The value of  $0.285 \pm 0.067$  was measured during the AS seed experiments by comparing the AS signals with SMPS measurements without organic coating. The CE was slightly higher than that reported by Matthew et al. (2008) (0.24  $\pm$  0.03), which may be caused by the uncertainties of SMPS-APM measurement or the bias of RIES04 from the default RIES04 (1.15).

25. "Page 12 Lines 286-290: The Matthew et al. work showed that the number of sulfate ions per particle did not change with additional water or organic coating, so it should be cited here that the RIE for sulfate does not change during the coating experiment. This point is important because any changes in the sulfate mass loadings during the course of the experiment are not because the RIE for sulfate changed."

Response: This sentence has been revised as: "A default  $RIE_{SO4}$  (1.15) was used here as suggested by previous studies (Canagaratna et al., 2007; Ng et al., 2011; Petit et al., 2015). As  $RIE_{SO4}$  would not change during the organic coating experiments, any changes in sulfate signal was due to variations in  $CE_{AS}$  (Matthew et al., 2008)." (Lines 239-242)

26. "Page 12 Line 292 and following: the "coating thickness" should be calculated using the size distribution data and included in Table 2. The size distribution data would also indicate other changes, like

newly formed SOA particles for the "extreme condition" experiment."

Response: The coating thickness has been calculated and included in Table 2. The size distributions were also included in the supplementary information (Figs. S4 and S5): "For most of the experiments, all condensable organic vapor seemed to condense directly onto seed particles instead of forming new particles. While in one experiment (Exp. 22), there were small amounts of new particles formed, which was indicated by a smaller mode on the size distribution (Fig. S6) and further confirmed by a sudden increase in total particle number concentration. However, the newly formed particles only contributed slightly to the total particulate mass for their much smaller sizes. In addition, to avoid the influence of new particle formation on  $CE_{org}$  or  $RIE_{org}$  deduction, only  $PM_{ACSM}$  and  $PM_{SMPS}$  associated with AS signals were used to deduce the actual  $CE_{org}$  in our experiments." (Lines 271-277)

27. "Page 13 Lines 302-305: Both sentences appear to say that the light blue points are AS cores with an organic shell, but the connecting word is "however". Need to revise to clarify."

Response: Both sentences have been removed from the manuscript. The color represented the coating thickness in the current manuscript, instead of  $Org/SO_4$  ratio in the original manuscript. SOA layer of 18-25 nm can fully cover the AS seed particles.

28. "Page 13 Lines 314-315: "SOA may transit from liquid phase . . ." Need to revise/check grammar."

Response: This statement has been moved to lines 340-342 and revised as: "Decreasing  $CE_{org}$  with  $f_{44}$  might be explained by the phase transition from liquid-state into more solid-like/glassy state of SOA particles consisting of more oxidized organic compounds that normally assume a higher  $\rho_{eff}$ ."

29. "Page 13 Lines 315-316: Higher oxidation state and higher effective density are not necessarily linked to changing phase states if the composition is also changing. Need to revise."

Response: The statement has been revised as: "Decreasing  $CE_{org}$  with  $f_{44}$  might be explained by the phase transition from liquid-state into more solid-like/glassy state of SOA particles consisting of more oxidized organic compounds that normally assume a higher  $\rho_{eff}$ ."

30. "Page 14 Line 323: To complete the section on CE, studies on CE of HOA and aged ambient OA could be added/mentioned, such as (Slowik et al., 2004;Zhang et al., 2005;Middlebrook et al., 2012)."

Response: Former studies on  $CE_{org}$  was discussed briefly in the introduction as: "Although organics quantified using a  $CE_{org}$  of 0.5 has been found correlating well with independent organic carbon (OC) or VOC measurements in most field works (Allan et al., 2004; de Gouw et al., 2005; Takegawa et al., 2005; Venkatachari et al., 2006; Middlebrook et al., 2012), studies have shown that  $CE_{org}$  can vary substantially for various chamber generated organic particles (Bahreini et al., 2005; Docherty et al., 2013)." (Lines 96-100).

Detailed inter-comparison between our results and former  $CE_{org}$  calibrations has also been discussed in detail in lines 303-317 as "In previous studies conducted by AMS, CE was demonstrated to be variable for particles with different chemical composition, phase state and under different RH, with  $CE_{org}$  ranging from less than 0.15 to ~1 (Alfarra, 2004; Matthew et al., 2008; Docherty et al., 2013; Robinson et al., 2015). However, during field studies AMS measurement results based on default RIE = 1.14 and CE = 0.5 appeared to be in reasonable agreement with other aerosol measurement techniques (Slowik et

al., 2004; Zhang et al., 2005; Middlebrook et al., 2012). CE for biogenic SOA has been reported to be close to 1 based on both chamber experiments (Kiendler-Scharr et al., 2009) and field measurements conducted in Amazon, where aerosols were dominated by liquid SOA (Chen et al., 2009; Allan et al., 2014). However, it has been proposed in theoretical, chamber, and field studies that besides liquid state, organic particles can exist in semi-solid or solid state (Vaden et al., 2010; Virtanen et al., 2010; Shiraiwa et al., 2011), which might lead to lower CEorg. For example, CEorg of liquid squalane (CE = 1) decreased after SOA condensed on the particle(Robinson et al., 2015). Most recent CEorg calibration with light-scattering single-particle (LSSP) module also suggested that CEorg (0.25-0.4) could be much less than 1 (Robinson et al., 2017). An inverse relationship between CEorg and  $f_{44}/f_{57}$  ratio in chamber SOA generation experiments (Docherty et al., 2013) was very similar to our results. However, our results could not be quantitatively compared with the previous ones because the Q-ACSM might report comparably higher  $f_{44}$  than the AMS (Fröhlich et al., 2015)."

31. "Page 14 Lines 336-337: While this statement may be true, the section on CE as a function of f44 seems to indicate that the large changes in RF are mostly due to changes in CE. Once the CE is properly taken into account for the RF, RIE for the organics studied here does not seem to be varying by as much as the studies cited in the previous sentence. Consider adding more statements at the end of this section."

Response: After conducting additional calibrations, the uncertainty of measured  $CE_{org}$  was better constrained. Accordingly, we revised the statement as: "With our experiments, we still cannot rule out the contribution of RIEorg to the overall RRF trend even though the variation of CEorg with  $f_{44}$  could largely explain the observed RFF changes." (Lines 297-298)

32. "Table 1: Add errors, mainly to PM, density, f44, and ACSM/SMPS columns."

Response: Errors have been inserted into Table 1.

33. "Table 2: Digits listed for ammonium sulfate, Org, f44, and CE are not significant. Revise and add errors to these columns. Add one more column (with errors) for the calculated thickness of organic coating, based on size distribution data and derived effective density."

Response: Error bars and coating thickness have been inserted into Table 2.

34. "Figure 1: The air in the chamber (how generated and introduced) is not shown. Is the chamber filled like a balloon or is it a flowing reactor? How is air pumped out of the chamber? Is air added to the chamber while the instruments are sampling? It is unclear how the ammonium sulfate seeds are dried and added to the chamber. The figure caption needs a few more details explaining a typical experiment."

Response: Air inside the chamber was generated with a zero-air generator (Aadco 737, USA). The chamber was basically a flexible Teflon bag. When inflated the chamber was assuming a cubic shape with a total volume of 4.5 m3 and the inside pressure was equal to the ambient pressure. During the experiment, the chamber zero-air inlet was sealed and air was sucked out by the instruments. Total of  $\sim 0.7 \text{ m}^3$  air was consumed for each experiment. At the end, the chamber collapsed a little but would still assume a cubic shape and hold the same pressure as the room pressure. The description of the chamber operation was given in lines 134-139.

35. "Figure 2: This is the only figure with number distributions. They should be converted into mass

distributions, the effect of the ACSM lens transmission should be shown (calculated transmission into the ACSM), and additional mass distributions should be added to supplementary information. Should add the "start time" for the experiment, along with an arrow indicating when the system was considered to be in "relative steady-state." Add that the AMS mass concentrations were calculated using the default RIE=1.4 and CE=0.5. Also include in supplementary information how the system was calibrated for the ionization efficiency of ammonium nitrate and how accurately the ammonium nitrate effective density was measured using this method."

Response: Average number size distributions during "relative stable state" in all the experiments and the measured lens transmission efficiency have been given in the supplementary information (Figs. S4 and S6). Arrows were added to indicate the start time and the "relative stable state". The default values of RIE = 1.4 and CE = 0.5 were added on Fig. 2. Details of Q-ACSM calibration with ammonium nitrate have been given in the supplementary information (section 1-2).

36. "Figure 3: Should state in the caption what value of effective density was applied to the SMPS mass concentrations. Also add the default RIE=1.4 and CE=0.5 was used for the ACSM mass concentrations. Note that a bias of 22% (slope here) is within the +/-38% uncertainties of the AMS measurements (Bahreini et al., 2009)."

Response: The original Fig. 3 has been removed. New Fig. 3 includes both the correlation between RRF vs  $f_{44}$  and effective density vs  $f_{44}$ . The error bars in new Fig. 3 were based on all data points obtained during all experiments listed in Table 1. The ±38% measurement uncertainty reported by Bahreini et al. (2009) has been indicated by the shaded area in new Fig. 3a.

37. "Figure 4: This could be moved to the supplementary information, since it is not directly relevant to the discussion."

Response: Figure 4 has been moved into the supplementary information (Fig. S7).

38, "Figure 5: The conclusions of this paper about the response factor (and effective density) varying with f44 rely mainly on a few points affecting the trends (linear relationships). Errors need to be propagated to provide strength to this observation. There is a cluster of data points near f44=0.14, that have average effective densities of 1.25+/-0.05 and response factors of 1.25+/-0.3. These values are within the combined uncertainties. Toluene is listed twice in the effective density (bottom plot) legend – green triangles should be isoprene. Add to the caption the default RIE=1.4 and CE=0.5 were used for the calculations of the ACSM mass concentrations."

Response: Figure 3 is the original Fig. 5. Figure 3a shows that RRF for the chamber-generated SOA decreased from 2.16 to 0.81 with  $f_{44}$  increasing from 0.057 to 0.191. Fig. 3b shows that  $\rho_{eff}$  increased from 1.09 to 1.34 g cm-3 with increasing  $f_{44}$ . After comprehensively evaluating the uncertainties, the error bars in both Fig. 3a and Fig. 3b increased a lot. However, RRF for the two isoprene derived SOA points were still outside of the range of Q-ASCM measurement uncertainty (the gray area). The anti-correlation between RRF and  $f_{44}$  was still very clear. Moreover, the overall positive deviations of RRF could possibly be explained by a higher than default RIEorg value (1.6±0.5), as suggested by Xu et al. (2018). (Lines 327-330)

39. "Figure 6: The experiment shown here is described in the text but does not appear to be listed in Table 2. As a second panel, the mass distributions should be shown. Why are the sulfate concentrations not "zero"

before the seeds were added? What VOC precursor was added? When were the lights turned on? What are the various organic coating thicknesses estimated as the organic mass increases for a relatively constant sulfate mass? This experiment possibly indicates that the CE changed slightly when the SOA began to be detected, then did not change (within some uncertainty band) as the sulfate particles become coated with more organic material. Unfortunately, the sulfate mass concentrations do not appear to be stable enough to state this conclusively. Were there other experiments where the sulfate concentrations were more stable be- fore adding the VOC and turning on the lights? See other comments on the discussion of this figure on the lack of stability in the sulfate concentration during the course of the experiment and changes to the mass distributions potentially affecting lens transmission."

Response: All the experiments were numbered (#1 to #22) in the revised paper, the two typical experiments shown in details were #1 and #19. Figure 4 is the original Fig. 6. Another panel of aerosol 3-D size distribution time series plot was inserted into Fig. 4a. The details about the experiment conditions and the default CE and RIE values have been inserted into Fig. 4b. Arrows were added to show the relative stable states that were used for average. The experiment started when AS was first started to injected into the chamber. Arrow 1 indicated the AS particle concentration became stabled after stopping injection. Arrow 2 indicates when SOA started to form. The period between Arrows 3 and 4 was when the SOA coated AS particles reached "relative stable state" and was used for data averaging. In all the experiments, Sulfate concentrations had more or less variations, possibly due to the use of a relatively low sampling interval ( $\sim$ 8 min). As inspired by the reviewer, we found that 18-25 nm organic shell can fully cover the 80 nm AS seed particles. As the situation in this experiment (Exp. 19), "Two hours after the experiment started, the Q-ACSM measured organic concentration was still increasing but the measured sulfate stopped increasing and kept constant during the rest of the experiment. By the time when sulfate signal stopped increasing,  $d_{\sigma}$ increased to 115 nm, the average coating thickness (growth in particle radius) was about 18 nm assuming uniform coating. This indicated that 18 nm SOA shell was already enough to cover the 80 nm seed particles in this experiment." (Lines 253-257)

---

## Author Comment (AC3) · 1 Oct 2018

**We are very grateful for the reviewer's comments. A point-to-point response to reviewers' comments is provided below, and the manuscript has been revised accordingly. The line numbers in the response refer to those in the revised version of the manuscript.**

**Reviewer #3:** *"In this work, Li et al. have investigated the response factor of a Q-ACSM to secondary organic aerosols generated under different oxidation states and from various VOC precursors. The subject of this work is certainly within the scope of AMT and the results of this work are of practical interests to the ACSM/AMS community. The advantage of this work is that the aerosol standards were generated in situ and their oxidation states can be pre-defined. The experiments were designed with some novelty and were carefully executed. The manuscript was clear written. However, the manuscript still can be improved. A native English speaker is highly recommended to proofread the manuscript before it can be published. Overall, I think this work can be a valuable contribution to the AMS/ACSM community. Therefore, I would recommend this manuscript for publication after the authors address the following comments:"*

Response: As suggested by the reviewer, we have conducted additional calibrations (supplementary information section 1 and 2) to fully address the uncertainties of our measurement results. The discussion of this research is also limited to chamber-generated SOA only. The manuscript has been thoroughly proofread by a native English speaker. Following please find our specific responses.

**Specific comments:**
1. *"The authors shall pay more attention to the special terms and jargons used by the ACSM/AMS community. For example, the RF is not clearly defined here. The sensitivity of the ACSM/AMS has been investigated extensively in previous works. The results of this work seem to be in line with previous results. But the authors need to reorganize the manuscript to make the results of this work inter-comparable with other similar works, i.e., the results of this work can be evaluated under the same established framework."*

Response: The term "sensitivity" has been redefined as "response factor (RF)" and the Q-ACSM performance in measurements of organics has been defined as "relative response factor (RRF)" with respective to mass concentration determined by a SMPS-APM system (Lines 177-181). We have added another section of *3.4 Comparison with AMS Calibrations* (Lines 299-332) to inter-compare our results with previous works. One of the strength of our calibration comparing to the former calibrations is that we related the variations of $CE_{org}$ to possible SOA phase changes (due to oxygen content changes) indicated by the effective density measurements. Another strength of this work is that we demonstrated that a relatively low resolution (for both response time and m/z) Q-ACSM can be used to study chamber-generated SOA. Overall, our $CE_{org}$ calibration results were in accordance with former researches done with AMS.

2. *"For Fig. 8, how is this figure generated and how valid is this result? The author may want to provide more discussion and support information for validation."*

Response: Since HOA was not studied in this work, we limited our study to chamber-generated SOA only. To avoid possible overstatement, we have removed Fig. 8 from the manuscript. More information on instrument calibrations for Q-ACSM, SMPS, and APM, average number and mass distributions in all of the experiments were given in the supplementary information. The accuracies and uncertainties of the experiments were recalculated in a more careful way (e.g. we performed extra experiments to calibrate the ACSM lens transmission efficiency ($E_L$) and the calibrated $E_L$ was used to modify $PM_{SMPS}$; we better evaluated and propagated the uncertainties in SMPS and APM measurements; we calculated the SOA coating thickness for the AS seed particles; we redefined "relative stable states" in each experiment) to make the results more reliable and precise. All the efforts above were meant to constrain the measurements uncertainty of a Q-ACSM. In the revised paper, we concluded that the variation of $CE_{org}$ with $f_{44}$ could explain a large fraction of the observed decrease in RRF, while the influence of $RIE_{org}$ cannot be excluded.

The trends in RRF and CE$_{org}$ for Q-ACSM were in accordance with those done with the high-resolution AMS, demonstrating the capability of the low-resolution Q-ACSM in doing SOA laboratory studies and the necessity to calibrate RF when conducting laboratory SOA experiments (Lines 36-41).

---

## Author Comment (AC4) · 1 Oct 2018

**We are very grateful for the reviewer's comments. A point-to-point response to reviewers' comments is provided below, and the manuscript has been revised accordingly. The line numbers in the response refer to those in the revised version of the manuscript.**

**Reviewer #4:** *"This manuscript describes measurements of the sensitivity of an ACSM to OA for three oxidant/VOC SOA systems produced in a mid-sized Teflon chamber using an ACSM, SMPS, DMA-APMA. Relationships of sensitivity and effective density to f44 are explored. Experiments were conducted in attempt to separate effects on sensitivity due to collection efficiency (CE) and relative ionization efficiency (RIE) of OA. Conclusions are made on trends of CE, RIE, phase state, HOA vs OOA, effective density, and oxidation state.*
*The manuscript is generally poorly written and makes a range of conclusions using flawed logic and speculation based on incomplete experimental evidence and lack of context from the existing literature. Due to some critical flaws (discussed below), I do not believe the manuscript is publishable in its present form. If it was published, it would "pollute" the literature with trends that are not well supported and may be due to experimental factors completely unrelated to those that are discussed."*

Response: As suggested by the reviewer, we have thoroughly revised the manuscript, which has been proofread by a native English speaker. Since HOA was not specifically studied in this work, we have limited our conclusions on chamber-generated SOA only to avoid possible overstatements. Therefore, Figure 8 has been removed from the manuscript. Additional discussion of the inter-comparison between this work and previous AMS calibrations have been added in section 3.4. Moreover, additional laboratory calibrations of instruments used in this work have been conducted to fully address the uncertainties associated with our measurements.

All the efforts above were meant to constrain the measurements uncertainty of a Q-ACSM. Our results strongly indicate that measurements of OA with various oxygen contents by a Q-ACSM using constant conversion factors may induce significant errors in mass concentration measurements for laboratory generated SOA. The results of this work appeared to be consistent with the results of previous work using different experimental settings and reconfirmed the reliability of the AMS/ACSM techniques. Our work demonstrated the capability of the low-resolution Q-ACSM in doing SOA chamber studies and the necessity to calibrate RF in laboratory experiments. Although the results of this work may not be directly applicable to ambient Q-ACSM measurements, it is reasonable to believe that the RRF of a Q-ACSM could be highly variable under different ambient conditions and thus similar comprehensive calibrations of Q-ACSM for complicated ambient conditions is critically needed in the future work.

*"With substantial additional work (including additional experimental evidence, analysis, and calibrations) and documentation, greatly scaled back conclusions, more thorough context, and, the dataset may eventually be publishable. Even in that case, the authors would need to present a convincing case of what parameters they are truly quantifying, including quantifying their uncertainties, and why the results are useful and better inform the analytical study of OA in the laboratory. In its current form, the manuscript is misleading and does not advance findings that are adequately constrained, new, or useful to the literature. Therefore, I strongly recommend that the paper is rejected, and the authors are encouraged to do the additional work and resubmit a much improved paper at a later time. See details below."*

Response: As suggested by the reviewer, we have revised the manuscript completely. The major revisions of the manuscript include:

1.  The special terms (e.g., RF, CE, RIE) were precisely defined in the "introduction" and their mathematical correlations were also clearly stated. The term "sensitivity" was replaced with "relative response factor (RRF)" and its calculation method was given in the new section *2.5 Calculation and Uncertainty Evaluation*.
2.  We performed additional experiments to calibrate the ACSM lens transmission efficiency ($E_L$) and the specifically calibrated $E_L$ was used in $PM_{SMPS}$ calculations; Both SMPS and APM were calibrated

before the experiment and the results have been used to precisely evaluate their measurement uncertainties, which were used to constrain the possible errors of measured CE, RIE, and RRF. Therefore, the overall measurement uncertainties of this research were well defined.

3. More information on instrument calibrations, uncertainty evaluations were given in the supplementary information. Average number and mass distributions in all of the experiments were also plotted in the supplementary information (Figs. S4-6).

4. The manuscript was reorganized and fully revised. Inter-comparisons between our experiments and previous calibration work using AMS were added in a new section *3.4 Comparison with AMS Calibrations*.

5. The SOA coating thickness for the AS seed particles were calculated (18-25 nm), which was found to be enough to fully cover the 70 - 80 nm (in mobility diameter) AS seed particles. AS seeds were fully covered with SOA in most of our experiments. This support our claim of possible morphology change during AS seed experiments.

6. Although $RIE_{org}$ was not directly measured in our experiments, it appeared that the variation of $CE_{org}$ with $f_{44}$ could largely explain the observed anti-correlation between RRF and $f_{44}$, indicating that $RIE_{org}$ may not vary significantly with $f_{44}$ within the studied SOA range.

7. Figure 8 is removed from the manuscript to limit our discussion within the experiment setting only.

**Specific comments:**
1. *"The paper really only quantifies an empirically-defined response factor, which is the ratio of the mass concentrations reported by the ACSM and calculated from the SMPS volume and the measured particle density. As thoroughly documented in the AMS literature, that response factor can have trends due to several effects:*
*RF = (RIEreal / RIEassumed) * (Eb,real / Eb,assumed) * (EL,real / EL,assumed) * (ES,real / ES,assumed)*
*\* (SMPScalc \* SMPSvolreal) \* (ρreal / ρassumed)*
*where CE = Eb \* EL \* ES shows the 3 components of CE, due to particle bounce in the vaporizer, incomplete transmission by the aerodynamic lens, and non-spherical particle effects, respectively as defined by Huffman et al. (2005). It is also clear that errors in the estimation of particle volume from the SMPS number distribution and/or the density will also appear in RF.*

Response: We agree with the reviewer that many factors (e.g., $E_L$, $E_S$, $E_b$, SMPS size-distribution, and $\rho_{eff}$) would affect the response factor of the Q-ACSM. Typically, the response factor of the Q-ACSM is defined as: (line 84)

$$RF_{org} = IE_{NO3} \cdot RIE_{org} \cdot CE_{org} \qquad (1)$$

In our study, a term "relative response factor (RRF)" was defined to express how Q-ACSM measured mass concentrations using default $RIE_{org} = 1.4$ and $CE_{org} = 0.5$ deviate from the SMPS-APM measured mass concentration, where RIE, $E_b$, $E_L$, and $\rho_{eff}$ were all considered in the calculation: (line 182)

$$RRF = \frac{PM_{ACSM}(assuming\ RIE=1.4,CE=0.5)}{PM_{SMPS}(amended\ by\ \rho_{eff},Es,E_L)} = \frac{RIE_{org} \cdot CE_{org}}{1.4 \cdot 0.5} \qquad (2)$$

The detailed calculation procedures have been given in section 2.4 (Lines 177-195).

2. *"When one is trying to characterize a widely used instrument and publishing the observed results as if they are representative of all instruments of the same type, great care needs to be taken to examine each of these terms and their uncertainties. The present paper has some speculative interpretations of RF based on RIE and Eb, even though it completely ignores all the other effects! My best guess is that EL explains a substantial fraction, and potentially most of the trend in RF, as when the particles grow in the SOA experiments, the upper end of the distribution may fall outside of the ACSM lens transmission, thus creating*

*an experiment and time-dependent trend on RF. It is possible that some of the other factors that are ignored in this paper may also contribute to the observed trend. For this reason, the paper is not publishable."*

Response: The $E_L$ of the aerodynamic lens system of the Q-ACSM was measured and applied to the particle size distributions of all the experiments. For the experiments without SA seed particles, a small portion of smaller particles (<100 nm) were lost in the lens (Fig. S4). For the experiments with SA seed particles, more than 95% particles were transported through the lens (Fig. S6).

[Figure]

**Figure S4**. Average number size distributions measured with the SMPS as a function of $d_{va}$ (nm) during the relative stable state for experiments using (a) α-pinene (b) toluene, and (c) isoprene, respectively. $d_m$ was converted to $d_{va}$ via $d_{va} = d_m \cdot \rho_{eff}/\rho_0$. Gray lines represent the Q-ACSM aerodynamic transmission efficiency ($E_L$) as a function of $d_{va}$ (nm). For most of the experiments, more than 94% of the total particle mass was transported into the Q-ACSM through aerodynamic lens. While for three of them (Exp. 4, 5, and 7), a portion of particles were too small to pass the lens and only 87-89% of the total mass was transported.

[Figure]

**Figure S6.** Average number size distributions ($d_{va}$) measured before (dashed lines) and after (solid lines) SOA (panel a: α-pinene; panel b: toluene) was coated. Gray lines represent the Q-ACSM $E_L$ as a function of $d_{va}$ (nm). Gray lines represent the Q-ACSM aerodynamic transmission efficiency ($E_L$) as a function of $d_{va}$ (nm). For all the experiments, more than 95% particles were transported through the lens.

3. *"The authors need to go back to the lab and present multiple additional pieces of information, including:*

*1) A detailed characterization of EL vs. particle size for their own instrument. It is well-know that EL varies for different instruments (and sometimes in time for a given instrument), especially at the upper end of the particle size. However, Figure 2 suggests that losses at the small end may also be a problem. These are challenging experiments, but they are doable. If the authors cannot accomplish this, they should not*

*resubmit this paper elsewhere, as the results would be ambiguous, and it would be very misleading to report them as trends in Eb and RIEorg while ignoring EL. Using the transmission curves from the literature is not acceptable, given substantial variability in EL across instruments. See below for literature references.*

Response: We have fully characterized the $E_L$ as a function of particle size in our laboratory. (SI Lines 32-51)

"The lens transmission efficient ($E_L$) of the Q-ACSM was calibrated using the same setup as in Figure S1. The only difference was that the DMA was set to select a variety of particles with diameters ranging from 200 nm to 600 nm instead of a fixed single diameter of 300 nm. To minimize multiple charge problem for larger particles, we conducted $E_L$ calibration under two sets of comparably low $NH_4NO_3$ solution concentrations (~2 mM and ~0.5 mM). However, for the set of experiment with extremely low $NH_4NO_3$ concentration (~0.5 mM), the Q-ACSM sensitivity was not high enough to detect $NH_4NO_3$ particles, which therefore was not shown here. Consequently, the calibration result of the lens transmission efficiency was shown in Figure S2.

[Figure]

**Figure S2.** Lens transmission efficiency as a function of particle vacuum aerodynamic diameter ($d_{va}$).

Lens transmission efficiencies for 300-600 nm particle were nearly unity, which was in reasonable accordance with some previous studies (Jayne et al., 2000; Knote et al., 2011; Hu et al., 2018), but the diameter range with a unity lens transmission efficiency was much broader than that reported by Liu et al. (2007). The blue line in Figure S2 was used to modify size distribution in our experiments (i.e., 0% transmission for $d_{va}$ below 40 nm; linear increase in transmission vs log($d_{va}$), from 0% to 100% at $d_{va}$ = 100 nm; 100% transmission efficiency from $d_{va}$ = 100 nm up to $d_{va}$ = 580 nm; linear decrease in transmission vs log($d_{va}$) from 100% at 580 nm to 0% at 1200 nm). For $d_p$ below 300 nm, we used the results reported by Knote et al. (2011) indicated by the red dashed line in Figure S2. About 10% of uncertainty (shaded area) was applied to the $E_L$ used in this study as indicated by the blue curve in Figure S2."

*2) Using the measured SMPS volume distributions, the measured density, and the measured ACSM transmission vs particle size (in dva space, see DeCarlo et al., 2004), the authors can "trim" the SMPS volume to correspond to the volume actually sampled by the ACSM.*

Response: The SMPS-APM measured mass concentration was obtained as following:
$E_L$ as a function of particle diameter in the aerodynamic lens was applied to SMPS-APM measured mass concentration. Es was assumed to be 1, the possible non-spherical situation was considered by adding an error of 5% for Es (Lines 182-195):

$$"PM_{SMPS} = V_{SMPS} \cdot \rho_{eff} \cdot E_L \cdot E_s = \sum_{i=1}^{n} \{\frac{\pi}{6} d_{m,i}^3 \cdot dN_i \cdot \rho_{eff,i} \cdot E_{L,i} \cdot E_{s,i}\} \quad (3)$$

Eq. (3) is used to calculate $PM_{SMPS}$, where $V_{SMPS}$ is the volume concentration calculated with SMPS measured size distribution; n is the total number of bins for SMPS measurements; $d_{m,i}$ and $dN_i$ are

respectively the geometric mean diameter and the number of counts in size bin i; $E_{L,i}$ and $E_{S,i}$ are the aerodynamic lens transmission efficiency and shape factor for particles of $d_{m,i}$. After applying $E_{L,i}$ and $E_{S,i}$, any deviation in $CE_{org}$ should be caused only by particle bounce on the vaporizer surface ($E_b$). The measured $E_L$ (Fig. S2) is in reasonable accordance with previous studies (Jayne et al., 2000; Knote et al., 2011; Hu et al., 2017), while larger than other reports (Liu et al., 2007) as for 300 nm~1000 nm ($d_{va}$) particles. $E_S$ was assumed to be 1±5%. $d_m$ and $d_{va}$ were convertible via $d_{va} = d_m \cdot \rho_{eff}/\rho_0$ for a reference density $\rho_0$ of 1000 kg m$^{-3}$ (Jayne et al., 2000; DeCarlo et al., 2004). The uncertainties of $d_{m,i}$ (±4%), $dN_i$ (±10% for 20-200 nm and ±20% for 200-800 nm particles) (Wiedensohler et al., 2012; Wiedensohler et al., 2018), $\rho_{eff}$(±3%), $E_L$ (±10%) and $E_s$ (±5%) were well propagated to the calculation. As a result, the relative uncertainty of $PM_{SMPS}$ was between 19.4~26.0%, depending on the actual size distribution. Details of uncertainty evaluation can be found in SI."

*3) It does not seem acceptable to present density measurements at only one size, especially if it is the peak of the number distribution. A few points across the distribution, and importantly the peak of the volume distribution, should be included for each system.*

Response: In this work, we consider the chamber-generated SOA were mostly unimodal distributed (as shown in Fig. S4), especially for the AS seeded experiments, when only a single mode of AS particles was present in the chamber. The number-size distribution measured by the SMPS was used to deduce the aerosol total volume. The peak size of the distribution was considered the most representative size of the SOA population. The APM was used to measure the true mass of a certain size (mobility diameter) of particle and thus APM is usually used with a DMA for size selection first (McMurry et al., 2002). The mass distribution obtained by the APM was determined by the resolution of the APM, i.e., its rod rotational speed and its physical radius. The magnitude of the APM peak was only used to determine the exact mass of a certain size particle not the actual number concentration of that size of particle in the chamber. Accordingly, the effective density of the SOA population was calculated, which was then used to calculate the total mass of the SOA. As SOA continued to grow as the experiment proceeded, it was important to track the SOA changes with high time-resolution. Typically, one SMPS-APM scan can be completed within 10 min, which cannot be achieved by the traditional filter-based aerosol measurement technique. We have tested the APM with a certain size of PSL, slight change in peak selection of the DMA setting by a few nm, the mass measured by the APM was not significantly affected but the magnitude of the peak was substantially reduced. Therefore, we believe slightly off in the peak size selection would not affect the APM mass measurement. In fact, our results showed that the average effective density from each experiment did not show any dependence on the particle size. It has been stated in the revised paper that "Before applying this measured $\rho_{eff}$ to the whole size distribution, $\rho_{eff}$ of particles in all sizes in the same moment were assumed to be the same." (Lines 168-170).

*4) Evidence of good alignment of the aerodynamic lens in the ACSM should be presented. The particle beam width depends on particle size and shape, and if the lens is not well-aligned, then there can be particle losses that are strongly size-dependent, giving rise to another source of variation of EL that would show up in RF.*

Response: The lens alignment was done right before the RIE calibration using $NH_4NO_3$ particles (Line 155). The setup of lens alignment was the same as that of $NH_4NO_3$ RIE calibration, except that CPC was not used. 300 nm $NH_4NO_3$ dry particles were atomized, dried, and size-selected by DMA. The $NO_3$ signal intensity was monitored with the "Lens alignment tool" while the particle lens was moved across one of the planes (horizontal or vertical axis) to the limit where no $NO_3$ signal was seen. The center position of the four edges was assigned for the lens plate.

*5) Evidence of calibration of the SMPS size (using PSLs) and especially, volume concentration, needs to be presented. It is typical in field studies when multiple SMPSs are present (all operated by reputable*

*groups) that the reported volume concentrations range a factor of 4-5. That's the typical accuracy that can be expected for an SMPS that is running on a lab and that has not been thoroughly calibrated and quality-checked. Even after careful work, discrepancies often remain, see for example Figure 8 in Wiedensohler et al. (2012). There are a number of papers that specify the checks that one has to do in order to ensure quantitative SMPS measurements, and it was especially glaring that this topic was not even mentioned in the present paper. See for example Wiedensohler et al. (2018) and Stolzenburg and McMurry (2018). An intercomparison to other instruments such as additional SMPSs and OPCs (for example UHSAS) would be useful to gain confidence in the measurements. Evidence of quantitative response of the CPC, ideally by comparison to an electrometer, but otherwise by comparison to several other CPCs (across the size range) is also needed.*

*6) Evidence of accurate density measurements using known materials, and across the particle size range contributing to the measurements shown, needs to be presented.*

Response: The SMPS-APM system was fully calibrated before the experiments. Detailed operation procedures have been given in lines 170-175:

"The flow rates and the voltages applied to the DMAs were well calibrated before the experiments. Three sizes of polystyrene latex (PSL) spheres ($81 \pm 3$ nm, $147 \pm 3$ nm, $269 \pm 5$ nm) with a density of 1.05 g cm$^{-3}$ were nebulized with an atomizer (TSI, Model 3076) and then dried by a diffusion dryer filled with silica gel for size calibration of the SMPS system. The same PSL spheres were also used for the calibration of APM (see Fig. S3). The bias of SMPS size and APM effective density measurements were within $\pm 4\%$ and $\pm 3\%$, respectively."

The uncertainty analysis of the SMPS measurements was as following (SI Lines 71-90):

"PM$_{SMPS}$ was calculated using Eq. (3) in the main text. Theoretically, the uncertainty of PM$_{SMPS}$ ($\sigma_{PM-SMPS}$) was associated with $d_{m,i}$, $dN_i$, $\rho_{eff,i}$, $E_L(d_{m,i})$, and $E_{s,i}$. As discussed in the instrument calibration in Section 1, uncertainties of $d_{m,i}$ ($\sigma_{dm}$) and $\rho_{eff}$ ($\sigma_\rho$) were within $\pm 4\%$ and $\pm 3\%$, respectively. The uncertainty of $E_L$ ($\sigma_{EL}$) in the range of 40-600 nm ($d_{va}$) was estimated to be $\pm 10\%$ according to Figure S2. The uncertainty of $dN_i$ ($\sigma_N$), which was the combination of the uncertainties of particle charging efficiency ($\sigma_{char}$), diffusion loss ($\sigma_{diff}$), and CPC counting efficiency ($\sigma_{count}$) (Buonanno et al., 2009), was assigned to be $\pm 10\%$ for particles with a diameter of 20-200 nm and $\pm 20\%$ for 200-800 nm (Wiedensohler et al., 2012; Wiedensohler et al., 2018). The uncertainty of $E_s$ ($\sigma_{ES}$) was estimated to be $\pm 5\%$. The uncertainty of PM$_{SMPS}$ was hence estimated by Eq. (S3) using the averaged sized distribution (as shown in Figures S4 and S6) and $\rho_{eff}$ in each experiment:

$$\sigma_{PM-SMPS}$$

$$= \sqrt{\sum_{i=1}^{n} PM_{SMPS}^2(d_{m,i}) + \sum_{i=1}^{n} PM_{SMPS}^2(dN_i) + \sum_{i=1}^{n} PM_{SMPS}^2(\rho_{eff,i}) + \sum_{i=1}^{n} PM_{SMPS}^2(E_L(d_{m,i})) + \sum_{i=1}^{n} PM_{SMPS}^2(E_{s,i})}$$

(S3)

where $PM_{SMPS}(x_i)$ is the product of the uncertainties of each bin $x_i$ ($\sigma_{x,i}$), and the corresponding sensitivity coefficient $\frac{\partial PM_{SMPS,i}}{\partial x_i}$, as shown in Eq. (S4):

$$PM_{SMPS}(x_i) = \frac{\partial PM_{SMPS,i}}{\partial x_i} \cdot \sigma_{x,i}$$

(S4)

Then, Eq (S3) can be simplified as Eq. (S5):

$$\sigma_{PM-SMPS} = \sqrt{\sum_{i=1}^{n}(9\sigma_{dm,i}^2 + \sigma_{dN,i}^2 + \sigma_{\rho_{eff,i}}^2 + \sigma_{EL,i}^2 + \sigma_{ES,i}^2)PM_{SMPS,i}^2} =$$

$$\sqrt{\sum_{i=1}^{n^*} 0.0378 PM_{SMPS,i}^2 + \sum_{i=n^*}^{n} 0.0678 PM_{SMPS,i}^2}$$

(S5)

where $n^*$ corresponds to the bins with $d_m$ larger than 200 nm. Overall, the uncertainty of $PM_{SMPS}$ was estimated to be between ±19.4% and ±26.0% with the exact value depending on the size distribution (Figures S4 and S6)."

*7) The uncertainty in each term of the equation for RF, as well as in 1-6 above, needs to be quantified, and the uncertainty needs to be propagated. Only then we can know if any of the reported trends for (for example) Eb have any meaning.*

Response: The uncertainty analysis is detailed in the supplementary information section 2 (SI Lines 63-110):
"**2 Uncertainty Analysis**

In this work, the reported quantity was considered as a function (y) of measured variables ($x_1$, $x_2$, $x_3$ …$x_n$) in the form of $f(x_1, x_2,…, x_n)$. The uncertainties associated with measured $x_1$, $x_2$, $x_3$ …$x_n$ were $\sigma_1$, $\sigma_2$, …$\sigma_n$, respectively. Accordingly, the mean value of y ($\bar{y}$) and its uncertainty ($\sigma_y$) were calculated using Eq. (S1) and Eq. (S2) as following:

$$\bar{y} = f(\overline{x_1}, \overline{x_2}, …, \overline{x_n})$$

(S1)

$$\sigma_y = \sqrt{(\frac{\partial y}{\partial x_1})^2\sigma_1^2 + (\frac{\partial y}{\partial x_2})^2\sigma_2^2 + \cdots + (\frac{\partial y}{\partial x_n})^2\sigma_n^2}$$

(S2)

**2.1 PM$_{SMPS}$ uncertainty**

PM$_{SMPS}$ was calculated using Eq. (3) in the main text. Theoretically, the uncertainty of PM$_{SMPS}$ ($\sigma_{PM-SMPS}$) was associated with $d_{m,i}$, $dN_i$, $\rho_{eff,i}$, $E_L(d_{m,i})$, and $E_{s,i}$. As discussed in the instrument calibration in Section 1, uncertainties of $d_{m,i}$ ($\sigma_{dm}$) and $\rho_{eff}$ ($\sigma_\rho$) were within ±4% and ±3%, respectively. The uncertainty of $E_L$ ($\sigma_{EL}$) in the range of 40-600 nm ($d_{va}$) was estimated to be ±10% according to Figure S2. The uncertainty of $dN_i$ ($\sigma_N$), which was the combination of the uncertainties of particle charging efficiency ($\sigma_{char}$), diffusion loss ($\sigma_{diff}$), and CPC counting efficiency ($\sigma_{count}$) (Buonanno et al., 2009), was assigned to be ±10% for particles with a diameter of 20-200 nm and ±20% for 200-800 nm (Wiedensohler et al., 2012; Wiedensohler et al., 2018). The uncertainty of $E_s$ ($\sigma_{ES}$) was estimated to be ±5%. The uncertainty of PM$_{SMPS}$ was hence estimated by Eq. (S3) using the averaged sized distribution (as shown in Figures S4 and S6) and $\rho_{eff}$ in each experiment:

$$\sigma_{PM-SMPS}$$

$$= \sqrt{\sum_{i=1}^{n} PM_{SMPS}^2(d_{m,i}) + \sum_{i=1}^{n} PM_{SMPS}^2(dN_i) + \sum_{i=1}^{n} PM_{SMPS}^2(\rho_{eff,i}) + \sum_{i=1}^{n} PM_{SMPS}^2(E_L(d_{m,i})) + \sum_{i=1}^{n} PM_{SMPS}^2(E_{s,i})}$$

(S3)

where $PM_{SMPS}(x_i)$ is the product of the uncertainties of each bin $x_i$ ($\sigma_{x,i}$), and the corresponding sensitivity coefficient $\frac{\partial PM_{SMPS,i}}{\partial x_i}$, as shown in Eq. (S4):

$$PM_{SMPS}(x_i) = \frac{\partial PM_{SMPS,i}}{\partial x_i} \cdot \sigma_{x,i} \qquad (S4)$$

Then, Eq (S3) can be simplified as Eq. (S5):

$$\sigma_{PM-SMPS} = \sqrt{\sum_{i=1}^{n}(9\sigma_{dm,i}^2 + \sigma_{dN,i}^2 + \sigma_{P_{eff,i}}^2 + \sigma_{EL,i}^2 + \sigma_{ES,i}^2)PM_{SMPS,i}^2} =$$

$$\sqrt{\sum_{i=1}^{n^*} 0.0378 PM_{SMPS,i}^2 + \sum_{i=n^*}^{n} 0.0678 PM_{SMPS,i}^2}$$

$$(S5)$$

where n* corresponds to the bins with $d_m$ larger than 200 nm. Overall, the uncertainty of $PM_{SMPS}$ was estimated to be between ±19.4% and ±26.0% with the exact value depending on the size distribution (Figures S4 and S6).

**2.2 RRF uncertainty**

    RRF was calculated with Eq. (2) in the main text. The uncertainty of RRF ($\sigma_{RRF}$) was related to $PM_{ACSM}$ and $PM_{SMPS}$ and was calculated using Eq. (S6). $PM_{SMPS}$ uncertainty obtained from Section 2.1 of the SI was used to calculate the RRF uncertainty, instead of using the standard deviations of the $PM_{SMPS}$ measured in the "relative stable state" of chamber studies. While for the case of Q-ACSM, the standard deviation in the stable state was used to evaluate the RRF uncertainty.

$$\sigma_{RRF} = \sqrt{\left(\frac{\partial RRF}{\partial PM_{ACSM}}\right)^2 \sigma_{PM-ACSM}^2 + \left(\frac{\partial RRF}{\partial PM_{SMPS}}\right)^2 \sigma_{PM-SMPS}^2} = \sqrt{\left(\frac{\sigma_{PM-ACSM}}{PM_{SMPS}}\right)^2 + \left(\frac{PM_{ACSM}\sigma_{PM-SMPS}}{PM_{SMPS}^2}\right)^2} \quad (S6)$$

**2.3 CE uncertainty**

    For the experiments using AS seeds, the collection efficiencies before and after SOA coating were calculated using Eq. (S7) and Eq. (S8), respectively:

$$CE_{AS} = \frac{0.5 PM_{ACSM-AS1}}{PM_{SMPS-AS}} \qquad (S7)$$

$$CE_{org} = \frac{0.5 PM_{ACSM-AS2}}{PM_{SMPS-AS}} \qquad (S8)$$

where, $PM_{ACSM-AS1}$ and $PM_{ACSM-AS2}$ represents average Q-ACSM measured mass concentrations before and after SOA coating, respectively. $PM_{SMPS-AS}$ was the mass concentration of AS measured by SMPS before SOA coating. The default CE is 0.5. The uncertainties of $CE_{AS}$ and $CE_{org}$ were calculated with the Eq. (S9) and Eq. (S10), respectively.

$$\sigma_{CE-AS} = 0.5\sqrt{(\frac{\sigma_{PM-ACSM-AS1}}{PM_{SMPS-AS}})^2 + \left(\frac{PM_{ACSM-AS1}\sigma_{PM-SMPS-AS}}{PM_{SMPS-AS}^2}\right)^2}$$

$$(S9)$$

$$\sigma_{CE-org} = 0.5 \sqrt{\left(\frac{\sigma_{PM-ACSM-AS2}}{PM_{SMPS-AS}}\right)^2 + \left(\frac{PM_{ACSM-AS2}\sigma_{PM-SMPS-AS}}{PM_{SMPS-AS}^2}\right)^2}$$

(S10)"

4. *"For Also it should be noted that for internally mixed particles such as those generated here Eb will be a property of the particle that is influenced by its sulfate and OA content. So it is nonsensical to write Eb,SO4 and Eb,OA (as the authors do in their notation) as if those were different quantities in the seeded experiments, or as if they were a property of the species and not the mixture.*

*Moreover, it is not standard practice to use ACSM or AMS measurements for quantitative yield experiments for chamber experiments in the first place, due to the complexities outlined above. Rather, researchers typically have used SMPS volume together with density estimates (or comparisons with filters as well as other methods), and OA/SO4 ratios, to quantify yields. Jimenez et al. 2016 (which is referenced in this manuscript) states "Finally, we reiterate the need for direct calibration in laboratory experiments utilizing specific organic compounds or mixtures." So, this issue is not new and nor do the authors offer an advance in addressing the issue.*

*As discussed on that reference, the trends of AMS response observed for field and laboratory experiments are often different, due to the wider range of particle materials that can be made in the lab. Lab experiments typically have trouble replicating the OA composition observed in the atmosphere. Therefore, no implication should be made that the results from a lab study apply to field measurements. Any statements about the performance of the AMS / ACSM in field measurements should be derived from the analysis of field measurements, of which there is an extensive record in the published literature. Comparisons of AMS with SMPS in field studies are not consistent with the trends shown here, see for example Jimenez et al. (2016) and references therein.*

Response: The SOA particles generated during the experiments using AS seeds should be assume a core-shell morphology. We have calculated the coating thickness and found that 18-25 nm should be thick enough to fully cover the AS seeds. AS in most of the experiments were fully covered (Lines 278-282). After the AS particles are fully covered by organic shell, the collection efficiency should reflect the property of the SOA shell. Thus, the CE measured with/without SOA coating were considered as the CE$_{AS}$ and CE$_{org}$, respectively. We agree with reviewer that filter-based aerosol measurement can achieve higher precision but it also requires longer analysis time and pretreatment of the filter. SMPS-APM technique can achieve the same goal within 10 min and thus can be used to track the changes of SOA insider the chamber with higher time resolution and high precision (+/-3%).

Due to the low-cost and durability, Q-ACSMs have been widely used in both ambient and laboratory experiments to quantify aerosol chemical compositions, especially for long-term observations in highly polluted regions (such as in China). However, the performance of Q-ACSM in organics measurement was rarely evaluated because of its low time and m/z resolution and the lack of particle size selection. To the best of our knowledge, this study represents the first comprehensive chamber calibration of CE$_{org}$ and RRF$_{org}$ for a Q-ACSM. Our results of RRF and CE$_{org}$ calibration were generally in line with previous studies conducted with the AMS.

5. *"Other Important Comments:*
*As was extensively detailed by Reviewer 2, the manuscript is full of grammatical errors, confusing text, logical flaws and overstatements of conclusions. I agree with the comments of Rev. 2, and will thus focus on the most important additional points, and a few similar points with additional context or emphasis. However, the list is not comprehensive as it would be too time consuming to point out all the issues and any effort to revise and resubmit the paper should go beyond just the issues detailed by the reviewers. A detailed review of the Abstract is below, followed by other issues organized by topic.*

Response: We have revised the manuscript thoroughly as suggested by all reviewers. The revised manuscript has been proofread by a native English speaker. Point-to-point responses to reviewer 2's comments have been uploaded separately.

6. *"Abstract: L21: "comprehensively" is an extreme overstatement.*

Response: It has been deleted.

*L25/L31/L32: "oxidation states" were not measured in this study.*

Response: "oxidation states" have been replaced with "$f_{44}$" in the revised manuscript.

*L27: "exact mass" doesn't make sense.*

Response: It has been removed.

*L35: The cause of decreasing RF was not "pinpointed" or even significantly constrained.*
*L36: The CE was not determined quantitatively.*
*L37-39: "Our experiment results along with previous literature reports strongly implied that as the SOA oxidation state increases, SOA will transform gradually from a liquid state (CE » 1) into a solid (or glassy) state with a CE of 0.2~0.5." This is a gross overstatement of what the evidence presented in this manuscript supports. The trends observed may have been any combination of lens transmission effects, Eb effects of mixed dry ammonium sulfate / OA particles, OA RIE effects, OA CE effects, or variable errors on the SMPS or CPC used for comparison. Neither CE effects nor phase state effects were isolated in this study.*

Response: After extensive calibrations and uncertainty analysis, the statement related to RF and CE has been revised as (lines 30-40): "Our results showed that RRF for a specific type of SOA anti-correlated with $f_{44}$ regardless of the VOC precursors. RRF decreased from 2.16 to 0.81 when $f_{44}$ increased from 0.057 to 0.191. Meanwhile, $\rho_{eff}$ increased from 1.09 to 1.34 g cm$^{-3}$, implying more compact structure for more oxidized SOA. The anti-correlation between RRF and $f_{44}$ might be due to the decreased RIE$_{org}$ or CE$_{org}$ for more oxidized SOA. To further explore the actual cause, ammonium sulfate (AS) seed particles were injected into the chamber before SOA were produced. After the 80 nm AS particles were fully coated with SOA (coating thickness > 18~25 nm), CE of the aerosol changed from CE$_{AS}$ (0.285±0.067) to CE$_{org}$ of the SOA shell. The experiment showed that CE$_{org}$ decreased from 0.882 to 0.313~0.475 when $f_{44}$ increased from 0.127 to 0.209, most likely caused by a phase change of SOA from liquid to solid/glassy.

7. *"L39-41: "Meanwhile, the RIE of OA decreased substantially when SOA transformed from hydrocarbon-like OA (HOA) into more oxygenated OA (OOA) and may further decrease as O/C continued to increase." Like with Eb, the effects of RIE of OA were NOT isolated in this study. Moreover, the results presented here have no bearing on HOA since only chamber SOA was studied. Also, no data on O/C was presented here (inferring O/C from f44 for chamber SOA is not justified – see details below)"*

Response: Since HOA particles were not measured in this study, we have limited our conclusion to be based on chamber-generated SOA only. Therefore, Fig. 8 has been removed from the manuscript.

8. *"L41-44: "Our results indicated that the current Q-ACSM calibration procedure using a constant RIE may lead to somewhat underestimation of more oxidized OOA but overestimation of less oxidized HOA, i.e., a variable RIE shall be applied, most likely as a function of the SOA oxidation state." As stated above (and further explained below), RIE effects were not isolated."*

Response: The sentences have been revised as (Lines 37-41): "The variation of $CE_{org}$ with $f_{44}$ could explain a large fraction of the observed decrease in RRF, while the influence of $RIE_{org}$ cannot be excluded. The trends in RRF and $CE_{org}$ for Q-ACSM were in accordance with those done with the high-resolution AMS, demonstrating the capability of the low-resolution Q-ACSM in doing SOA laboratory studies and the necessity to calibrate RF when conducting laboratory SOA experiments."

9. *"More Detail on Lens Transmission Effects: The ACSM lens transmission declines to zero for both small and large particles. In particular the upper size cut depends on the specific lens being used (Hu et al., 2017), and can even change in time after an instrument is moved to a different location. The transmission also depends strongly on the pressure inside the AMS aerodynamic lens, which can decline dramatically if the pressure in the lens is reduced (e.g. Bahreini et al., 2003), for example due to a partially clogged critical orifice, or to just being set at the wrong pressure by using an improperly sized critical orifice. This value should be documented here for all experiments, and care should be taken to ensure that all SOA experiments are done with the same lens pressure used in characterizing EI vs. size. The trends of the response factor shown here may be significantly influenced, or may even be dominated, by differences in the particle size ranges analyzed by the ACSM and SMPS. It is not acceptable to publish a paper that attempts to quantify CE and RIE and the ACSM without a detailed and careful quantitative calibration of the transmission of the ACSM vs size, and applying that transmission curve to all SMPS measurements. Using transmission curves from the literature is not acceptable, as there is considerable variation in those, and it is not known which literature curve (if any) may apply to the ACSM used here. I would expect to see several figures and pages of text devoted to calibrating the AMS size transmission before any conclusions about CE or RIE could be credible."*

Response: The $E_L$ calibration procedure and results (Fig. S2) have been described in the supplementary information (SI Lines 32-51). A 100 μm critical orifice was used in front of ACSM, the pressure in the chamber was 1.22±0.02 torr among all the experiments (Lines 157-158). When the pressure gets lower that 1.20 torr, the critical orifice would be cleaned. As a result, $E_L$ should be relatively constant during the experiments. More details regarding $E_L$ calibration has also been given in the response to comment #3.

*No volume size distributions are shown here for the different experiments, so it is impossible for the reader to even estimate the extent to which differences in particle size transmission could be a problem here.*

Response: The size distributions of particles during different experiments and the changes of size distribution within one experiments have been provided into the supplementary information (Figs. S4 and S6). The supplementary information also provided the detailed procedure of $PM_{SMPS}$ mass concentration calculation.

*As Reviewer 2 points out, the decreasing trend in RF vs f44 (Fig. 5a) could be simply due to an artifact of increasing particle size extending beyond the upper size limit of 100% aerodynamic lens transmission efficiency. While a well-performing lens can transmit particles with 100% efficiency up to ~550 nm vacuum aerodynamic diameter (~400-450 nm mobility diameter; Knote et al., 2011; Hu et al., 2017), lenses with reduced transmission at larger sizes is not uncommon (Liu et al., 2007; Takegawa et al., 2009; Hu et al., 2017; Campuzano-Jost et al. 2017), resulting in large losses for sizes as small as 450 nm vacuum aerodynamic diameter (~300-350 nm mobility diameter). Such effect may also play a role in the results presented in Fig. 6 and Fig. 7, as suggested by Rev. 2. Only then should the trends in RF can be considered reliable.*

Response: According to our calibration, "Lens transmission efficiencies for 300-600 nm particle were nearly unity, which was in reasonable accordance with some previous studies (Jayne et al., 2000; Knote et al., 2011; Hu et al., 2018), but the diameter range with a unity lens transmission efficiency was much broader than that reported by Liu et al. (2007). The blue line in Figure S2 was used to modify size

distribution in our experiments (i.e., 0% transmission for $d_{va}$ below 40 nm; linear increase in transmission vs log($d_{va}$), from 0% to 100% at $d_{va}$ = 100 nm; 100% transmission efficiency from $d_{va}$ = 100 nm up to $d_{va}$ = 580 nm; linear decrease in transmission vs log($d_{va}$) from 100% at 580 nm to 0% at 1200 nm). For $d_p$ below 300 nm, we used the results reported by Knote et al. (2011) indicated by the red dashed line in Figure S2. About 10% of uncertainty (shaded area) was applied to the $E_L$ used in this study as indicated by the blue curve in Figure S2." (SI Lines 42-51)

10. *"Collection Efficiency and Relative Ionization Efficiency Sections:   The sections on CE (sect. 3.3) and RIE (sect. 3.4) are extremely under-supported, misleading, and highly speculative. Given the experiments conducted and methods used, separation of the effects of CE and RIE does not seem possible. This applies to both relative trends and absolute values. Those sections should be completely eliminated unless unambiguous new evidence can be provided. The RIE section provides no experimental evidence and only an incomplete discussion of the published literature on this topic. The CE section discusses experiments where SOA was coated on top of dry ammonium sulfate seed and the "Response Factor (RF)" was calculated. However interpretation of the results are flawed. The RIE of sulfate was not calibrated, even though it may vary substantially (e.g., Zhang et al., 2017). Despite the lack of sulfate calibration, the authors assume an RIE of sulfate and calculate a CE of ammonium sulfate that depends proportionally on that value. While it is stated that the absolute value is not important but rather just trends matter, in the same paragraph the argument that when a lot of SOA is added to the ammonium sulfate seed, a CE of unity is observed which supports that the aerosol is now in a liquid state. Not only does that line of logic require that ammonium sulfate is calibrated, it also requires that the RIE of the OA is known. So, generally the main figure of that section (Fig. 7) is very problematic since the RIE of SO4 and OA are not known. Also, note that calculation of a CE for dry ammonium sulfate of 0.28 in this work does not narrow the range of uncertainty from lack of calibration for ammonium sulfate RIE since a range of a factor of 2 (0.2-0.4) has been observed for dry ammonium sulfate (Hu et al., 2017 and references therein). Moreover, equating the trends in the RF to changes in the RIE of the SOA is not necessarily justified since the effects may be the "results of a combination of a AS core and organic shell" as the authors state in Line 305. Thus, any trends in Fig. 7 may be dominated by the SOA coating thicknesses. However, changes in CE and RIE of the SOA may also be factors. Consequently, any quantitative conclusions that can be drawn about CE of OA is extremely under-constrained here. In order to separate RIE and CE effects, a direct, unambiguous measurement of CE needs to be made (e.g., see Xu et al., 2018)."*

Response: $CE_{AS}$ was measured during the AS seed experiments. Our results were a little higher than the reported value. The manuscript has been revised as (Lines 268-277) : "$CE_{AS}$ for dry AS particles varied from 0.265 to 0.298 in different experiments, with an average value of 0.285 ± 0.067 which was slightly higher than the reported 0.24 ± 0.03 (Matthew et al., 2008). The large uncertainty mainly originated from SMPS measurements and the bias might come from the using of default RIE value. In the case of $RIE_{SO4}$, a value of 1.15 was used here as suggested by previous studies (Canagaratna et al., 2007; Ng et al., 2011; Petit et al., 2015), which would not likely change during the organic coating experiments (Matthew et al., 2008). However, recent studies have shown that $RIE_{SO4}$ might vary with different instruments (Budisulistiorini et al., 2014; Crenn et al., 2015). For most of the experiments, all condensable organic vapor seemed to condense directly onto seed particles instead of forming new particles. While in one experiment (Exp. 22), there were small amounts of new particles formed, which was indicated by a smaller mode on the size distribution (Fig. S6) and further confirmed by a sudden increase in total particle number concentration. However, the newly formed particles only contribute slightly to the total particulate mass for their much smaller sizes. In addition, to avoid the influence of new particle formation on $CE_{org}$ or $RIE_{org}$ deduction, only $PM_{ACSM}$ and $PM_{SMPS}$ associated with AS signals were used to deduce the actual $CE_{org}$ in our experiments"

11. *"Incomplete Literature Context / Major Omissions:   Several instances stand out as having a major lack of literature review/support or simply omitting fundamental references on topics discussed. For*

*example, the Kuwata et al. (AS&T 2012) paper which formulates the relationship between SOA density and O/C and H/C is never referenced or discussed (or any other paper on such relationships). A reader might get the impression that this manuscript was the first to show a correlation between SOA density and oxidation. Also, the paper Xu et al. (AS&T 2018), which extensively investigates RIE of OA for a large range of compounds and SOA (unambiguously separating CE and RIE), is completely missing. Another example is the section "Effects of f44" (3.2), where only a few peripheral references are provided in discussion of these results; whereas, there is a large body of work discussing trends in f's and elemental ratios for SOA chamber studies. Even the papers introducing the f44 vs f43 diagram (Ng et al., 2010; 2011) are not referenced! Without such context, it is impossible to determine if anything new was learned from the present studies."*

Response: The related studies on SOA densities were referenced in lines 225-231 (George and Abbatt, 2010; Kuwata et al., 2012). The most recent studies about $RIE_{org}/CE_{org}$ were cited in the revised manuscript (line 107 and lines 307-328) (Robinson et al., 2017; Xu et al., 2018). The papers introducing $f_{44}$ vs $f_{43}$ diagram was referenced in line 68 (Ng et al., 2010; Ng et al., 2011). A specific section (see *3.4 Comparison with AMS Calibrations*) was added to compare with all the related works we know that was conducted by AMS. One of the strength of our calibration compared to the former calibrations was that we related $CE_{org}$ variation with $f_{44}$ to effective density variations, which possibly indicated a phase change. This was the first attempt to precisely evaluate the $CE_{org}$ of a low-resolution Q-ACSM for various chamber-generated SOAs, which indicated the critical need for more complicated calibrations before deploying a Q-ACSM for long-term field measurements.

12. *"f44, O/C, Oxidation State:   The use of f44 is used throughout the manuscript to be equivalent to O/C and "oxidation state," which is an unjustified step for chamber data. The authors state in Lines 200-203: "Since m/z 44 signal basically reflected the oxygen content in OA, the O/C ratio can be deduced directly from f44, both of which have been widely used to represent the oxidation state of OA (Canagaratna et al., 2015)." While this may be true for ambient OA, this does not necessarily apply to chamber data. Canagaratna et al. (2015) (Fig. 8a) shows that relationship for ambient, standards, and laboratory produced SOA (a-pinene, toluene, isoprene, sesquiterpenes). Ambient OA and some standards follow a well-established relationship. However, taken together the chamber SOA shows no correlation, which is not surprising since the presence of acids (which produce CO2+ in the AMS/ACSM) and other oxidized functional groups would not necessarily be expected to track for specific oxidant/VOC systems. Only sesquiterpenes and a-pinene fall on the ambient parameterization line and are clustered within a narrow region.*
*Moreover, if the RIEs of OA indeed vary a lot for the systems studied then it would seem very possible that changes in f44 could be controlled as much by changes in the overall OA sensitivity (the denominator of f44 = m/z 44 / OA) as by the actual relative abundance of m/z 44 in the SOA. CO2+ in the AMS is expected to be largely formed from the decarboxylation of organic acid groups on the vaporizer, forming CO2 gas, which would have a constant RIE, and thus would not track changes in overall OA RIE.*
*Therefore, the use of the terms O/C and oxidation state should be avoided beyond an explanation that it might be roughly indicative of oxygen content and possibly of acid content."*

Response: As suggested by the reviewer, O/C and "oxidation state" have been replaced with the term "$f_{44}$".

13. *"Clarity in Scope:   All conclusions and interpretation should be scaled back to clearly state to apply to only RF (CE x RIE), effective density, f44 (not O/C or oxidation state), three oxidant/VOC SOA systems, dry conditions, and laboratory SOA. No extrapolation to ambient OA should be given or implied, since no evidence is presented and current evidence in literature suggests that such an extrapolation is not warranted – see e.g., Xu et al., AS&T 2018, Jimenez et al., AS&T 2016."*

Response: Extrapolation to HOA or ambient OA were removed from the revised manuscript.

14. *"Figure 8:   This figure should be removed: no axis are given, and the data for several of the trends have not been derived in this paper, so this is really a cartoon from the author's imagination, which has no place in a paper. If the authors do all the extra work to make this a serious study, then the figure should be included with quantitative axes and with error bars for both X and Y.*
*In addition, the trend in density is not new and was published at least half a decade ago (Kuwata et al., 2012). The trends in CE and RIE of OA are not demonstrated in this paper and are especially misleading. The trends in RF may be dominated by lens transmission effects, and may have a contribution from the instrument that is being compared with, and can only be supported with further evidence. HOA is not studied here at all.*
*Also note that a published study (Pajunoja et al., 2016) reported the opposite trend for bounce vs. O/C, in that study bounce decreased as O/C increased."*

Response: Figure 8 has been removed.

15. *Other Miscellaneous Comments (substantial and minor):*
*L75: Real time composition measurements is the key development that AMS made possible/routine (not sized-resolved).*

Response: It has been revised as (Line 57): "real time size-resolved aerosol measurements".

*L93-103: In this paragraph where CE and other effects on AMS sensitivity are introduced, it would be useful to instead use the more precise terms of each of the components of CE: lens, shape-related, and bounce effects (CE = EL * Es * Eb; Huffman et al., AS&T 2005).*

Response: These terms have been described as (Lines 88-93): "CE is mainly composed of three parts: $E_L$ (the transmission efficiency of particles as a function of vacuum aerodynamic diameter ($d_{va}$) through the aerodynamic lens), $E_s$ (to account for the disperse loss of non-spherical particles from the main air stream), $E_b$ (to account for the bouncing of particles from the vaporizer surface). Among these, $E_L$ can be explicitly taken into account by applying transmission efficiency in the aerodynamic lens, $E_s$ is usually near unit for most particle types according to previous beam width measurements, and $E_b$ is usually the most variable and uncertain part (Huffman et al., 2005; Salcedo et al., 2007)."

*L101-103: This statement about the high uncertainty in CE and IE is unclear whether it applies to ambient or laboratory data and should be clarified (since it is not true for ambient data, per the Jimenez et al. 2016 reference cited).*

Response: It has been revised as (Line 82): "laboratory generated SOA".

*L124-125. AMS light scattering does not employ laser-based vaporization.*

Response: It has been changed into (Line 108): "light scattering-based detections".

*L126-130. This statement claiming that the increased fragmentation in the capture vaporizer will "highly complicate the AMS quantification process" does not reflect conclusions of the Hu et al., 2017 paper cited here, nor any of the several other papers investigating the capture vaporizer. For example Hu et al. (2018a) state that "tracer ion marker fractions, which are used to characterize the impact of different sources are still present and usable in the CV." and Hu et al. (2018b) state "Consistent time series of positive matrix factorization (PMF) factors and their fractions of total OA were found across the CV and SV in the three very different ambient data sets ranging from biogenic- to anthropogenic-dominated, indicating limited loss of source determination information despite the increased fragmentation." It appears that this*

*statement was added to dismiss this method as a practical way to reduce CE uncertainties. In fact, in the recent paper by Xu et al., AS&T, 2018, it was shown that the capture vaporizer can be used as a method to isolate changes in OA RIE. Given the other limitations of the ACSM, the authors may want to consider performing a similar study using a CV, which would make Eb~1 and thus substantially simplify the interpretation of the trends on RF.*

Response: It has been revised as "Most recently, a new type of capture vaporizer has been developed to achieve a unit CE (Hu et al., 2017). $RIE_{org}$ can be directly measured with this capture vaporizer (Xu et al., 2018). Nevertheless, the determination of $RIE_{org}$ and $CE_{org}$ appears to be a complicated but essential issue for accurate AMS/ACSM OA measurements that necessitate further investigations, including to explore new ways of SOA standard generation and to constrain the measurement uncertainty with various measurement techniques." (Lines 110-114)

*L162: "self-synthesized" is confusing. Consider alternative phrasing such as "synthesized in house".*

Response: The phrase "self-synthesized" has been deleted.

*L177: "m/z calibration", not "mass calibration"*

Response: The phrase "mass calibration" has been replaced with "m/z calibration". (Line 155)

*L195: add "a" before "strong"*

Response: It has been revised accordingly. (Line 199)

*L235-238: "Especially, laboratory studies have shown that the ozonolysis of α-pinene can lead to considerably higher SOA yield than that from OH initiated reactions (Yao et al., 2014). Therefore, in this work we were focusing on the isoprene-OH and α-pinene-O3 reactions only".*
*This does not seem like a good reason to omit these data from the analysis. OH oxidation is an important loss for a-pinene in the atmosphere and makes substantial SOA. This data should be shown and discussed. Also, Yao et al. should not be the primary reference here.*

Response: Both OH and $O_3$ initiated oxidations of α-pinene were included in the study and the reaction conditions were listed in Table 1. The results in Yao et al. (2014) only showed that for the same amount of α-pinene, ozonolysis reaction can lead to more SOA yield, which may not necessarily imply the actual relative importance of α-pinene oxidation channels under ambient conditions. These sentences have been removed.

*L243: grammar.*

Response: It has been revised as (Line 283): "For α-pinene initiated SOA,"

*L291: grammar*

Response: It has been revised as (Line 264): "Eight experiments using AS seeds were conducted in total (Table 2)".

*L293: grammar*

Response: The sentence has been deleted.

References:
Jayne, J. T., Leard, D. C., Zhang, X. F., Davidovits, P., Smith, K. A., Kolb, C. E., and Worsnop, D. R.: Development of an aerosol mass spectrometer for size and composition analysis of submicron particles, Aerosol Science and Technology, 33, 49-70, 10.1080/027868200410840, 2000.

Knote, C., Brunner, D., Vogel, H., Allan, J., Asmi, A., Äijälä, M. Y. E., Carbone, S., Denier van der Gon, H., Jiminez, J., and Kiendler-Scharr, A.: Towards an online-coupled chemistry-climate model: evaluation of trace gases and aerosols in COSMO-ART, Geosci Model Dev, 2011.

Hu, W., Day, D. A., Campuzano-Jost, P., Nault, B. A., Park, T., Lee, T., Croteau, P., Canagaratna, M. R., Jayne, J. T., and Worsnop, D. R.: Evaluation of the new capture vaporizer for Aerosol Mass Spectrometers: Characterization of organic aerosol mass spectra, Aerosol Sci Tech, 1-44, 2018.

Liu, P. S. K., Deng, R., Smith, K. A., Williams, L. R., Jayne, J. T., Canagaratna, M. R., Moore, K., Onasch, T. B., Worsnop, D. R., and Deshler, T.: Transmission efficiency of an aerodynamic focusing lens system: Comparison of model calculations and laboratory measurements for the Aerodyne Aerosol Mass Spectrometer, Aerosol Sci Tech, 41, 721-733, 10.1080/02786820701422278, 2007.

Hu, W., Campuzano-Jost, P., Day, D. A., Croteau, P., Canagaratna, M. R., Jayne, J. T., Worsnop, D. R., and Jimenez, J. L.: Evaluation of the new capture vaporizer for aerosol mass spectrometers (AMS) through field studies of inorganic species, Aerosol Sci Tech, 51, 735-754, 10.1080/02786826.2017.1296104, 2017.

DeCarlo, P. F., Slowik, J. G., Worsnop, D. R., Davidovits, P., and Jimenez, J. L.: Particle morphology and density characterization by combined mobility and aerodynamic diameter measurements. Part 1: Theory, Aerosol Sci Tech, 38, 1185-1205, 10.1080/027868290903907, 2004.

McMurry, P. H., Wang, X., Park, K., and Ehara, K.: The Relationship between Mass and Mobility for Atmospheric Particles: A New Technique for Measuring Particle Density, Aerosol Science and Technology, 36, 227-238, 10.1080/027868202753504083, 2002.

Buonanno, G., Dell'Isola, M., Stabile, L., and Viola, A.: Uncertainty Budget of the SMPS–APS System in the Measurement of PM1, PM2. 5, and PM10, Aerosol Sci Tech, 43, 1130-1141, 2009.

Matthew, B. M., Middlebrook, A. M., and Onasch, T. B.: Collection efficiencies in an Aerodyne Aerosol Mass Spectrometer as a function of particle phase for laboratory generated aerosols, Aerosol Sci Tech, 42, 884-898, 10.1080/02786820802356797, 2008.

Canagaratna, M. R., Jayne, J. T., Jimenez, J. L., Allan, J. D., Alfarra, M. R., Zhang, Q., Onasch, T. B., Drewnick, F., Coe, H., Middlebrook, A., Delia, A., Williams, L. R., Trimborn, A. M., Northway, M. J., DeCarlo, P. F., Kolb, C. E., Davidovits, P., and Worsnop, D. R.: Chemical and microphysical characterization of ambient aerosols with the aerodyne aerosol mass spectrometer, Mass Spectrometry Reviews, 26, 185-222, 10.1002/mas.20115, 2007.

Ng, N. L., Herndon, S. C., Trimborn, A., Canagaratna, M. R., Croteau, P. L., Onasch, T. B., Sueper, D., Worsnop, D. R., Zhang, Q., Sun, Y. L., and Jayne, J. T.: An Aerosol Chemical Speciation Monitor (ACSM)

for Routine Monitoring of the Composition and Mass Concentrations of Ambient Aerosol, Aerosol Sci Tech, 45, 780-794, 2011.

Crenn, V., Sciare, J., Croteau, P. L., Verlhac, S., Froehlich, R., Belis, C. A., Aas, W., Aijala, M., Alastuey, A., Artinano, B., Baisnee, D., Bonnaire, N., Bressi, M., Canagaratna, M., Canonaco, F., Carbone, C., Cavalli, F., Coz, E., Cubison, M. J., Esser-Gietl, J. K., Green, D. C., Gros, V., Heikkinen, L., Herrmann, H., Lunder, C., Minguillon, M. C., Mocnik, G., O'Dowd, C. D., Ovadnevaite, J., Petit, J. E., Petralia, E., Poulain, L., Priestman, M., Riffault, V., Ripoll, A., Sarda-Esteve, R., Slowik, J. G., Setyan, A., Wiedensohler, A., Baltensperger, U., Prevot, A. S. H., Jayne, J. T., and Favez, O.: ACTRIS ACSM intercomparison - Part 1: Reproducibility of concentration and fragment results from 13 individual Quadrupole Aerosol Chemical Speciation Monitors (Q-ACSM) and consistency with co-located instruments, Atmos Meas Tech, 8, 5063-5087, 10.5194/amt-8-5063-2015, 2015.

George, I. J., and Abbatt, J. P. D.: Chemical evolution of secondary organic aerosol from OH-initiated heterogeneous oxidation, Atmos Chem Phys, 10, 5551-5563, 10.5194/acp-10-5551-2010, 2010.

Kuwata, M., Zorn, S. R., and Martin, S. T.: Using Elemental Ratios to Predict the Density of Organic Material Composed of Carbon, Hydrogen, and Oxygen, Environ Sci Technol, 46, 787-794, 10.1021/es202525q, 2012.

Robinson, E. S., Onasch, T. B., Worsnop, D., and Donahue, N. M.: Collection efficiency of [alpha]-pinene secondary organic aerosol particles explored via light-scattering single-particle aerosol mass spectrometry, Atmos Meas Tech, 10, 1139, 2017.

Xu, W., Lambe, A., Silva, P., Hu, W., Onasch, T., Williams, L., Croteau, P., Zhang, X., Renbaum-Wolff, L., and Fortner, E.: Laboratory evaluation of species-dependent relative ionization efficiencies in the Aerodyne Aerosol Mass Spectrometer, Aerosol Sci Tech, 1-16, 2018.

Ng, N. L., Canagaratna, M. R., Zhang, Q., Jimenez, J. L., Tian, J., Ulbrich, I. M., Kroll, J. H., Docherty, K. S., Chhabra, P. S., Bahreini, R., Murphy, S. M., Seinfeld, J. H., Hildebrandt, L., Donahue, N. M., DeCarlo, P. F., Lanz, V. A., Prevot, A. S. H., Dinar, E., Rudich, Y., and Worsnop, D. R.: Organic aerosol components observed in Northern Hemispheric datasets from Aerosol Mass Spectrometry, Atmospheric Chemistry and Physics, 10, 4625-4641, 10.5194/acp-10-4625-2010, 2010.

Huffman, J. A., Jayne, J. T., Drewnick, F., Aiken, A. C., Onasch, T., Worsnop, D. R., and Jimenez, J. L.: Design, modeling, optimization, and experimental tests of a particle beam width probe for the aerodyne aerosol mass spectrometer, Aerosol Sci Tech, 39, 1143-1163, 10.1080/02786820500423782, 2005.

Salcedo, D., Onasch, T. B., Canagaratna, M. R., Dzepina, K., Huffman, J. A., Jayne, J. T., Worsnop, D. R., Kolb, C. E., Weimer, S., Drewnick, F., Allan, J. D., Delia, A. E., and Jimenez, J. L.: Technical Note: Use of a beam width probe in an Aerosol Mass Spectrometer to monitor particle collection efficiency in the field, Atmos Chem Phys, 7, 549-556, 2007.

Budisulistiorini, S. H., Canagaratna, M. R., Croteau, P. L., Baumann, K., Edgerton, E. S., Kollman, M. S., Ng, N. L., Verma, V., Shaw, S. L., Knipping, E. M., Worsnop, D. R., Jayne, J. T., Weber, R. J., and Surratt, J. D.: Intercomparison of an Aerosol Chemical Speciation Monitor (ACSM) with ambient fine aerosol

measurements in downtown Atlanta, Georgia, Atmos Meas Tech, 7, 1929-1941, 10.5194/amt-7-1929-2014, 2014.

Petit, J. E., Favez, O., Sciare, J., Crenn, V., Sarda-Esteve, R., Bonnaire, N., Mocnik, G., Dupont, J. C., Haeffelin, M., and Leoz-Garziandia, E.: Two years of near real-time chemical composition of submicron aerosols in the region of Paris using an Aerosol Chemical Speciation Monitor (ACSM) and a multi-wavelength Aethalometer, Atmospheric Chemistry and Physics, 15, 2985-3005, 10.5194/acp-15-2985-2015, 2015.

Wiedensohler, A., Birmili, W., Nowak, A., Sonntag, A., Weinhold, K., Merkel, M., Wehner, B., Tuch, T., Pfeifer, S., and Fiebig, M.: Mobility particle size spectrometers: harmonization of technical standards and data structure to facilitate high quality long-term observations of atmospheric particle number size distributions, Atmos Meas Tech, 5, 657-685, 2012.

Wiedensohler, A., Wiesner, A., Weinhold, K., Birmili, W., Hermann, M., Merkel, M., Müller, T., Pfeifer, S., Schmidt, A., and Tuch, T.: Mobility particle size spectrometers: Calibration procedures and measurement uncertainties, Aerosol Sci Tech, 52, 146-164, 2018.

---

## Author Comment (AC5) · 1 Oct 2018

| 1
2 | Response factor of a Q-ACSM to chamber generated SOA with different average
oxygen content                                                                                                                     |
|--------|--------------------------------------------------------------------------------------------------------------------------------------------------------------------------------------------------------------------------|
| 3      |                                                                                                                                                                                                                          |
| 4      | Xiaoxiao Li 1 , Yan Ma 2,3* , Hui Chen 2,3 , Youling Jiang 2,3 , Xin Ma 2,3 , Rujin Yin 1 , Dongsen Yang 2,3 , Xiaowen Shi 2,3 , |
| 5      | Jiming Hao 1 , Jingkun Jiang 1 , and Jun Zheng 2,3*                                                                                                                                     |
| 6      | 1 State Key Joint Laboratory of Environment Simulation and Pollution Control, School of Environment, Tsinghua                                                                                                 |
| 7      | University, 100084 Beijing, China                                                                                                                                                                                        |
| 8      | 2 Collaborative Innovation Center of Atmospheric Environment and Equipment Technology, Nanjing University                                                                                                     |
| 9      | of Information Science & Technology, Nanjing 210044, China                                                                                                                                                               |
| 10     | 3 Joint Laboratory for Air Quality and Climate, Nanjing University of Information Science & Technology, Nanjing                                                                                               |
| 11     | 210044, China                                                                                                                                                                                                            |
| 12     |                                                                                                                                                                                                                          |
| 13     |                                                                                                                                                                                                                          |
| 14     | Corresponding authors: Drs. Jun Zheng and Yan Ma                                                                                                                                                                         |
| 15     | Email: zheng.jun@nuist.edu.cn and mayan@nuist.edu.cn                                                                                                                                                                     |
| 16     | Tel.: +86-18251919852                                                                                                                                                                                             |
| 17     | Fax: +86-25-58731090                                                                                                                                                                                                     |
| 18     | Abstract                                                                                                                                                                                                                 |
| 19     | The response factor (RF) of a quadrupole based aerosol chemical speciation monitor (Q-ACSM) for                                                                                                                          |
| 20     | secondary organic aerosols (SOA) has been investigated in this work. SOA were generated under simulated                                                                                                                  |
| 21     | photochemical oxidation conditions in a 4.5 m 3 Teflon chamber from three different volatile organic compounds                                                                                                |
| 22     | (VOC): $\alpha$ -pinene, isoprene and toluene, representing both biogenic and anthropogenic VOCs. For some                                                                                                               |

23 experiments, different degrees of average oxygen content of SOA, indicated by the ratio of m/z 44 signal over total

24 organic ion signal intensity ( $f_{44}$ ), were obtained by changing the ratio of the VOC precursors to the oxidants (O3 or 25 OH). An aerosol particle mass analyzer (APM) and a scanning mobility particle sizer (SMPS) were used to 26 determine SOA effective density ( $\rho_{eff}$ ) and volume concentrations. Relative response factor (RRF) of the Q-ACSM 27 to organics was determined by dividing Q-ACSM measured mass (the relative ionization efficiency (RIEore) and 28 organic collection efficiency (CEorg) were set to default values of 1.4 and 0.5, respectively) by SMPS-APM deduced 29 mass (amended by aerodynamic lens transmission efficiency  $(E_L)$  and particle loss due to non-spherical shape  $(E_s)$ ). 30 Our results showed that RRF for a specific type of SOA anti-correlated with  $f_{44}$  regardless of the VOC precursors. RRF decreased from 2.16 to 0.81 when  $f_{44}$  increased from 0.057 to 0.191. Meanwhile,  $\rho_{eff}$  increased from 1.09 to 31 1.34 g cm-3, implying more compact structure for more oxidized SOA. The anti-correlation between RRF and  $f_{44}$ 32 33 might be due to the decreased RIEorg or CEorg for more oxidized SOA. To further explore the actual cause, 34 ammonium sulfate (AS) seed particles were injected into the chamber before SOA were produced. After the 80 nm 35 AS particles were fully coated with SOA (coating thickness > 18-25 nm), CE of the aerosol changed from  $CE_{AS}$ 36  $(0.285 \pm 0.067)$  to CEorg of the SOA shell. The experiment showed that CEorg decreased from 0.882 to 0.313-0.475 37 when  $f_{44}$  increased from 0.127 to 0.209, possibly caused by a phase change of SOA from liquid to solid/glassy. The variation of  $CE_{org}$  with  $f_{44}$  could explain a large fraction of the observed decrease in RRF, while the influence of 38 39 RIEorg cannot be excluded. The trends in RRF and CEorg for Q-ACSM were in accordance with those done with the 40 high-resolution aerosol mass spectrometer (AMS), demonstrating the capability of the low-resolution Q-ACSM in 41 doing SOA laboratory studies and the necessity to calibrate RF when conducting laboratory SOA experiments.

42

43 Key words: Organic Aerosols; ACSM Calibration; Relative Response Factor; Collection Efficiency; Average
44 Oxygen Content

45

**46 **1. Introduction**

47 Organic aerosols (OA) have been recognized as a major component of ambient particulate matters, 48 contributing 20-90% to the total submicron particles around the world (Kanakidou et al., 2005; Salcedo et al., 2006; 49 Zhang et al., 2007; Hallquist et al., 2009), which can substantially affect the climate directly by interacting with 50 solar radiation and indirectly by affecting cloud microphysics (IPCC, 2014). Recent studies demonstrated that haze 51 events in China were largely driven by secondary organic aerosols (SOA) (Huang et al., 2014) and may exert 52 significant adverse effects on human health (Poschl, 2005; Poschl and Shiraiwa, 2015). The chemical composition 53 and the dramatic changes of OA in ambient aerosol remain less understood compared to the inorganic species. 54 Measurement of OA has been a challenging task not only for the fact that OA contains nearly countless chemical 55 species but also due to its relatively short lifetime and rapid transformation in the atmosphere.

56 A powerful method, known as aerosol mass spectrometry (AMS) has been successfully developed and 57 widely deployed to do real time size-resolved aerosol measurements (Jayne et al., 2000). A typical AMS is equipped 58 with a set of aerodynamic lens (Liu et al., 1995a, b) to effectively focus and transmit particles (~50-1000 nm) (PM1) 59 into the instrument, a time-of-flight (ToF) chamber to determine the aerosol size, a thermal vaporizer (heated to 60  $\sim 600^{\circ}$ C) to evaporate non-refractory (NR) components into the gas-phase, and a 70 eV electron impact (EI) 61 ionization source to ionize the gaseous samples before they can be analyzed by a mass analyzer (either a quadrupole 62 or a time-of-flight mass spectrometer) (Jayne et al., 2000; DeCarlo et al., 2004). Evidently, compared to previous 63 filter-based techniques, AMS can provide elemental composition of the organic species and can achieve much 64 higher time and size resolution. More recently, a newer version of AMS, i.e., the aerosol chemical speciation 65 monitor (ACSM) was developed (Ng et al., 2011a). ACSM is basically a simplified AMS without the aerosol ToF 66 chamber and thus is much smaller and affordable. Although ACSM cannot obtain high resolution mass spectra, 67 elementary information can still be readily recovered from some symbolic fragments, such as  $C_n H_{2n\pm 1}^+$  (*m/z* 27, 29, 41, 43, 55, 57, 69...) and  $CO_2^+$  (*m/z* 44), representing hydrocarbon-like organic aerosol (HOA) and oxygenated 68 69 organic aerosol (OOA) (Ng et al., 2011a). Specially,  $f_{44}$  is widely used to reflect the average oxygen content in OA 70 (Ng et al., 2010; Ng et al., 2011b; Canagaratna et al., 2015). Therefore, ACSM is especially suitable for long term field operation. Nevertheless, Aerodyne AMS/ACSM have been widely used to conduct researches on NR-PM1
around the world in both field and chamber studies (Zhang et al., 2007; Mohr et al., 2009; Zhou et al., 2016).

73 Although the performance of AMS/ACSM has been demonstrated to be in accordance with many other 74 measuring techniques (Drewnick et al., 2003; Takegawa et al., 2005; Kondo et al., 2007; Jimenez et al., 2016), the 75 quantification of aerosol composition by AMS/ACSM still needs to be further refined. In theory, the accuracy of 76 AMS/ACSM measurements is affected by particle collection efficiency (CE), molecule ionization efficiency (IE), 77 fragmentations during thermo-vaporization and EI ionization (70 eV) processes, and ion transmission efficiency 78 inside the mass analyzer (Jayne et al., 2000; Allan et al., 2003; Jimenez et al., 2003; Canagaratna et al., 2015). With 79 the last two influencing factors well understood and calibrated, response factor (RF) of AMS/ACSM to a specific 80 species is mainly determined by CE and IE. In practice, AMS calibrations in terms of CE and IE were mostly 81 conducted with inorganic species. The calibration factors for organics were indirectly inferred and could be highly 82 uncertain for laboratory generated SOA (Jimenez et al., 2016). Typically, RF for OA (RForg) is determined indirectly 83 using ammonium nitrate as reference and could be mathematically expressed as the product of three factors:

84

$$RF_{org} = IE_{NO3} \cdot RIE_{org} \cdot CE_{org}$$
(1)

i.e.,  $IE_{NO3}$  is the ionization efficiency of nitrate;  $RIE_{org}$  is the relative ionization efficiency of OA to nitrate and  $CE_{org}$ is the collection efficiency of OA.  $IE_{NO3}$  can be calibrated with SMPS and is independent from the properties of OA. However, the other two terms may vary with different OA species.

CE is mainly composed of three parts:  $E_L$  (the transmission efficiency of particles as a function of vacuum aerodynamic diameter ( $d_{va}$ ) through the aerodynamic lens),  $E_s$  (to account for the disperse loss of non-spherical particles from the main air stream),  $E_b$  (to account for the bouncing loss of particles from the vaporizer surface). Among these,  $E_L$  can be explicitly taken into account by applying transmission efficiency in the aerodynamic lens,  $E_s$  is usually near unit for most particle types according to previous beam width measurements, and  $E_b$  is usually the most variable and uncertain part (Huffman et al., 2005; Salcedo et al., 2007). Based on inter-comparisons in sulfate measurements among AMS, particle-into-liquid-sampler/ion chromatography system (PILS-IC) and other

95 instruments, a default CE of 0.5 for all compounds has been recommended with the assumption that particles are 96 internally mixed (Drewnick et al., 2003; Takegawa et al., 2005). Although organics quantified using a default CEorg 97 of 0.5 has been found correlating well with independent organic carbon (OC) or VOC measurements in most field 98 works (Allan et al., 2004; de Gouw et al., 2005; Takegawa et al., 2005; Venkatachari et al., 2006; Middlebrook et 99 al., 2012), studies have shown that CEorg can vary substantially for various chamber generated organic particles (Bahreini et al., 2005; Docherty et al., 2013). Similarly, OA is assigned with a default RIEorg of 1.4. However, RIEorg 100 101 has been suggested to be significantly different for different OA species (Jimenez et al., 2016; Murphy, 2016). For 102 instance, it appears that RIEorg for primary OA (POA) are significantly different from those for SOA (Slowik et al., 103 2004; Dzepina et al., 2007; Jimenez et al., 2016). Any uncertainty associated with RIEorg may lead to erroneous 104 AMS/ACSM measurement results, especially in the case of interpreting aerosol samples from various environments 105 around the world. Therefore, comprehensive researches on the CEorg and RIEorg for different SOA species are of 106 practical importance to constrain the AMS/ACSM measurements.

Some methods have been introduced to quantify or constrain the side-effects of RIEorg and CEorg in AMS 107 108 measurements. For example, light scattering-based detections make it possible to directly measure aerosol CE 109 (Cross et al., 2007; Docherty et al., 2013; Robinson et al., 2017; Xu et al., 2018). However, the laser can only be 110 used for larger particles ( $d_{va} > 200$  nm). Most recently, a new type of capture vaporizer has been developed to achieve a unit CE (Hu et al., 2017). RIEorg can be directly measured with this capture vaporizer (Xu et al., 2018). 111 112 Nevertheless, the determination of RIEorg and CEorg appears to be a complicated but essential issue for accurate 113 AMS/ACSM OA measurements that necessitate further investigations, including to explore new ways of SOA 114 standard generation and to constrain the measurement uncertainty with various measurement techniques.

In this work, RF of a quadrupole based ACSM (Q-ACSM) for chamber-generated SOA under atmospheric relevant conditions was investigated. Three different VOC precursors (i.e.,  $\alpha$ -pinene, isoprene, and toluene) were chosen to represent biogenic ( $\alpha$ -pinene and isoprene) and anthropogenic (toluene) VOC. The mass concentrations of the SOA standards were precisely determined by a SMPS-APM system according to the first principle. RForg and CEorg of Q-ACSM to SOA with different average oxygen content were quantified by comparing Q-ACSM measured mass concentrations with SMPS-APM deduced mass concentrations for SOA produced by both homogeneous nucleation of organics or condensation of organics on dry AS seed particles. The effects of SOA average oxygen contents, roughly indicated by  $f_{44}$ , on RForg and CEorg of the Q-ACSM were investigated.

**123 2. Experimental Methods**

**124 2.1 Chamber Setup**

A 4.5 m3 collapsible atmospheric-pressure fluoropolymer (Teflon) smog chamber (L = 1.8 m; W = 1.5 m; 125 H = 1.7 m (see Fig. 1) was used to generate SOA. The chamber was essentially the same as the one used in our 126 127 previous work and has been described in detail elsewhere (Yao et al., 2014; Yuan et al., 2017). Before each 128 experiment, the chamber was flushed with pure air generated by a zero-air generator (Aadco 737, USA) until the particle concentration was less than 10 cm-3. Reactants and scavengers (if used) were introduced into the chamber 129 130 through a T-shaped glass bulb by zero air. The glass bulb was heated gently to ensure complete injections. A Teflon-131 coated fan was installed at the bottom center of the smog chamber. It was operated for 15 seconds after all reactants 132 were injected to the chamber to achieve rapid mixing. The chamber temperature and RH were maintained at  $20 \pm$ 1°C and 10-15%, respectively. 133

All the detection instruments were sampling from the same outlet which was located in the opposite side of the inlet. The sampling tubing of SMPS, APM and Q-ACSM were kept at the same length to achieve identical diffusion losses. During the experiments, no zero air was injected into the chamber and the amount of air in the chamber would decrease over the time. The total sampling flow rate was ~1.6 liters per minute (LPM). The experiments usually lasted for 4-7 hours, resulting the total consumption of chamber air of ~0.7 m3. Therefore, a constant pressure inside the collapsible Teflon bag could be maintained during the experiments.

**140 2.2 SOA Generation**

During each experiment, known amount of α-pinene (Sigma-Aldrich, >98%), isoprene (Sigma-Aldrich,
 >99%), or toluene (Sigma-Aldrich, >99.5%) was firstly dissolved into cyclohexane (TEDIA Inc., HPLC grade

143 >99.5%) and then was injected into the chamber through a stream of zero air. Ozone was generated by exposing 144 pure oxygen (O2) to a low-pressure mercury (Hg) lamp (Jelight, Model 600). When O3 was used as the oxidant, an 145 OH radical scavenger, cyclohexane, was injected into the chamber before the experiment. When OH was used as 146 the oxidant, trace amount of methyl nitrite was firstly injected into the chamber and then the black light bulbs around 147 the chamber were turned on (Yao et al., 2014). The mixing ratios of O3 inside the chamber were monitored 148 continuously by a Thermo Fisher Scientific ozone monitor (Model 49i) throughout each experiment. Different  $f_{44}$ 149 were achieved by changing the ratio between the VOC precursor and the oxidants (O3 or OH).

**150 2.3 Instrument Operation**

151 The sampling interval of the Q-ACSM was set to be 8-15 minutes, depending on the particle mass 152 concentration generated in the chamber. The sampling line was a piece of stainless steel tubing with an inside 153 diameter (ID) of ~0.4 cm and a length of 0.8 m. The vaporizer temperature of the Q-ACSM was set to be  $585 \pm 2$ 154 °Cwhich was within the recommended operation temperature range for aerosol measurements (Jimenez et al., 2003). 155 The sampling flow rate, aerodynamic lens alignment, m/z calibration, m/z-dependent ion transmission efficiency 156 of the quadrupole mass spectrometer were well calibrated by following the manufacture's recommendation (see 157 Supplementary Information for details). The lens pressure was kept at  $1.22 \pm 0.02$  throughout the experiment. The 158 response factor of NO3- (RFNO3) was calibrated before the experiment using atomized 300 nm NH4NO3 particles 159 following the procedure described in previous studies (Fig. S1) (Jayne et al., 2000; Ng et al., 2011a). EL as a function 160 of  $d_{va}$  was calibrated with 200-600 nm ( $d_m$ ) NH4NO3 particles (Fig. S2).

161 A scanning mobility particle sizer (SMPS), consisting of a differential mobility analyzer (DMA, TSI Model 162 3081) and an ultrafine condensation particle counter (UCPC, TSI Model 3776), was used for real-time monitoring 163 of the particle number size distribution with mobility diameter ( $d_m$ ) ranging between 15 nm and 650 nm. The 164 sampling time interval was 3 min. An aerosol particle mass analyzer (APM, Kanomax Model 3601) combined with 165 a DMA (TSI Model 3081) and an ultrafine condensation particle counter (UCPC, TSI Model 3776) were used to 166 determine the particle effective density ( $\rho_{eff}$ ) (McMurry et al., 2002; DeCarlo et al., 2004). During each experiment,

167 the diameter of the particles analyzed by APM was manually set to be the same as the peak value of the particle 168 number size distribution measured by the SMPS, as the particles were continuously growing in the chamber. Before 169 applying this measured  $\rho_{eff}$  to the whole size distribution,  $\rho_{eff}$  of particles in all sizes at the same moment were 170 assumed to be the same. Each DMA-APM scan took ~5 min. The flow rates and the voltages applied to the DMAs were well calibrated before the experiments. Three sizes of polystyrene latex (PSL) spheres ( $81 \pm 3$  nm,  $147 \pm 3$ 171 nm,  $269 \pm 5$  nm) with a density of 1.05 g cm-3 were nebulized with an atomizer (TSI, Model 3076) and then dried 172 by a diffusion dryer filled with silica gel for size calibration of the SMPS system. The same PSL spheres were also 173 174 used for the calibration of APM (see Fig. S3). The bias of SMPS size and APM effective density measurements 175 were within  $\pm 4\%$  and  $\pm 3\%$ , respectively.

**176 2.4 Calculation Method**

To elucidate how Q-ACSM measured mass concentration ( $PM_{ACSM}$ , using default  $RIE_{org} = 1.4$  and  $CE_{org} =$ 0.5) may deviate from the true mass concentration for particles entered Q-ACSM ( $PM_{SMPS}$ , the product of  $E_L$  and  $E_S$ -amended SMPS-measured volume concentration and APM-measured  $\rho_{eff}$ ), relative response factor (RRF) is defined as the ratio of  $PM_{ACSM}$  to  $PM_{SMPS}$ . The calculated RRF is thus proportional to the real  $RIE_{org}$  and  $CE_{org}$ . Any non-unit value of RRF might be caused by the deviation of  $RIE_{org}$  or  $CE_{org}$  from the default values.

182
$$RRF = \frac{PM_{ACSM}(assuming \ def \ ault \ RIE_{org} = 1.4, CE_{org} = 0.5)}{PM_{SMPS}(amended \ by \ \rho_{eff}, Es, E_L)} = \frac{RIE_{org} \cdot CE_{org}}{1.4 \cdot 0.5}$$
(2)

183
$$PM_{SMPS} = V_{SMPS} \cdot \rho_{eff} \cdot E_L \cdot E_s = \sum_{i=1}^n \{\frac{\pi}{6} d_{m,i}^3 \cdot dN_i \cdot \rho_{eff,i} \cdot E_{L,i} \cdot E_{s,i}\}$$
(3)

Eq. (3) is used to calculate  $PM_{SMPS}$ , where  $V_{SMPS}$  is the volume concentration calculated with SMPS measured size distribution; n is the total number of bins for SMPS measurements;  $d_{m,i}$  and  $dN_i$  are respectively the geometric mean diameter and the number of counts in size bin i;  $E_{L,i}$  and  $E_{S,i}$  are the aerodynamic lens transmission efficiency and the non-spherical shape influence for particles of  $d_{m,i}$ . After taking into account the effects of  $E_{L,i}$  and  $E_{S,i}$ , any deviation in  $CE_{org}$  should be caused only by particle bounce on the vaporizer surface ( $E_b$ ). The measured  $E_L$  in this work (Fig. S2) is in reasonable agreement with previous studies (Jayne et al., 2000; Knote et al., 2011; Hu et al., 2017), while larger than other reports (Liu et al., 2007) as for 300-1000 nm ( $d_{va}$ ) particles.  $d_m$  and  $d_{va}$  are inter-convertible via  $d_{va} = d_m \cdot \rho_{eff} / \rho_0$  for a reference density  $\rho_0$  of 1000 kg m-3 (Jayne et al., 2000; DeCarlo et al., 2004). The uncertainties of  $d_{m,i}(\pm 4\%)$ ,  $dN_i(\pm 10\%$  for 20-200 nm and  $\pm 20\%$  for 200-800 nm particles) (Wiedensohler et al., 2012; Wiedensohler et al., 2018),  $\rho_{eff}$  ( $\pm 3\%$ ),  $E_L$  ( $\pm 10\%$ ) and  $E_s$  ( $\pm 5\%$ ) were all considered in the calculation. As a result, the relative uncertainties of  $PM_{SMPS}$  were between 19.4-26.0%, depending on the actual size distribution. Details of the uncertainty analysis can be found in Section 2 of the Supplementary Information.

**196 **3. Results and Discussion**

**197 3.1 RRF for Chamber Generated SOA**

198 The evolution of SOA during a typical chamber experiment (Exp. 1) is shown in Fig. 2, where 80 ppby  $O_3$ 199 and 20 ppby  $\alpha$ -pinene were injected into the chamber. Shortly after injection, a strong nucleation event occurred 200 and was marked by a typical banana-shaped number size distribution plot (Fig. 2a). As the ozonolysis reaction 201 proceeded, PMACSM, PMSMPS, particle sizes, and peff swiftly increased during the initial two hours (Fig. 2b). The initial high values of  $f_{44}$  could partially due to  $f_{44}$  calculation bias in low concentration (0-5 µg m-3) or the fact that 202 203 initially formed SOA particles were highly oxidized because of gas to particle partitioning (Shilling et al., 2009). 204 PMACSM kept slightly higher than PMSMPS, indicating an over-unit RRF. After 2-3 hours, the gas phase reaction and 205 the gas to particle phase partition slowed down, a "relative stable state" was reached. During the "relative stable state", PMSMPS slightly increased from  $\sim$ 32 µg m-3 to  $\sim$ 37 µg m-3,  $\rho_{eff}$  slightly increased from  $\sim$ 1.17 to  $\sim$ 1.22 g cm-3, 206 207  $f_{44}$  fluctuated between ~0.118 and ~0.134, indicating that both the SOA mass and the composition were relative 208 stable. Therefore, an averaged RRF (1.32  $\pm$  0.30) and  $f_{44}$  (0.126  $\pm$  0.005) within the "relative stable state" were 209 obtained.

Following the similar procedure, fourteen independent chamber experiments were conducted in this work Table 1 listed the detailed experiment conditions. The mixing ratios of the VOC precursors ranged from 10 to 200 ppbv. The oxidant concentrations were 50-80 ppbv for O3 and 50-300  $\mu$ L for methyl nitrite. Each experiment typically lasted for 4-7 hours. In each experiment, a "relative stable state" was defined as  $f_{44}$ ,  $\rho_{eff}$ , and mass concentration all reached relatively constant values and the average values within the "relative stable state" were used for analysis. Evidently, the characteristics of the generated SOA appeared to be significantly different among various oxidation conditions as for the generated wide ranges of mass concentrations (7.8-137.0  $\mu$ g m-3) and  $f_{44}$ (0.057-0.191).

RRF for the chamber-generated SOA decreased from 2.16 to 0.81 as  $f_{44}$  increased from 0.057 to 0.191 (see Fig. 3a). Although nearly half of the points were still within the measurement uncertainties of Q-ACSM for organics (±38%) (Bahreini et al., 2009; Middlebrook et al., 2012), RRF were higher than unit for most of the points, especially for the three points when  $f_{44} < 0.10$ . The highest two points were from isoprene oxidation and the third highest was from toluene oxidation. The over-unit RRF indicated that either CEorg or RIEorg, or both of them were higher than the default values (1.4 and 0.5, respectively). RRF value was slightly lower than unit for one of the toluene oxidation experiment, which might indicate lower CEorg or RIEorg values.

225  $\rho_{eff}$  increased from 1.09 to 1.34 g cm-3 with increasing  $f_{44}$ , as shown in Fig. 3b. Although our APM 226 calibration with standard PSL spheres indicated some systematic positive deviations for larger particles, the ±3% 227 systematic errors could not explain such big positive trend observed in Fig. 3b. Similarly, George and Abbatt (2010) 228 observed an increasing trend of  $\rho_{eff}$  during simulated atmospheric aging of  $\alpha$ -pinene oxidized SOA, where  $f_{44}$ 229 increased substantially.  $\alpha$ -pinene oxidation experiments conducted by Kuwata et al. (2011) also indicated that  $\rho_{eff}$ 230 increased from 1.23 to 1.46 g cm-3 as oxygen to carbon ratio (O/C) increased from 0.38 to 0.48. In short, it appeared 231 that organics with higher oxygen contents tended to be more compact.

**232 *3.2 Effects of CE**

To investigate and evaluate the contribution of variation of  $CE_{org}$  to the observed anti-correlations between RRF and  $f_{44}$  in this work, a set of chamber experiments using  $(NH_4)_2SO_4$  (AS) as seed particles were conducted. The difference between this set of experiments from the previous experiments was that dry AS seed particles were injected into the chamber with an atomizer (model 3076 TSI) followed by a diffusion dryer before SOA were generated. After AS concentrations were stable in the chamber, VOC and the oxidants were injected. The gas phase oxidation products condensed onto the AS seeds instead of initiating new particle formations. As soon as AS seeds were coated with SOA, the Q-ACSM measured sulfate mass concentrations would change. A default RIESO4 (1.15) was used here as suggested by previous studies (Canagaratna et al., 2007; Ng et al., 2011a; Petit et al., 2015). As RIESO4 would not change after coated with SOA, any changes in sulfate signal was due to variations in CEAS (Matthew et al., 2008). After AS particles were fully coated with SOA shell, CEAS was equal to CEorg for the pure SOA particles (Bahreini et al., 2005; Docherty et al., 2013).

244 Figure 4 displayed the evolution of a typical AS coating experiment (Exp. 19), where 80 ppbv toluene and 245 150 µL methyl nitrite were used to generate SOA. The experiment started when AS was first injected into the chamber. The initial geometric mean diameter ( $d_g$ ) of AS seed particles in the chamber were 70-80 nm.  $\rho_{eff}$  of the 246 dry 80 nm AS particles was measured to be  $1.64 \pm 0.08$  g cm-3, which was consistent with 1.65 g cm-3 (Zelenyuk et 247 al., 2006) or slightly lower than 1.66-1.70 g cm-3 measured for 200 nm ( $d_m$ ) dry AS particles in a previous study 248 (Matthew et al., 2008). Initially, the Q-ACSM measured sulfate mass concentrations were  $\sim$ 35 µg m-3. The initial 249 250  $CE_{AS}$  was calculated to be 0.277 ± 0.066. An hour later, toluene and methyl nitrite were injected and the black light 251 bulbs were turned on. The Q-ACSM measured sulfate signal increased substantially as SOA gradually condensed 252 onto AS seeds, implying that the SOA cover caused  $CE_{AS}$  to gradually increase as coating layer became thicker. 253 Two hours after the experiment started, the Q-ACSM measured organic concentration was still increasing but the 254 measured sulfate stopped increasing and kept constant during the rest of the experiment. By the time when sulfate signal stopped increasing,  $d_g$  increased to 115 nm, the average coating thickness (growth in particle radius) was 255 256 about 18 nm assuming uniform coating. This indicated that an 18 nm SOA shell was already enough to cover the 257 80 nm seed particles in this experiment. Similarly,  $f_{44}$  did not change significantly as SOA increased, indicating 258 nearly identical particle chemical compositions during the entire experiment. The period when both sulfate and 259 organic concentration reached relative constant was considered as the "relative stable state" in this experiment (area 260 between Arrow 3 and Arrow 4 in Fig. 4). Points after Arrow 4 were obviously affected by wall losses and thus were 261 not included in the calculation. Average PMACSM of AS in "relative stable state" divided by average PMSMPS of AS

was used to calculate the CEAS after coating with SOA. Since AS was fully covered, CEorg should be equal to CEAS, which was  $0.441 \pm 0.110$  for an average  $f_{44}$  value of  $0.209 \pm 0.011$ .

264 Eight experiments using AS seeds were conducted in total (see Table 2). In each experiment, similar mass concentrations (76.0-98.2  $\mu$ g m-3) and sizes ( $d_g$  = 70-80 nm) of AS seeds were injected into the chamber. The slightly 265 different initial AS mass concentrations among different experiments were caused by the different initial AS seed 266 267 number concentrations.  $\rho_{eff}$  of AS measured in all experiments were very much the same as in Exp. 19, with an average of  $1.64 \pm 0.04$  g cm-3. CEAS for dry AS particles varied from 0.265 to 0.298 in different experiments, with 268 269 an average value of  $0.285 \pm 0.067$  which was slightly higher than the reported  $0.24 \pm 0.03$  (Matthew et al., 2008). The bias could come from the large uncertainties of SMPS measurement or the bias from the default  $RIE_{SO4}$  (1.15). 270 271 For most of the experiments, all condensable organic vapor seemed to condense directly onto seed particles instead 272 of forming new particles. While in one experiment (Exp. 22), there were small amounts of new particles formed, 273 which was indicated by a smaller mode on the size distribution (Fig. S6) and further confirmed by a sudden increase 274 in total particle number concentration. However, the newly formed particles only contributed slightly to the total 275 particulate mass for their much smaller sizes. In addition, to avoid the influence of new particle formation on CEorg or RIEorg deduction, only PMACSM and PMSMPS associated with AS signals were used to deduce the actual CEorg in 276 277 our experiments.

278 Figure 5 showed the variation of  $CE_{org}$  as a function of  $f_{44}$ . For most of the experiments, the SOA coating 279 layer was thick enough (18-25 nm) to cover AS seed particles. This was supported by the fact that after coating with 280 18-25 nm SOA, the Q-ACSM measured sulfate concentration stopped increasing with further organic increase. 281 While CEorg from Exp. 16, 20, and 21 (coating layer < 18-25 nm) might be the result of a combination of AS and organic particles because the Q-ACSM measured sulfate increased all the way until organics stopped increasing. 282 283 For  $\alpha$ -pinene initiated SOA, high CEorg (0.882 ± 0.207) was observed when  $f_{44} = 0.127 \pm 0.003$  at the beginning.  $CE_{org}$  decreased sharply to 0.313 ± 0.076 when  $f_{44}$  increased to 0.147 ± 0.009, then kept at 0.338 as  $f_{44}$  further 284 increased to 0.202. For toluene initiated SOA,  $CE_{org}$  varied between 0.441 and 0.475 when  $f_{44}$  varied between 0.190 285

and 0.219, which was the highest level of  $f_{44}$  obtained in this work. Nevertheless, it could be observed that SOA with the lowest  $f_{44}$  (0.127) was associated with the highest CE, which largely explained the trend in RRF.

**288 3.3 Possible Effects of RIE**

290

289 The effects of RIEorg can be evaluated using Eq. 5 (Jimenez et al., 2003; Canagaratna et al., 2007):

$$RIE_{org} = \frac{MW_{NO3}}{IE_{NO3}} \cdot \frac{IE_{org}}{MW_{org}}$$
(5)

291 Where, MWNO3 and MWorg are the molecular weights of nitrate and a specific organic compound, respectively. 292 Theoretically, IENO3 is directly proportion to  $\sigma$ , the electron impact ionization cross section of the molecule, which 293 is linearly related to the number of electrons in the molecule for molecules with similar structure and functionality. 294 Since the number of electrons is roughly proportional to the molecular weight, RIEorg for molecules with similar structure and functionality are suggested to be similar to each other. RIEorg for hydrocarbons and oxygenated 295 296 species, thus, are believed to be different since their oxygen contents can vary substantially (Canagaratna et al., 297 2007). With our experiments, we still cannot rule out the contribution of RIEorg to the overall RRF trend even though 298 the variation of  $CE_{org}$  with  $f_{44}$  could largely explain the observed RFF changes.

**299 3.4 Comparison with AMS Calibrations**

300 Since the Q-ACSM has similar characteristics as an AMS (aerodynamic lens followed by thermal 301 vaporization, electron impact ionization, and MS detection), the calibration results obtained from this work are 302 expected to be comparable with those done with the AMS.

In previous studies conducted by AMS, CE was demonstrated to be variable for particles with different chemical composition, phase state and under different RH, with  $CE_{org}$  ranging from less than 0.15 to ~1 (Alfarra, 2004; Matthew et al., 2008; Docherty et al., 2013; Robinson et al., 2015). However, during field studies AMS measurement results based on default RIE = 1.14 and CE = 0.5 appeared to be in reasonable agreement with other aerosol measurement techniques (Slowik et al., 2004; Zhang et al., 2005; Middlebrook et al., 2012). CE for biogenic SOA has been reported to be close to 1 based on both chamber experiments (Kiendler-Scharr et al., 2009) and field 309 measurements conducted in Amazon, where aerosols were dominated by liquid SOA (Chen et al., 2009; Allan et 310 al., 2014). However, it has been proposed in theoretical, chamber, and field studies that besides liquid state, organic 311 particles can exist in semi-solid or solid state (Vaden et al., 2010; Virtanen et al., 2010; Shiraiwa et al., 2011), which 312 might lead to lower  $CE_{org}$ . For example,  $CE_{org}$  of liquid squalane (CE = 1) decreased after SOA condensed on the 313 particle (Robinson et al., 2015). Most recent CEorg calibration with light-scattering single-particle (LSSP) module 314 also suggested that CEorg (0.25-0.4) could be much less than 1 (Robinson et al., 2017). An inverse relationship 315 between CEorg and  $f_{44}/f_{57}$  ratio in chamber SOA generation experiments (Docherty et al., 2013) was very similar to 316 our results. However, our results could not be quantitatively compared with the previous ones because the Q-ACSM 317 might report comparably higher  $f_{44}$  than the AMS (Fröhlich et al., 2015).

318 One possible explanation of the observed decreasing  $CE_{org}$  with  $f_{44}$  was the phase change. Particle 319 morphology research based on glass transition temperature  $(T_{e})$  indicated that during oxidation, the SOA products 320 might vary between liquid state, semi-solid state and solid (glassy) state, resulting from a combined effect of 321 increasing molecular weight and O/C ratio (Koop et al., 2011). As a result, it was reasonably to assume that HOA particles were initially "sticky" liquid droplets with CEorg close to 1. As O/C ratio increases during oxidation 322 process, the liquid droplets changes into "bouncing" semisolid or solid ball with sharp or slow decrease in CEorg. 323 324 As the phase changes were not only determined by oxygen contents but also the molecular weight that was related 325 to the precursor gases, i.e., CEorg changes might be different for different VOC precursors. The phase change 326 speculation is partially supported by the positive relationship between  $\rho_{eff}$  and  $f_{44}$ .

Previous studies have shown that RIEorg ranged from less than 1 for SOA to more than 3 for HOA (Slowik et al., 2004; Dzepina et al., 2007; Jimenez et al., 2016). A most recent research showed that RIEorg for hydrocarbonlike organic aerosol might vary from 2 to 7. While RIEorg for organic with  $-1.0 < \overline{OSc}$  (average carbon oxidation states) < 0.5 varied between  $1.6 \pm 0.5$  ( $2\sigma$ ) (Xu et al., 2018), which is relatively higher than the widely used default value (RIEorg = 1.4). Clearly, a higher than default RIEorg value is consistent with the overall positive deviations of RRF observed in our experiments.

**333 4. Conclusions**

334 The quantification of SOA using a Q-ACSM has been investigated in a photochemical chamber. We found 335 an anti-correlation between RRF, defined as PMACSM divided by PMSMPS, and SOA f44, regardless of the type of VOC precursors. RRF decreased from 2.16 to 0.81 as  $f_{44}$  increased from 0.057 to 0.191, while  $\rho_{eff}$  increased from 336 1.09 to 1.34 g cm-3. In SOA coating experiments, an SOA coating layer with a thickness of 18-25 nm (in radius) 337 was found to be sufficient to fully cover the AS seed particles with mobility diameters of 70-80 nm. Collection 338 efficiency of organics-coated aerosols (CEorg) was found to decrease from 0.882 to 0.313-0.475 as  $f_{44}$  increased 339 340 from 0.127 to 0.174-0.210. Decreasing CEorg with  $f_{44}$  might be explained by the phase transition from liquid-state 341 into more solid-like/glassy state of SOA particles consisting of more oxidized organic compounds that normally 342 assume a higher  $\rho_{eff}$ . Although RIEorg was not directly measured in our experiments, it seemed that the variation of  $CE_{org}$  with  $f_{44}$  could largely explain the RFF changes with  $f_{44}$ , indicating that the effect of  $f_{44}$  on RIEorg might be 343 344 limited for SOA.

Our results strongly indicate that chamber measurements of OA with various  $f_{44}$  by a Q-ACSM using constant conversion factors may induce significant errors in aerosol mass concentration measurements. The exact values of these factors most likely will vary with different individual instrument. Accordingly, the Q-ACSM response factor for OA should be systematically calibrated with laboratory-generated SOA with various average oxygen contents. It is also important that quantification of laboratory and ambient OA by a Q-ACSM with size distribution information should include measurements of effective density instead of assuming a constant value.

Due to the low-cost and durability, Q-ACSMs have been widely used in both ambient and laboratory experiments to quantify aerosol chemical compositions, especially for long-term observations in highly polluted regions (such as in China). However, the performance of Q-ACSM in organics measurement was rarely evaluated because of its low time and m/z resolution and the lack of particle size selection. To the best of our knowledge, this study represents the first comprehensive chamber calibration of  $CE_{org}$  and  $RRF_{org}$  for a Q-ACSM. Our results of RRF, and  $CE_{org}$  calibration were generally in line with previous studies conducted with the AMS. Although the

| 357        | results of this work were not directly applicable to ambient Q-ACSM measurements, it is suggested that more                                                                                                                     |
|------------|---------------------------------------------------------------------------------------------------------------------------------------------------------------------------------------------------------------------------------|
| 358        | accurate and comprehensive calibrations of Q-ACSM for complicated ambient conditions shall be done to further                                                                                                                   |
| 359        | lower the uncertainties of OA measurement, which is still relatively high (±38%) currently.                                                                                                                                     |
| 360        | Acknowledgements                                                                                                                                                                                                                |
| 361        | This work was supported by the National Key Research and Development Project (2016VEC0202402)                                                                                                                                   |
|            | This work was supported by the reational Key Research and Development Project (201011 C0202402),                                                                                                                                |
| 362        | National Natural Science Foundation of China (41575122, 41675126, and 41730106), and the Priority Academic                                                                                                                      |
| 362
363 | National Natural Science Foundation of China (41575122, 41675126, and 41730106), and the Priority Academic
Program Development of Jiangsu Higher Education Institutions. The data used in this work are listed in the tables |

| Precursor | #  | VOC(ppbv) | O 3 /OH [1] (ppbv or
μL) | PM SMPS (µg m -3 ) [2] | $ ho_{eff}$ cm -3 ) [3] | f 44 [4] | RRF [5] |
|-----------|----|-----------|---------------------------------------------------|---------------------------------------------------------|-----------------------------------------------|--------------------------------|--------------------|
|           | 1  | 20        | 80 ppbv                                           | 35.9±7.2                                                | 1.19±0.05                                     | 0.126±0.005                    | 1.32±0.30          |
| a-pinene  | 2  | 60        | 75 μL                                             | 106.9±25.6                                              | 1.25±0.04                                     | 0.136±0.005                    | 1.49±0.37          |
| a pinene  | 3  | 10        | 80 ppbv                                           | 9.7±2.23                                                | 1.29±0.05                                     | 0.146±0.018                    | 1.31±0.37          |
|           | 4  | 60        | 60 ppbv                                           | 9.6±1.92                                                | 1.25±0.05                                     | 0.152±0.011                    | 1.23±0.30          |
|           | 5  | 30        | 75 μL                                             | 15.6±3.1                                                | 1.19±0.05                                     | 0.146±0.012                    | 0.99±0.23          |
| toluene   | 6  | 60        | 150 μL                                            | 38.2±7.6                                                | 1.34±0.04                                     | 0.191±0.012                    | 0.81±0.19          |
|           | 7  | 30        | 50 µL                                             | 14.2±2.8                                                | 1.10±0.04                                     | 0.079±0.010                    | 1.58±0.38          |
|           | 8  | 60        | 75 μL                                             | 10.6±2.8                                                | 1.24±0.05                                     | 0.127±0.006                    | 1.45±0.40          |
|           | 9  | 200       | 300 µL                                            | 137.0±32.9                                              | 1.09±0.04                                     | 0.057±0.003                    | 2.16±0.51          |
|           | 10 | 100       | 150 μL                                            | 15.1±3.2                                                | 1.11±0.05                                     | 0.080±0.025                    | 1.81±0.43          |
| isoprene  | 11 | 60        | 90 µL                                             | 7.8±1.6                                                 | 1.26±0.05                                     | 0.138±0.031                    | 1.42±0.34          |
|           | 12 | 80        | 120 µL                                            | 10.2±2.0                                                | 1.20±0.05                                     | 0.129±0.023                    | 1.47±0.33          |
|           | 13 | 160       | 60 µL                                             | 20.6±4.5                                                | 1.22±0.04                                     | 0.160±0.005                    | 1.28±0.31          |
|           | 14 | 200       | 60 µL                                             | 36.7±7.3                                                | 1.28±0.04                                     | 0.157±0.006                    | 1.47±0.30          |

**593 Table 1. Summary of SOA Chamber Experiments without Seed Particles**

[1] The amount of OH is expressed as the amount of methyl nitrite (μL) injected into the chamber.

 $^{[2]}$  PMSMPS was calculated from the average sized distribution (Fig. S4) in "relative stable state", PMSMPS was amended by EL of ACSM aerodynamic lens (Fig. S2) and Es (100 ± 5%). Details of the uncertainties calculation can be found in SI.

597 [3-4]  $\rho_{eff}, f_{44}$  were the average values in "relative stable state", uncertainties were standard deviations of all the points during "relative stable state".

598 [5] RRF was calculated with the average PMSMPS and PMACSM in "relative stable state". Details of the uncertainties calculation can be found in SI.

| Precursor | #  | VOC
(ppbv) | O 3 /OH
(ppbv or
μL) | PM ACSM-AS1 [1]
(μg m -3 ) | PM ACSM-AS2 [2]
(μg m -3 ) | PM SMPS-AS [3]
(μg m -3 ) | SOA
thickness [4]
(nm) | f 44       | CE AS [5] | CE org [6] |
|-----------|----|---------------|---------------------------------------|----------------------------------------------------------------|----------------------------------------------------------------|---------------------------------------------------------------|-----------------------------------------|-------------------|---------------------------------|----------------------------------|
|           | 15 | 20            | 50 µL                                 | 57.5±3.2                                                       | 66.0±4.9                                                       | 97.7±21.2                                                     | 18                                      | 0.202±0.018       | 0.294±0.068                     | 0.338±0.080                      |
| α-pinene  | 16 | 20            | 80 ppbv                               | 50.3±5.4                                                       | 56.2±6.6                                                       | 83.9±18.4                                                     | 15                                      | 0.193±0.023       | $0.299 \pm 0.074$               | (0.335±0.085)                    |
|           | 17 | 60            | 80 ppbv                               | 43.9±3.0                                                       | 48.4±4.7                                                       | 77.1±17.3                                                     | 47                                      | 0.147±0.009       | 0.284±0.067                     | 0.313±0.076                      |
|           | 18 | 60            | 50 µL                                 | 40.3±2.2                                                       | 134.2±8.7                                                      | 76.0±17.2                                                     | 73                                      | 0.127±0.003       | 0.265±0.061                     | 0.882±0.207                      |
|           | 19 | 80            | 150 μL                                | 51.2±3.4                                                       | 81.4±7.9                                                       | 92.3±21.3                                                     | 23                                      | 0.209±0.011       | 0.277±0.066                     | 0.441±0.110                      |
| toluene   | 20 | 80            | 100 µL                                | 55.6±4.5                                                       | 71.0±6.8                                                       | 98.2±22.1                                                     | 12                                      | 0.237±0.016       | $0.283 \pm 0.068$               | $(0.362 \pm 0.088)$              |
|           | 21 | 60            | 200 µL                                | 48.3±4.7                                                       | 57.5±5.3                                                       | 85.3±18.8                                                     | 20                                      | 0.210±0.013       | $0.283 \pm 0.068$               | $(0.337 \pm 0.080)$              |
|           | 22 | 80            | 200 µL                                | 53.2±1.8                                                       | 84.8±6.0                                                       | 89.2±20.0                                                     | 41                                      | $0.190 \pm 0.003$ | $0.298 \pm 0.068$               | 0.475±0.112                      |

**602 Table 2. Summary of SOA Chamber Experiments with AS Seed Particles**

603 [1,2] Average mass concentration of AS seed particles measured by ACSM (using default CE = 0.5, RIE = 1.4) before and after SOA was generated.

604 [3] Average mass concentration of AS particles measured by SMPS which was calculated using Eq. 3.

605 [4] Coating layer was calculated from geometric mean diameter before and after the coating (Fig. S5).

606 [5-6] CEAS and CEorg were calculated with Eq. (S7 & 8), uncertainties were evaluated with Eq. (S9 & S10). Wall losses during the SOA coating was ignored. CEorg values

607 with brackets were experiments that AS particles were not fully covered with SOA and CEorg is actually a combination result of organic shell and AS core.

Figure 1. Schematic of the collapsible atmospheric-pressure fluoropolymer (Teflon) smog
chamber and the instrument setup. AS generation unit was only used in the seed experiments.
During experiments, a total sample flow rate was 1.6 LPM, including 1.0 LPM for ACSM, 0.3 LPM
for DMA-APM-CPC, and 0.3 LPM for SMPS. No zero air was introduced into the chamber during
the experiments.

---

## Author Comment (AC6) · 1 Oct 2018

**1** Supplementary Information:**

| 2  | Response factor of a Q-ACSM to chamber generated SOA with different average oxygen content                                                                                                          |
|----|-----------------------------------------------------------------------------------------------------------------------------------------------------------------------------------------------------|
| 3  | Xiaoxiao Li 1 , Yan Ma 2,3* , Hui Chen 2,3 , Youling Jiang 2,3 , Xin Ma 2,3 , Rujin Yin 1 , Dongsen Yang 2,3 , Xiaowen |
| 4  | Shi 2,3 , Jiming Hao 1 , Jingkun Jiang 1 , and Jun Zheng 2,3*                                                                                           |
| 5  | 1 State Key Joint Laboratory of Environment Simulation and Pollution Control, School of Environment,                                                                                     |
| 6  | Tsinghua University, 100084 Beijing, China                                                                                                                                                          |
| 7  | 2 Collaborative Innovation Center of Atmospheric Environment and Equipment Technology, Nanjing University                                                                                |
| 8  | of Information Science & Technology, Nanjing 210044, China                                                                                                                                          |
| 9  | 3 Joint Laboratory for Air Quality and Climate, Nanjing University of Information Science & Technology,                                                                                  |
| 10 | Nanjing 210044, China                                                                                                                                                                               |
| 11 | Corresponding authors: Drs. Jun Zheng and Yan Ma                                                                                                                                                    |
| 12 | Email: zheng.jun@nuist.edu.cn and mayan@nuist.edu.cn                                                                                                                                                |
| 13 | Tel.: +86-18251919852                                                                                                                                                                        |
| 14 | Fax: +86-25-58731090                                                                                                                                                                                |
|    |                                                                                                                                                                                                     |

1 Calibration of instruments

**16 1.1 ACSM RFNO3, RIENH4 and EL Calibrations**

Figure S1. Instrument setup for Q-ACSM RFNO3 calibration and determination of the aerodynamic lens
 transmission efficiency.

The calibration of  $RF_{NO3}$  (which is also customarily called  $RIE_{NO3}$ ) was conducted by following the procedure recommended in by Jayne et al. (2000), Ng et al. (2011) and the manual of the Q-ACSM. Before the  $RF_{NO3}$  calibration, the sampling flow rate (0.85 LPM) of the Q-ACSM was calibrated with a Gilibrator-2 system (Sensdyne, USA). The nitrogen peak (m/z = 28) and the internal naphthalene standard peak (m/z = 128) were used to do m/z calibration before the experiments. The multiplier gain was set to 20,000 by adjusting the multiplier voltage. The sampling lines to the Q-ACSM and CPC were made as short as possible and were kept at the same length of 25cm. Dry NH4NO3 aerosols with a 300 nm diameter were generated using an atomizer (Model 3076, TSI) followed by a diffusion dryer and size-selected by a DMA. Lens alignment was done using the lens alignment tool prior to the calibration. Different concentrations of NH4NO3 aerosols were achieved by varying atomizer solution concentration (no more than 5 mM to minimize multiple charges) to do the RFNO3 calibration. A shape factor for NH4NO3 was set to be 0.8 as suggested by Jayne et al. (2000). As a result, RFNO3 and RIENH4 were determined to be  $3.56\pm0.06\times10^{-11}$ , and  $5.54\pm0.25$ , respectively.

The lens transmission efficient ( $E_L$ ) of the Q-ACSM was calibrated using the same setup as in Figure S1. The only difference was that the DMA was set to select a variety of particles with diameters ranging from 33 200nm to 600nm instead of a fixed single diameter of 300 nm. The lens pressure was kept at 1.22±0.02 34 throughout the experiment. To minimize multiple charge problem for larger particles, we conducted  $E_L$ 35 calibration under two sets of comparably low NH4NO3 solution concentrations (~2 mM and ~0.5 mM). 36 However, for the set of experiment with extremely low  $NH_4NO_3$  concentration (~0.5 mM), the Q-ACSM 37 38 sensitivity was not high enough to detect NH4NO3 particles, which therefore was not shown here. Consequently, the calibration result of the lens transmission efficiency was shown in Figure S2. 39

**Figure S2.** Lens transmission efficiency as a function of particle vacuum aerodynamic diameter (dva).

Lens transmission efficiencies for 300~600nm particle were nearly unity, which was in reasonable accordance with some previous studies (Jayne et al., 2000; Knote et al., 2011; Hu et al., 2018), but the diameter range with a unity lens transmission efficiency was much broader than that reported by Liu et al. (2007). The blue line in Figure S2 was used to modify size distribution in our experiments (i.e., 0% transmission for  $d_{va}$  below 40 nm; linear increase in transmission vs log( $d_{va}$ ), from 0% to 100% at  $d_{va}$ =100 nm; 100% transmission efficiency from  $d_{va}$ =100 nm up to  $d_{va}$ =550 nm; 100% transmission between 100 nm and 580 nm; linear decrease in transmission vs log( $d_{va}$ ) from 100% at 580 nm to 0% at 1200 nm). For EL below 49 300 nm, we used the results reported by Knote et al. (2011) indicated by the red dashed line in Figure S2. 50 About 10% of uncertainty (shaded area) was applied to the  $E_L$  used in this study as indicated by the blue curve 51 in Figure S2.

**52 **1.2 APM Calibration**

**Figure S3. Instrument Setup for APM calibration.**

The calibration of the APM followed the procedures of Ehara et al. (1996) and the APM manual (Figure 55 S2). Three sizes of polystyrene latex spheres (PSL),  $81 \pm 3$  nm,  $147 \pm 3$  nm, and  $269 \pm 5$  nm) were used for 56 size calibration of the the SMPS and mass calibration of the APM. Effective density of the standard PSL 57 spheres was 1.05 g cm-3, the measured values were  $1.046 \pm 0.005$  g cm-3,  $1.056 \pm 0.006$  g cm-3 and  $1.067 \pm$ 58 0.008 g cm-3 for PSL spheres with the size of 81 nm, 147 nm, and 269 nm, respectively. The results indicated 59 that the APM measured slightly lower values of effective density for smaller particles and slightly higher 60 values for larger particles. Nevertheless, the overall bias of the APM measurements were within  $\pm 3\%$  of the 61 62 standard values.

**63 2 Uncertainty Analysis**

In this work, the reported quantity was considered as a function (y) of measured variables (x1, x2, x3...xn)
in the form of f(x1, x2,..., xn). The uncertainties associated with measured x1, x2, x3...xn were σ1, σ2,...σn,
respectively. Accordingly, the mean value of y (y) and its uncertainty (σy) were calculated using Eq. (S1) and
Eq. (S2) as following:

$$\overline{y} = f(\overline{x_1}, \overline{x_2}, \dots, \overline{x_n}) \tag{S1}$$

$$\sigma_y = \sqrt{\left(\frac{\partial y}{\partial x_1}\right)^2 \sigma_1^2 + \left(\frac{\partial y}{\partial x_2}\right)^2 \sigma_2^2 + \dots + \left(\frac{\partial y}{\partial x_n}\right)^2 \sigma_n^2}$$
(S2)

**70 2.1 PMSMPS uncertainty**

$PM_{SMPS}$  was calculated using Eq. (3) in the main text. Theoretically, the uncertainty of  $PM_{SMPS}$  ( $\sigma_{PM-SMPS}$ )

was associated with  $d_{m,i}$ ,  $dN_i$ ,  $\rho_{eff,i}$ ,  $E_L(d_{m,i})$ , and  $E_{s,i}$ . As discussed in the instrument calibration in Section 1, uncertainties of  $d_{m,i}$  ( $\sigma_{dm}$ ) and  $\rho_{eff}$  ( $\sigma_{\rho}$ ) were within  $\pm 4\%$  and  $\pm 3\%$ , respectively. The uncertainty of EL ( $\sigma_{EL}$ ) in 73 the range of 40-600 nm ( $d_{va}$ ) was estimated to be ±10% according to Figure S2. The uncertainty of  $dN_i$  ( $\sigma_N$ ), 74 which was the combination of the uncertainties of particle charging efficiency ( $\sigma_{char}$ ), diffusion loss ( $\sigma_{diff}$ ), and 75 CPC counting efficiency ( $\sigma_{count}$ ) (Buonanno et al., 2009), was assigned to be ±10% for particles with a 76 diameter of 20-200 nm and  $\pm$  20% for 200-800 nm (Wiedensohler et al., 2012; Wiedensohler et al., 2018). The 77 78 uncertainty of  $E_s$  ( $\sigma_{ES}$ ) was estimated to be ±5%. The uncertainty of PMSMPS was hence estimated by Eq. (S3) using the averaged sized distribution (as shown in Figures S4 and S6) and  $\rho_{eff}$  in each experiment: 79

 $\sigma_{PM-SMPS}$

$$= \sqrt{\sum_{i=1}^{n} PM_{SMPS}^{2}(d_{m,i}) + \sum_{i=1}^{n} PM_{SMPS}^{2}(dN_{i}) + \sum_{i=1}^{n} PM_{SMPS}^{2}(\rho_{eff,i}) + \sum_{i=1}^{n} PM_{SMPS}^{2}(E_{L}(d_{m,i})) + \sum_{i=1}^{n} PM_{SMPS}^{2}(E_{s,i})}$$
(S3)

where  $PM_{SMPS}(x_i)$  is the product of the uncertainties of each bin  $x_i$  ( $\sigma_{x,i}$ ), and the corresponding sensitivity 83 coefficient  $\frac{\partial PM_{SMPS,i}}{\partial x_i}$ , as shown in Eq. (S4):

$$PM_{SMPS}(x_i) = \frac{\partial PM_{SMPS,i}}{\partial x_i} \cdot \sigma_{x,i}$$
(S4)

Then, Eq (S3) can be simplified as Eq. (S5):

$$\sigma_{PM-SMPS} = \sqrt{\sum_{i=1}^{n} (9\sigma_{dm,i}^{2} + \sigma_{dN,i}^{2} + \sigma_{\rho_{eff,i}}^{2} + \sigma_{EL,i}^{2} + \sigma_{ES,i}^{2}) P M_{SMPS,i}^{2}} = \sqrt{\sum_{i=1}^{n^{*}} 0.0378 P M_{SMPS,i}^{2} + \sum_{i=n^{*}}^{n} 0.0678 P M_{SMPS,i}^{2}}}$$
(S5)

where n\* corresponds to the bins with  $d_m$  larger than 200 nm. Overall, the uncertainty of PMSMPS was estimated to be between  $\pm 19.4\%$  and  $\pm 26.0\%$  with the exact value depending on the size distribution (Figures S4 and S6).

**91 **2.2 RRF uncertainty**

RRF was calculated with Eq. (2) in the main text. The uncertainty of RRF ( $\sigma_{RRF}$ ) was related to PMACSM

and  $PM_{SMPS}$  and was calculated using Eq. (S6).  $PM_{SMPS}$  uncertainty obtained from Section 2.1 of the SI was used to calculate the RRF uncertainty, instead of using the standard deviations of the  $PM_{SMPS}$  measured in the "relative stable state" of chamber studies. While for the case of Q-ACSM, the standard deviation in the stable state was used to evaluate the RRF uncertainty.

$$\sigma_{RRF} = \sqrt{\left(\frac{\partial RRF}{\partial PM_{ACSM}}\right)^2} \sigma_{PM-ACSM}^2 + \left(\frac{\partial RRF}{\partial PM_{SMPS}}\right)^2 \sigma_{PM-SMPS}^2 = \sqrt{\left(\frac{\sigma_{PM-ACSM}}{PM_{SMPS}}\right)^2 + \left(\frac{PM_{ACSM}\sigma_{PM-SMPS}}{PM_{SMPS}^2}\right)^2}$$
(S6)

**98 2.3 CE uncertainty**

For the experiments using AS seeds, the collection efficiencies before and after SOA coating were
calculated using Eq. (S7) and Eq. (S8), respectively:

$$CE_{AS} = \frac{0.5PM_{ACSM-AS1}}{PM_{SMPS-AS}}$$
(S7)

$$CE_{org} = \frac{0.5PM_{ACSM-AS2}}{PM_{SMPS-AS}}$$
(S8)

where,  $PM_{ACSM-AS1}$  and  $PM_{ACSM-AS2}$  represents average Q-ACSM measured mass 104 concentrations before and after SOA coating, respectively.  $PM_{SMPS-AS}$  was the mass concentration of AS 105 measured by SMPS before SOA coating. The default CE is 0.5. The uncertainties of CEAS and CEorg were 106 calculated with the Eq. (S9) and Eq. (S10), respectively.

$$\sigma_{CE-AS} = 0.5 \sqrt{\left(\frac{\sigma_{PM-ACSM-AS1}}{p_{M_{SMPS-AS}}}\right)^2 + \left(\frac{p_{M_{ACSM-AS1}}\sigma_{PM-SMPS-AS}}{p_{M_{SMPS-AS}}}\right)^2}$$

$$\sigma_{CE-org} = 0.5 \sqrt{\left(\frac{\sigma_{PM-ACSM-AS2}}{PM_{SMPS-AS}}\right)^2 + \left(\frac{PM_{ACSM-AS2}\sigma_{PM-SMPS-AS}}{PM_{SMPS-AS}^2}\right)^2}$$

(S10)

(S9)

**111 **3** Size distributions in different experiments**